# Dynamics is what you need for time-series forecasting!

## Abstract

While deep learning is facing a model homogenization across modalities, the usual successful deep models are still challenged by simple ones in the time-series forecasting task. Indeed, our hypothesis is that the nature of this task needs models able to learn the underlying dynamics, which is not often the case. We propose to validate this hypothesis through both systemic and empirical studies. We develop an original `PRO-DYN` nomenclature to analyze existing models through the lens of dynamics. Two observations thus emerge: **1.** under-performing architectures learn dynamics at most partially, **2.** the location of the dynamics block at the model end is of prime importance. We conduct extensive experiments to confirm our observations on a set of performance-varying models with diverse backbones. Results support the need to incorporate a learnable dynamics block and its use as the final predictor.

## 1 Introduction

In recent years, data-driven models, especially deep learning ones, have successfully processed data in various tasks. While specific models were designed regarding the modality, we face a model homogenization (Bommasani et al., 2022): Transformer models (Vaswani et al., 2017) originating from the text modality are becoming state-of-the-art (SOTA) across various fields (Veličković et al., 2018; Dosovitskiy et al., 2021; Chen et al., 2023). Boundaries between modalities are vanishing. However, in the specific case of time-series forecasting, these usual models are still challenged by quite simple ones (Zeng et al., 2023; Xu et al., 2024; Tan et al., 2024).

Most of the text-based models, including RNNs (Elman, 1990), LSTMs (Hochreiter & Schmidhuber, 1997), Transformers (Vaswani et al., 2017), or more recently, State-Space Models (SSMs) (Gu & Dao, 2024), follow the sequence-to-sequence paradigm, turning one sequence into another sequence. They received a lot of success in text generation, fitting quite well to this modality: the bigger the model capacity is, the better the performance, see (Hoffmann et al., 2022) scaling law. As time-series forecasting (TSF) can also be seen as a sequence-to-sequence task, or more precisely, a series-to-series task, the previous text-based models have naturally been adapted to it (Zhou et al., 2021; 2022a). However, it has been shown that these models are well challenged by basic ones, among which the LSTF-Linear models (Zeng et al., 2023) or FITS (Xu et al., 2024). These simple models map the input and output data by a linear layer after a non-learnable pre-processing. Recent SOTA approaches are now built upon the basic linear models, with complex backbones as pre-processing units (Nie et al., 2023; Liu et al., 2024b; Hu et al., 2024; Qiu et al., 2025).

These observations raise two points: **a.** generating time series is inherently different from generating text, even though they have a similar structure;[1] **b.** the problem does not seem to come from using text-based architectures but from the way we use them on this kind of data. To our knowledge no systemic study to explain these observations has been proposed in the literature. A previous work on Transformer failure in TSF (Ke et al., 2025) only focused on the attention mechanism, without explaining recent Transformer-based model achievements (Nie et al., 2023; Liu et al., 2024b).

Text-based models were designed to replicate the text generation mechanism (Rayner, 1998; Cheng et al., 2016). **We argue that TSF models should replicate the time-series generation mechanism**. This mechanism is, in a majority of fields, modeled as a data evolution law, called a dynamical

---

[1]In classification or anomaly detection, we don't observe this phenomenon, see results in (Xu et al., 2024)

system, a priori known in physics (Raissi et al., 2019; Li et al., 2021; Kovachki et al., 2023) or economics (Liu et al., 2024a) or estimated a posteriori (Shojaee et al., 2025). It legitimates the modeling of a time-series evolution by its underlying dynamics. **We thus hypothesize that TSF models should be able to learn a time-series dynamics**.

This work focuses on the study of this hypothesis. We develop an original nomenclature named `PRO-DYN`. It enables making explicit how dynamics is involved in a model (`DYN` function), surrounded by pre- and post-processing units (`PRO` functions). We explicit the dynamics learned by LSTF-Linear models (Zeng et al., 2023). We then perform a systemic study of existing TSF models (Qiu et al., 2024). We derive two main observations: **1.** under-performing models have no (or partial) learnable dynamics modeling (supporting our hypothesis); **2.** SOTA architectures do learn a dynamics (again supporting our hypothesis), combining deep blocks as pre-processing units and a dynamics block at the end as the predictor, giving clues to model design considerations.

To study empirically the first observation, we incorporate linear dynamics, without any structural hyperparameter modification, into targeted models which have no or partial dynamics modeling capabilities: two Transformer-based ones, Informer (Zhou et al., 2021), FEDformer (Zhou et al., 2022a), the CNN-based MICN (Wang et al., 2023), and the SSM-based FiLM (Zhou et al., 2022b). Our experiments show tangible performance improvements, which support that learnable dynamics modeling capabilities drive the performance. Then, to study the second observation, we add a Linear dynamics layer at the entry of recent SOTA foundation models, iTransformer (Liu et al., 2024b), PatchTST (Nie et al., 2023), and Crossformer (Zhang & Yan, 2023) to employ them as post-processing units, again without any structural hyperparameter modification. Our experiments show that pre-processing-like architectures are the best choice as they take better advantage of longer look-back windows.

## 2 RELATED WORK

**TSF by adapting text-based Transformers**    The main concern when adapting text-based models for time-series forecasting was efficiency; the development of the LogSparse attention was the pioneer work in Transformer-based TSF (Li et al., 2019), but it kept the slow autoregressive process. The most influential work became Informer, where they defined the ProbSparse attention and generated predictions in one forward pass (Zhou et al., 2021). Models like Autoformer (Wu et al., 2021), FEDformer (Zhou et al., 2022a), and Non-stationary Transformers (Liu et al., 2022), inherited from Informer computations: initialize a decoder with a simple non-learnable prediction (mean or zero-padding) without any learnable dynamics. Later, these models were beaten by the LSTF-Linear models (Zeng et al., 2023). The focus of complex deep model failure has been on attention mechanism (Zeng et al., 2023; Ke et al., 2025), but recent SOTA models are still attention-based (Nie et al., 2023; Liu et al., 2024b; Qiu et al., 2025). Our work focuses on the learning dynamics capabilities of TSF models, a possible major performance driver.

**Models inheriting from LSTF-Linear models**    LSTF-Linear models (Zeng et al., 2023) were introduced in earlier works on Direct Multi-step (DMS) forecasting (Chevillon, 2005), again for simplicity and efficiency, avoiding accumulated errors from Iterated Multi-step (IMS). They have been automatically adopted by the vast majority of diverse models for their performance and efficiency, as in TiDE (Das et al., 2023), iTransformer (Liu et al., 2024b), or Attraos (Hu et al., 2024), beating previous LSTF-Linear models. To our knowledge, no systemic justification has been proposed to support the integration of Linear functions. Our work proposes one based on Linear learning dynamics capabilities.

**Models in TSF when an a priori is known**    The a priori knowledge on time-series data is usually in the form of dynamics, which defines the relation between current and future states, as Partial Differential Equations (PDEs). It strongly conditions model design as in Physics-Informed Neural Networks (Raissi et al., 2019), which includes the PDE residuals into the loss term, Neural operators (Li et al., 2021; Kovachki et al., 2023), which map initial states and boundary conditions to the PDE solution, and Neural ODEs (Chen et al., 2018; Liu et al., 2024a) which apply the data evolution law to the latent state evolution. A recent work on TSF thus supposes a strong a priori PDE knowledge and combines patching and Neural ODEs (Qi et al., 2024). Different from this work, we hypothesize TSF models should be able to learn a dynamics and analyze how they do so.

## 3 SYSTEMIC ANALYSIS THROUGH THE LENS OF DYNAMICS

**Time-series and time-series space** We consider a time-series $\mathbf{X} = \{x_d(t_1), \ldots, x_d(t_L)\}_{d=1}^{D} \in \mathbb{R}^{L \times D}$ which is the historical data of $D$ variates along $L$ regularly sampled timestamps $t_i \in \mathbb{R}^+$ with $t_i < t_j, \forall i, j \in \{1, \ldots, L\} | i < j$. A time-series space $\mathfrak{T}$ is a real-valued space from the product of a time interval $\mathcal{T} \subset [0, +\infty[$ and a latent space dimension $\mathcal{D} \in \mathbb{N}$ (we consider time-series with multiple univariate series; for 2D-videos, values would be $[0; 255]^3$, with $\mathcal{D} \in \mathbb{N}^2$ - horizontal and vertical axis -). A time interval of a time-series $\mathbf{X}$ is the smallest interval containing $\{t_1, \ldots, t_N\}$. With $\mathcal{T}_{\mathbf{X}} = [t_1, t_L]$, the historical time interval of $\mathbf{X}$, and $\mathcal{D}_{\mathbf{X}} = D$ the number of studied variates, the time-series space of $\mathbf{X}$ is $\mathfrak{T}_{\mathbf{X}} = \mathbb{R}^{\mathcal{T}_{\mathbf{x}} \times \mathcal{D}_{\mathbf{x}}}$.

**TSF task** The time-series forecasting task of $\mathbf{X}$ is to infer the $H$ future timestamps $\mathbf{Y} = \{\tilde{x}_d(t_{L+1}), \ldots, \tilde{x}_d(t_{L+H})\}_{d=1}^{D}$ based on the $L$ historical ones, i.e. $\mathbf{X}$. We denote $\mathfrak{T}_{\mathbf{Y}} = \mathbb{R}^{\mathcal{T}_{\mathbb{Y}} \times D}$ the time-series space of $\mathbf{Y}$, with $\mathcal{T}_{\mathbf{Y}} = [t_{L+1}, t_{L+H}]$ the prediction time interval.

**Dynamical systems** Based on a current system state $\mathbf{x}(t) \in \mathbb{R}^D$ at time $t \in \mathbb{R}^+$, the system is called *dynamic* when there exists an evolution function $\Phi : \mathbb{R}^+ \times \mathbb{R}^+ \times \mathbb{R}^D \to \mathbb{R}^D$ mapping the current state $\mathbf{x}(t)$ to the future states at $t + \tau, \forall \tau \in \mathbb{R}^+$ such as: $\Phi(t, \tau, \mathbf{x}(t)) = \mathbf{x}(t + \tau)$. It defines a direct link between current observations and future ones (dynamics can evolve through time).

### 3.1 THE PRO-DYN NOMENCLATURE

Our nomenclature is based on how computations are performed regarding time in TSF models.

Let $\mathfrak{T}_{\mathcal{E}} = \mathbb{R}^{\mathcal{T}_{\mathcal{E}} \times d_{\mathcal{E}}}$ a time-series space. We consider a function $f$ mapping $\mathfrak{T}_{\mathcal{E}}$ to another time-series space $\mathfrak{T}_{\mathcal{F}} = \mathbb{R}^{\mathcal{T}_{\mathcal{F}} \times d_{\mathcal{F}}}$. Depending on the task assigned to $f$, it involves temporal relations between $\mathcal{T}_{\mathcal{E}}$ and $\mathcal{T}_{\mathcal{F}}$. In this paper, we rely on the popular Allen's interval algebra (Allen, 1983) that defines relations between two time intervals (illustrated in Figure 1 and Appendix A). We introduce the notions of PRO (PROcessing) and DYN (DYNamics) function, based on Allen's temporal interval relations: $f$ is PRO if and only if $\mathcal{T}_{\mathcal{E}}$ **contains, started by, finished by, or equals** $\mathcal{T}_{\mathcal{F}}$. $f$ is DYN if and only if $\mathcal{T}_{\mathcal{E}}$ **starts, overlaps, meets, or before** $\mathcal{T}_{\mathcal{F}}$.

Based on time-evolution considerations, we propose to introduce three types of functions that can be used to decompose any model designed for a TSF task. More precisely, we consider that any model $\mathcal{M}_\theta$ designed for a TSF task, with learnable parameters $\theta$, that takes data points in the historical time interval $\mathcal{T}_{\mathbf{X}}$ and outputs predictions in the future $\mathcal{T}_{\mathbf{Y}}$, can be decomposed as follows:

$$\mathcal{M}_\theta : \mathbf{X} \xrightarrow[f_{\theta_{pre}}^{pre}]{} \mathbf{X}_{pre} \xrightarrow[f_{\theta_{dyn}}^{dyn}]{\oplus\{\mathbf{X}\}} \mathbf{X}_{dyn} \xrightarrow[f_{\theta_{post}}^{post}]{\oplus\{\mathbf{X}, \mathbf{X}_{pre}\}} \mathbf{Y} \tag{1}$$

where $f_{\theta_{dyn}}^{dyn}$ (in orange), a DYN function, defines $\mathcal{M}_\theta$ dynamics performing a prediction going from $\mathcal{T}_{\mathbf{X}}$ to $\mathcal{T}_{\mathbf{Y}}$ (or $\mathcal{T}_{\mathbf{X}} \to \mathcal{T}_{\mathbf{X}} \cup \mathcal{T}_{\mathbf{Y}}$ in a start/overlap case); $f_{\theta_{pre}}^{pre}, f_{\theta_{post}}^{post}$, two PRO functions[2], are pre and post (relatively to $f_{\theta_{dyn}}^{dyn}$) processing functions, performing computations while staying in their input time interval. $\oplus\{\}$ over arrows illustrate eventual skip connexions. We illustrate our framework in Figure 1 for univariate time-series ($D = 1$) for the sake of simplicity.

Based on this, we introduce our original PRO-DYN nomenclature. For any TSF model :

1. we decompose it as a composition of PRO and DYN functions;
2. we identify the nature of the DYN function;
3. we identify the backbone of the PRO functions.

From the PRO-DYN nomenclature, we first analyze the LSTF-Linear models (Zeng et al., 2023), the basic models challenging deep complex ones, through the lens of dynamics.

---

[2]Non-learnable invertible step employed both at the entry and inverted in the end (e.g. normalization) are not taken into account.

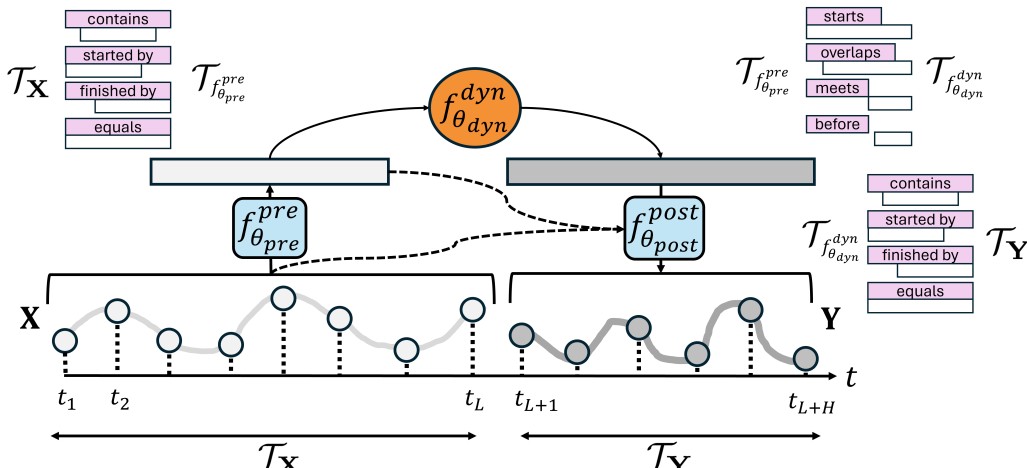

Figure 1: `PRO` and `DYN` functions illustrated in the processing chain of a TSF model $\mathcal{M}_\theta$ where "$\mathcal{T}_{\mathbf{X}}$ **before** $\mathcal{T}_{\mathbf{Y}}$". `PRO` functions are framed and blue while `DYN` function is encircled and orange. Solid lines represent the main data flow. $f_{\theta_{post}}^{post}$ can be fed by $\mathbf{X}$ or/and $f_{\theta_{pre}}^{pre}(\mathbf{X})$ (dotted lines). Dotted line from $\mathbf{X}$ to $f_{\theta_{dyn}}^{dyn}$ are not drawn for better clarity. Allen relations defined by each function based on its type are illustrated, where the blue interval (resp. white) corresponds to the input - left - (resp. output - right -) interval. The relation name is written in each blue interval.

## 3.2 LSTF-LINEAR DYNAMICS

Without loss of generality, let us suppose $H \leq L$ (see Appendix B for the $H > L$ case). Let denote $\mathbf{X}(t_L) = \{x_d(t_1), \ldots, x_d(t_L)\}_{d=1}^{D}$ the $L$ previous timestamps (the look-back window) of $\mathbf{X}$ from $t = t_L$. LSTF-Linear models are the composition of: a non-learnable invertible pre-processing step $g_{pre} : \mathbf{X}(t_L) \mapsto \mathbf{X}_g(t_L)$ (where $g_{pre}$ is identity for Linear, normalization for NLinear, or seasonal-trend decomposition[3] for DLinear), a learnable `DYN` function `Linear`$_\theta$ - highlighted in orange -, and a non-learnable post-processing step $g_{post} = g_{pre}^{-1}$ such as:

$$\mathbf{X}(t_{L+H})_{|_{[t_{L+1}, t_{L+H}]}} = g_{pre}^{-1} \circ \boxed{\texttt{Linear}_\theta} \circ g_{pre}(\mathbf{X}(t_L)) = g_{pre}^{-1}(\boxed{W_\theta}\, \mathbf{X}_g(t_L) + \boxed{b_\theta}) \quad (2)$$

where $\mathbf{X}(t_{L+H})_{|_{[t_{L+1}, t_{L+H}]}}$ is the extraction of $\mathbf{X}(t_{L+H})$ on the prediction time interval $[t_{L+1}, t_{L+H}]$; $W_\theta \in \mathbb{R}^{H \times L}$ and $b_\theta \in \mathbb{R}^{H \times D}$ - a $\mathbb{R}^H$ vector repeated $D$ times along the variate dimension - are the parameters of `Linear`$_\theta$ (for DLinear, $g_{pre}^{-1}$ is the sum of seasonal and trend linear layer outputs). These parameters are, respectively, in terms of dynamics, the dynamics matrix, and an external force applied to the system. LSTF-Linear models training corresponds to studying one iteration of this dynamics at different timestamps $t_L$. **LSTF-Linear models do have learnable dynamics modeling capabilities, which would explain their performance.**

## 3.3 TSF MODELS THROUGH THE `PRO-DYN` NOMENCLATURE

The goal here is to identify features from the `PRO-DYN` nomenclature driving model performance. We analyze deep models tested in the benchmark (Qiu et al., 2024) (chosen for its diversity of datasets). We end up with Table 1, keeping the same row-order performance as in the benchmark on the multivariate TSF task.

There are from Table 1 two *performance-based* groups: models better ($\uparrow$) than NLinear (chosen as the reference as it is the best performing simple model), and models worse ($\downarrow$) than it. From the `PRO-DYN` nomenclature, models $\uparrow$ have two features (identified with green color) in common:

---

[3]The trend component $\mathbf{T}$ is a moving average over the input and the seasonal component $\mathbf{S}$ is the input without the trend component, such as $\mathbf{X} = \mathbf{S} + \mathbf{T}$.

Table 1: TSF deep models through the `PRO-DYN` nomenclature. ↑ (resp. ↓) are models better (resp. worse) than NLinear. Tf. stands for Transformer, CNN for Convolution Neural Network, MLP for MultiLayer Perceptron, 0-pad. for 0-padding, discr. for discretization, NS for Non-stationary. Colors correspond to features identified to drive (green) or drag (magenta) the performance on the TSF task.

| Model | Complete learnable dynamics (RQ1) | Config. (RQ2) | DYN function | PRO backbone | Reference |
|---|---|---|---|---|---|
| **DUET** ↑ | ✓ | PRE-DYN | Linear | Transformer | (Qiu et al., 2025) |
| **PDF** ↑ | ✓ | PRE-DYN | Linear | Tf. & CNN | (Dai et al., 2024) |
| **Pathformer** ↑ | ✓ | PRE-DYN | Linear | Transformer | (Chen et al., 2024) |
| **iTransformer** ↑ | ✓ | PRE-DYN | Linear | Transformer | (Liu et al., 2024b) |
| **PatchTST** ↑ | ✓ | PRE-DYN | Linear | Transformer | (Nie et al., 2023) |
| **Crossformer** ↑ | ✓ | PRE-DYN | Linear | Transformer | (Zhang & Yan, 2023) |
| **TimeMixer** ↑ | ✓ | PRE-DYN | Linear | MLP | (Wang et al., 2024) |
| NLinear | ✓ | DYN | Linear | None | (Zeng et al., 2023) |
| *TimesNet* ↓ | ✓ | PRE-DYN-POST | Linear | CNN | (Wu et al., 2023) |
| *FITS* ↓ | ✗ | PRO-DYN | Linear & 0-pad. | Filtering | (Xu et al., 2024) |
| *FEDformer* ↓ | ✗ | PRE-DYN-POST | Mean & 0-pad. | Transformer | (Zhou et al., 2022a) |
| **Triformer** ↓ | ✓ | PRE-DYN | Linear | Transformer | (Cirstea et al., 2022) |
| *MICN* ↓ | ✗ | PRE-DYN-POST | Linear & 0-pad. | CNN | (Wang et al., 2023) |
| *FiLM* ↓ | ✗ | PRE-DYN-POST | Legendre discr. | SSM | (Zhou et al., 2022b) |
| *Informer* ↓ | ✗ | PRE-DYN-POST | 0-padding | Transformer | (Zhou et al., 2021) |
| *NS Transformer* ↓ | ✗ | PRE-DYN-POST | 0-padding | Transformer | (Liu et al., 2022) |

a **complete learnable `DYN` function** and a **`PRO` function for pre-processing only**, while in the second group, the main shared features (identified with magenta color) are an, at most **partially**, non-learnable `DYN` function and learnable `PRO` functions for both pre- and post-processing.

We thus identify, directly in Table 1, two *feature-based* groups: in green/bold, models with two green features and, in magenta/italic, models with at least one magenta feature. They almost coincide (see conclusion for Triformer case) with the performance-based groups. We thus derive two observations: **1.** a (partially) non-learnable `DYN` function drowns the performance, and **2.** a learnable `PRO` function for pre-processing just before the final `DYN` function drives the performance.

We derive these two observations into two research questions (RQ) to validate them experimentally:

- (**RQ1**) Can we enhance model performance by adding a full learnable dynamics?

- (**RQ2**) Is (Pre-processing)-`DYN` the best-performing configuration? If so, why?

**(RQ1) Dynamics addition** We choose to study Informer (Zhou et al., 2021), FiLM (Zhou et al., 2022b), MICN (Wang et al., 2023), and FEDformer (Zhou et al., 2022a), for their diverse performance, `DYN` functions, and backbones. We incorporate full learnable dynamics, always after the eventual non-learnable normalization step, for prediction in these models by adding a linear `DYN` layer ($\text{Linear}_\theta$ in Section 3.2), while keeping the original structures (see Figure 2):

- for Informer, a Transformer-based model, we feed the decoder with the encoder output, which is processed by a linear `DYN` layer. It replaces the zero-padding;

- for FiLM, an SSM-based model, we add a linear `DYN` layer at the model entry. FiLM turns into a `PRO` post-processing block;

- for MICN, a CNN-based model, we feed the seasonal block with the seasonal component processed by the trend block, which is a linear `DYN` layer, replacing the zero-padding;

- for FEDformer, a Transformer-based model, we add a linear `DYN` layer before the encoder embedding layer, while recomputing the end of the input $\mathbf{X}_{trunc}$ to fit the original decoder temporal embedding size. The decoder input (zero-padding for the seasonality $\mathbf{S}$ and input mean for the trend $\mathbf{T}$) is not changed, but Keys ($\tilde{\mathbf{K}}$) and Values ($\tilde{\mathbf{V}}$) are now computed from initial predictions performed by the added learnable `DYN` function.

Better performances of the `DYN` versions of the chosen models would answer positively to RQ1. The variety of studied models and locations to incorporate dynamics would validate the generality of dynamics considerations.

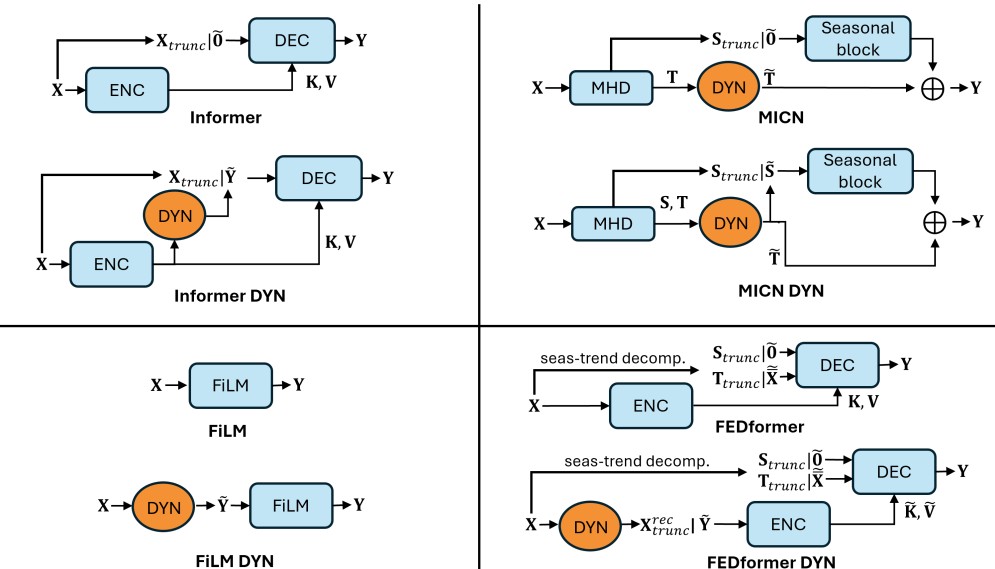

Figure 2: RQ1 models with now full learnable dynamics capabilities. DYN is linear dynamics layer, ENC-DEC is encoder-decoder, MHD is multi-scale hybrid decomposition (defined in MICN, see (Wang et al., 2023)). $\mathbf{A}|\mathbf{B}$ means $\mathbf{A}$ concatenated with $\mathbf{B}$ along the time axis, Tilde refers to intermediate variables in the time prediction interval, $\tilde{\mathbf{Y}}$ an intermediate prediction from a DYN function, $\overline{\mathbf{X}}$ is the mean of $\mathbf{X}$, $trunc$ subscript corresponds to input start tokens.

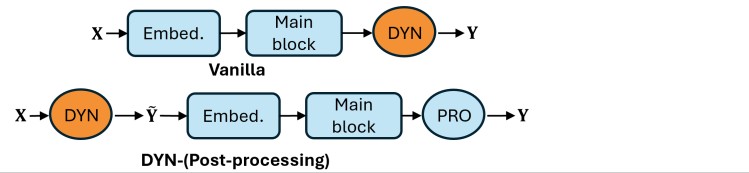

Figure 3: RQ2 models general derivation in their post-processing configuration. For iTransformer, part of the input is concatenated to the DYN output. Embed. stands for embedding layer.

**(RQ2) DYN-(Post-processing) configuration** We identify three well-performing foundation models for their diversity in time consideration and structure: iTransformer (Liu et al., 2024b), an encoder-only model where time and variate dimensions are inverted, considers time as a latent dimension, losing its sequential property; PatchTST (Nie et al., 2023), a patching-based method, thus respecting time sequentially, in an encoder-only fashion; Crossformer (Zhang & Yan, 2023), also a patching-based method, with an encoder-decoder serving as a PRO function. In each model, illustrated in Figure 3, we add a learnable linear DYN layer just before the embedding one, without removing the linear DYN layer at the end, which becomes a linear PRO layer (not possible to remove it while keeping the same model design and hyperparameters): the main computation block acts now as a post-processing function. Only the DYN output feeds PatchTST and Crossformer, while part of the input is concatenated to it to feed iTransformer. A performance drop in the modified models would answer positively to RQ2.

## 4 EXPERIMENTS

We conduct extensive experiments to answer RQ1 and RQ2. Modified models in RQ1 are referred to as "DYN added models", while ones in RQ2 are referred to as "post-processing models". Original models are referred to as "vanilla".

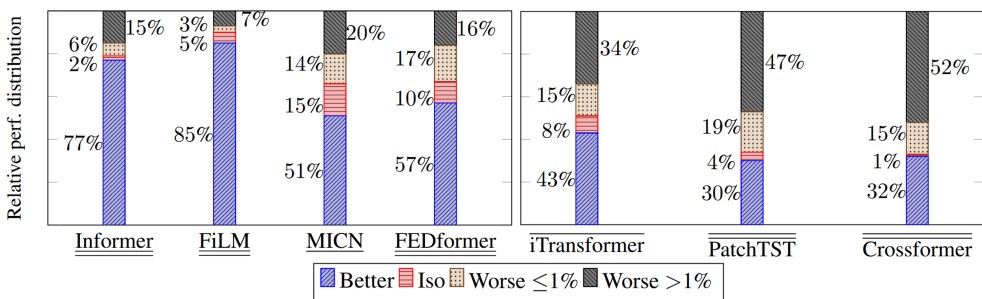

Figure 4: Performance distribution of RQ1 (left) and RQ2 (right) models. A model name is underlined (resp. double-underlined) when the `DYN` added model is statistically better than its vanilla version on either MSE or MAE (resp. both). Similarly, it is overlined (resp. double-overlined) when the vanilla model is statistically better than the post-processing version on one (resp. both) metric.

### 4.1 EXPERIMENTAL SETUP

**Datasets**   We consider TSF on the 25 datasets of the TFB benchmark (Qiu et al., 2024), with 4 forecasting horizons each, including the well-established ETTs, Exchange, Weather, Electricity, ILI, Traffic, and Solar datasets (Wu et al., 2021). Details on datasets are shown in Appendix C.

**Settings**   To evaluate the TSF task, we compute the Mean Square Error (MSE) and Mean Average Error (MAE) across each dataset and forecasting horizon (200 scores per model) for the modified models and compare them to the vanilla results obtained in the TFB benchmark. We keep the same architecture hyperparameters as their vanilla versions. We only adjust by hand learning hyperparameters (epochs, learning rate, patience), and also test them on their vanilla versions for fair comparison. Each configuration can be found in the code, and the implementation details are in Appendix D. Raw results can be found in Appendix E.1, and prediction visualizations in Appendix F.

**Results**   We count the number of cases where the modified models are better, equal (iso), worse by at most 1% (low degradation), or worse by at least 1%, than their vanilla version. For MSE and MAE, the lower, the better. Resulting distributions are shown in Figure 4. For RQ1 `DYN` added models, we compute the p-values of the unilateral Wilcoxon test to assess if the MSE and MAE are lower than the vanilla versions with statistical significance (p-value $< 0.05$). For RQ2 models, we perform the opposite test to assess if the vanilla versions are better than the post-processing ones with statistical significance. In addition, for RQ1, we compute in Table 2 the average score normalized by NLinear to measure the quantitative impact of the `DYN` addition, while comparing each model to the NLinear baseline. Detailed results can be found in Appendix E.2.

### 4.2 FIRST ANALYSIS

**RQ1**   Results seem to **support having full learning dynamics capabilities** for TSF. Indeed, from Figure 4, all `DYN` added models **are better or comparable in more than** 80% **of the cases** than their vanilla versions, and **better with statistical significance** on at least one metric. In particular, Informer and FiLM are greatly improved by the linear `DYN` layer addition. With just a data flow update, MICN gets better or equal scores 66% of the time. Moving to Table 2, for each model, **the mean performance is better** with the `DYN` layer addition, supporting our model update. Moreover, FiLM `DYN` gets slightly better results than NLinear on average. However, `DYN` models (except FiLM) are still worse than NLinear - normalized scores below zero -, where possible reasons are: keeping the same hyperparameters is constraining, or `DYN` models are not in a pre-processing configuration (not feasible to do so while keeping the same original architecture).

**RQ2**   Results from Figure 4 **support the (Pre-processing)-`DYN` configuration** as PatchTST and Crossformer in the `DYN`-(Post-processing) version **get worse results with statistical significance**. However, post-processing iTransformer is only statistically worse on one metric, with better or equal results in 51% of the time, supporting the possibility of using such models as post-processing blocks. As PatchTST and Crossformer take into account the time sequential aspect in their computations,

Table 2: Comparison of average performance normalized by NLinear scores: $\frac{\text{score(NLinear)} - \text{score(Model)}}{\text{score(NLinear)}}$, where score is MSE or MAE, for each dataset and forecasting horizon. Best score between `DYN` added model and its vanilla version is in **bold**. Some outliers are removed for consistency (see Appendix E.2 for further details).

| Model | Informer | FiLM | MICN | FEDformer |
|---|---|---|---|---|
| DYN added | $\mathbf{-0.228}$ | **0.006** | $\mathbf{-0.164}$ | $\mathbf{-0.360}$ |
| Vanilla | $-0.333$ | $-0.036$ | $-0.176$ | $-0.398$ |

they are more responsive to modifications with a temporal meaning, explaining the difference with iTransformer in response to the `DYN` layer addition.

**Intermediate conclusion** Overall, current results seem to answer positively to both RQs. However, adding a linear layer comes with two side effects: a slight parameter addition and data length modification, which have an impact on performance. The modified models (except Informer and MICN) process inputs of different lengths than vanilla ones. We thus conduct an additional study to identify the performance drivers.

## 4.3 PERFORMANCE DRIVER ANALYSIS

We perform additional experiments to isolate two possible performance drivers, which are side effects of the `DYN` layer addition:

- **parameter addition** can be the principal performance driver in the RQ1 case. To isolate that side effect, we compare `DYN` added models to their `PRO` added version, in which the added linear `DYN` layer is replaced by a feed-forward one which does not change the time dimension, turning it into a linear `PRO` layer. MICN is excluded here as no layer is added. For Informer `PRO`, we either pad with zeros if $H > L$ or truncate the output if $H < L$ to fit the decoder input dimension. RQ2 vanilla models are still compared to their post-processing versions, as it is a worst-case scenario for (Pre-processing)-`DYN` configuration;

- **data length variation** due to differences between context and horizon size can be a performance driver. We thus condition the results on three possible setups: $(H > L)$, $(H = L)$, and $(H < L)$, and analyze distribution shifts. The setup repartition varies from one model to another ($L$ is a hyperparameter in the benchmark). For the $(H = L)$ setup, `DYN` and `PRO` are the same, except for FEDformer due to temporal embeddings and $\mathbf{X}_{trunc}^{rec}$ addition.

**Results** To perform the comparison, `PRO` versions are trained with the same hyperparameters as `DYN` ones. We count the number of cases when each `DYN` (resp. post-processing) model is better, equal (iso), or worse than its `PRO` (resp. vanilla) version on MSE and MAE. Global and conditioned distributions with cases when p-values are below 0.05 are shown in Figure 5. Detailed results are shown in Appendix E.3. Conditioned comparisons between `DYN` added models and their vanilla versions are presented in Appendix E.4. Prediction visualizations are shown in Appendix F.

**Preliminary observation** We observe from Figure 5 that **when models process greater data length, they get an advantage**. As the setup distribution is not uniform across each model, results are biased. In the following, if distributions are inverted from $(H > L)$ to $(H < L)$, the performance would then be driven by data length variation. If it is in favor of `DYN` (resp. RQ2 vanilla) model even when it is disadvantaged, then the performance gain is mainly driven by the learnable dynamics capabilities (resp. dynamics block located at the end).

**RQ1** Results confirms that **the performance mainly comes from the dynamics**. Indeed, from Figure 5, for Informer and FEDformer, overall, `DYN` versions are statistically better than their `PRO` versions. For Informer, `DYN` and `PRO` are statistically similar on $(H < L)$ while `DYN` should be disadvantaged (fewer added parameters). For FEDformer, `DYN` is statistically better in both setups. For $(H = L)$, FEDformer `DYN` and `PRO` are statistically similar: the timestamp embedding, in line with (Zeng et al., 2023), doesn't influence the performance, and the recomputed $\mathbf{X}_{trunc}^{rec}$ doesn't give any data length advantage. On the contrary, for FiLM, the `DYN` version is not statistically better than

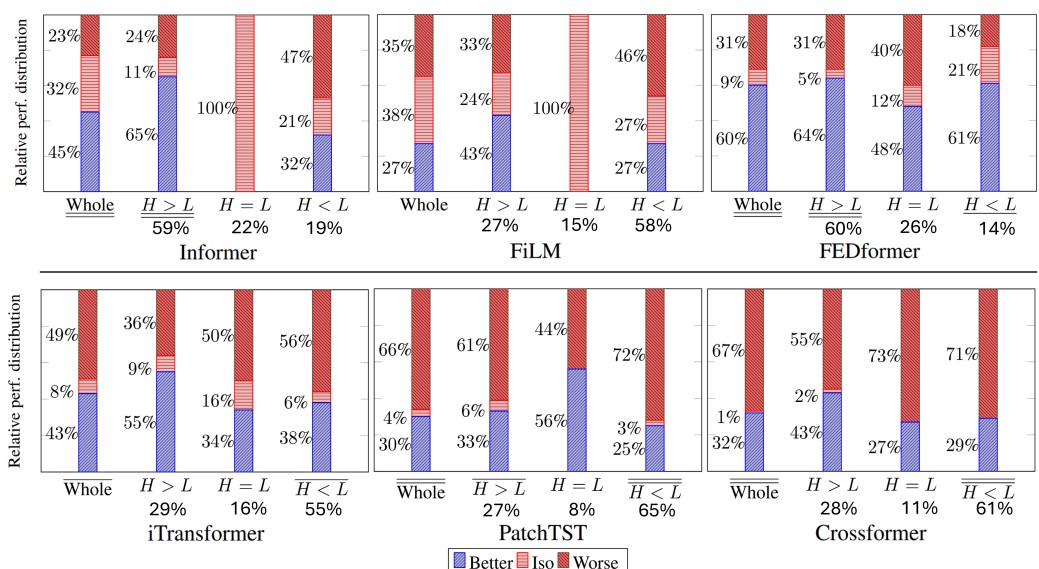

Figure 5: `DYN` added model performance distribution against their `PRO` version (up); Post-processing model performance distribution against their vanilla version (down), with setup conditioning. *Whole* bar is the overall distribution. Different from Figure 4, *Worse* $\leq 1\%$ and *Worse* $> 1\%$ are put together under *Worse*. Again, a setup is underlined (resp. double-underlined) when the `DYN` model is statistically better than the `PRO` one on either (resp. both) MSE or MAE. It is over-lined (resp. double-overlined) when the vanilla version is statistically better than the post-processing one on one (resp. both) metric. Setup distribution for each model is shown under the proper case.

the `PRO` one, with a symmetry in the distributions and a majority of $(H < L)$ disadvantaged cases for `DYN`: the performance against the vanilla version comes from the parameter addition. Indeed, the `DYN` layer **could be in conflict with its SSM encoding part**, which learns a dynamics (Gu & Dao, 2024).

**RQ2** From the experiments, it emerges that predicting points based on a greater number of observations is an advantageous setup **when there are learning dynamics capabilities located at the model end**. Indeed, from Figure 5, an overall tendency emerges: vanilla models are similar to post-processing versions when $(H \geq L)$ while surpassing them with statistical significance in the $(H < L)$ setup. RQ1 `PRO` models, without learnable dynamics capabilities, don't surpass `DYN` models when $(H < L)$, while they should be advantaged. Post-processing models don't take advantage of larger data length against vanilla ones (especially for PatchTST), confirming the superiority of the (Pre-processing)-`DYN` configuration.

## 5 CONCLUSION AND FUTURE WORK

This work considers TSF models through the lens of dynamics. We propose the original `PRO-DYN` nomenclature, identify the dynamics defined by LSTF-Linear models, then assess which features can contribute the most to model performance, which emerge to be **1.** the ability to learn a dynamics, **2.** located at the end of the model. We perform experiments validating the hypothesis that **models should be able to learn dynamics**, showing they take **better advantage of a longer look-back window with learning dynamics capabilities**: they are powerful (Zeng et al., 2023). We limit our `DYN` function consideration to Linear dynamics, but future work should explore richer `DYN` functions (e.g. autoregressive mechanisms) and extend beyond Transformers to various backbones as SSMs where dynamics arises naturally. Moreover, results against NLinear and Triformer position suggest performance depends not only on dynamics but also on the choice of `PRO` functions and computations along the time dimension. Finally, the role of dataset domains in shaping underlying dynamics remains an open question for future study.

ETHICS STATEMENT

This work is foundational research on already existing open-source models, tested on a public benchmark. This work has no impact on society at large, beyond aiming to get a better understanding of deep learning models. This work could reduce model energy consumption by guiding research on dynamics modeling.

**Responsible use of LLMs**    In preparing this manuscript, we occasionally used suggestions from LLMs (GPT-5) to guide improvements in clarity, grammar, and overall readability. All scientific content, including experimental design, codebase, data analysis, results, and interpretations, is independently developed by the authors. LLMs are not involved in generating, modifying, or interpreting any experimental results, nor in producing code or analyses. Their use is strictly limited to selectively refining language to ensure clear and effective communication of our research.

REPRODUCIBILITY STATEMENT

For transparency and reproducibility, as detailed in the supplementary material in Appendix D, we base our code on the public repository developed by the TFB benchmark. We provide, along with the paper, the full code that includes the studied models and their hyperparameters used to evaluate them on the TSF task.

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

## A  ALLEN'S TEMPORAL INTERVAL RELATIONS

We propose a detailed visualization, in Figure 6, inspired by (Mate et al., 2019), illustrating the relations defined by Allen's interval algebra (Allen, 1983) considered in the paper. In our work, with the same notations as in Section 3.1, we study functions going from a temporal interval $\mathcal{T}_\mathcal{E}$ to $\mathcal{T}_\mathcal{F}$. If we denote `rel` an Allen's temporal interval relation, with $\mathcal{T}_\mathcal{E}$ `rel` $\mathcal{T}_\mathcal{F}$, we only consider relations `rel` that stay in the same temporal interval or go into the future (in accordance with the time-series forecasting task), which are displayed in the figure below.

| Relation | Pictoral example $\quad t$ | Function type $f\colon \mathbb{R}^{\mathcal{T}_\mathcal{E} \times d_\mathcal{E}} \to \mathbb{R}^{\mathcal{T}_\mathcal{F} \times d_\mathcal{F}}$ |
|---|---|---|
| $\mathcal{T}_\mathcal{E}$ before $\mathcal{T}_\mathcal{F}$ | $\mathcal{T}_\mathcal{E}$ $\quad$ $\mathcal{T}_\mathcal{F}$ | DYN |
| $\mathcal{T}_\mathcal{E}$ equals $\mathcal{T}_\mathcal{F}$ | $\mathcal{T}_\mathcal{E}$ / $\mathcal{T}_\mathcal{F}$ | PRO |
| $\mathcal{T}_\mathcal{E}$ meets $\mathcal{T}_\mathcal{F}$ | $\mathcal{T}_\mathcal{E}$ $\mathcal{T}_\mathcal{F}$ | DYN |
| $\mathcal{T}_\mathcal{E}$ overlaps $\mathcal{T}_\mathcal{F}$ | $\mathcal{T}_\mathcal{E}$ / $\mathcal{T}_\mathcal{F}$ | DYN |
| $\mathcal{T}_\mathcal{E}$ contains $\mathcal{T}_\mathcal{F}$ | $\mathcal{T}_\mathcal{E}$ / $\mathcal{T}_\mathcal{F}$ | PRO |
| $\mathcal{T}_\mathcal{E}$ starts $\mathcal{T}_\mathcal{F}$ | $\mathcal{T}_\mathcal{E}$ / $\mathcal{T}_\mathcal{F}$ | DYN |
| $\mathcal{T}_\mathcal{E}$ is started by $\mathcal{T}_\mathcal{F}$ | $\mathcal{T}_\mathcal{E}$ / $\mathcal{T}_\mathcal{F}$ | PRO |
| $\mathcal{T}_\mathcal{E}$ finished by $\mathcal{T}_\mathcal{F}$ | $\mathcal{T}_\mathcal{E}$ / $\mathcal{T}_\mathcal{F}$ | PRO |

Figure 6: Allen's temporal interval relations illustration. In our work, we only consider the relations that stay in the same temporal interval or go into the future. DYN cases are highlighted in orange. Input interval $\mathcal{T}_\mathcal{E}$ is in blue, output interval $\mathcal{T}_\mathcal{F}$ is in white. They are juxtaposed to better visualize how they relate along the time axis.

## B  LINEAR DYNAMICS WHEN $H > L$

Following Section 3.2 with the same notations, with here $H > L$, let denote $\mathbf{X}(t_{L+H})$ the $H$ next timestamps (the prediction window) of $\mathbf{X}$ up to $t = t_{L+H}$ as features. We then have:

$$\mathbf{X}(t_{L+H}) = g_{pre}^{-1} \circ \text{Linear}_\theta \circ g_{pre}(\mathbf{X}(t_L)_{|[t_1,t_L]}) = g_{pre}^{-1}(W_\theta \, \mathbf{X}_g(t_L)_{|[t_1,t_L]} + b_\theta) \tag{3}$$

where $\mathbf{X}(t_L)_{|[t_1,t_L]}$ is the extraction of $\mathbf{X}(t_L)$ on the historical time interval $[t_1, t_L]$; $W_\theta \in \mathbb{R}^{H \times L}$ (the dynamics matrix) and $b_\theta \in \mathbb{R}^{H \times D}$ (an external force $\mathbb{R}^H$ repeated $D$ times along the variate dimension), are the parameters of $\text{Linear}_\theta$. Thus, in any case, LSTF-Linear models (Zeng et al., 2023) do have learnable dynamics modeling capabilities.

## C  DETAILS ON THE DATASETS

We provide, in Table 3, the detailed statistics of the datasets used in our experiments. It includes the domain, the sampling frequency, the variate dimension, and the data split for training, validation, and testing.

Table 3: Statistics of multivariate datasets of the TFB benchmark (Qiu et al., 2024) taken from (Qiu et al., 2025).

| Dataset | Domain | Frequency | Lengths | Dim | Split | Description |
|---|---|---|---|---|---|---|
| METR-LA | Traffic | 5 mins | 34,272 | 207 | 7:1:2 | Traffic speed dataset collected from loop detectors in the LA County road network |
| PEMS-BAY | Traffic | 5 mins | 52,116 | 325 | 7:1:2 | Traffic speed dataset collected from the CalTrans PeMS |
| PEMS04 | Traffic | 5 mins | 16,992 | 307 | 6:2:2 | Traffic flow time series collected from the CalTrans PeMS |
| PEMS08 | Traffic | 5 mins | 17,856 | 170 | 6:2:2 | Traffic flow time series collected from the CalTrans PeMS |
| Traffic | Traffic | 1 hour | 17,544 | 862 | 7:1:2 | Road occupancy rates measured by 862 sensors on San Francisco Bay area freeways |
| ETTh1 | Electricity | 1 hour | 14,400 | 7 | 6:2:2 | Power transformer 1, comprising seven indicators such as oil temperature and useful load |
| ETTh2 | Electricity | 1 hour | 14,400 | 7 | 6:2:2 | Power transformer 2, comprising seven indicators such as oil temperature and useful load |
| ETTm1 | Electricity | 15 mins | 57,600 | 7 | 6:2:2 | Power transformer 1, comprising seven indicators such as oil temperature and useful load |
| ETTm2 | Electricity | 15 mins | 57,600 | 7 | 6:2:2 | Power transformer 2, comprising seven indicators such as oil temperature and useful load |
| Electricity | Electricity | 1 hour | 26,304 | 321 | 7:1:2 | Electricity records the electricity consumption in kWh every 1 hour from 2012 to 2014 |
| Solar | Energy | 10 mins | 52,560 | 137 | 6:2:2 | Solar production records collected from 137 PV plants in Alabama |
| Wind | Energy | 15 mins | 48,673 | 7 | 7:1:2 | Wind power records from 2020-2021 at 15-minute intervals |
| Weather | Environment | 10 mins | 52,696 | 21 | 7:1:2 | Recorded every for the whole year 2020, which contains 21 meteorological indicators |
| AQShunyi | Environment | 1 hour | 35,064 | 11 | 6:2:2 | Air quality datasets from a measurement station, over a period of 4 years |
| AQWan | Environment | 1 hour | 35,064 | 11 | 6:2:2 | Air quality datasets from a measurement station, over a period of 4 years |
| ZafNoo | Nature | 30 mins | 19,225 | 11 | 7:1:2 | From the Sapflux data project includes sap flow measurements and environmental variables |
| CzeLan | Nature | 30 mins | 19,934 | 11 | 7:1:2 | From the Sapflux data project includes sap flow measurements and environmental variables |
| FRED-MD | Economic | 1 month | 728 | 107 | 7:1:2 | Time series showing a set of macroeconomic indicators from the Federal Reserve Bank |
| Exchange | Economic | 1 day | 7,588 | 8 | 7:1:2 | ExchangeRate collects the daily exchange rates of eight countries |
| NASDAQ | Stock | 1 day | 1,244 | 5 | 7:1:2 | Records opening price, closing price, trading volume, lowest price, and highest price |
| NYSE | Stock | 1 day | 1,243 | 5 | 7:1:2 | Records opening price, closing price, trading volume, lowest price, and highest price |
| NN5 | Banking | 1 day | 791 | 111 | 7:1:2 | NN5 is from banking, records the daily cash withdrawals from ATMs in UK |
| ILI | Health | 1 week | 966 | 7 | 7:1:2 | Recorded indicators of patients data from Centers for Disease Control and Prevention |
| Covid-19 | Health | 1 day | 1,392 | 948 | 7:1:2 | Provide opportunities for researchers to investigate the dynamics of COVID-19 |
| Wike2000 | Web | 1 day | 792 | 2,000 | 7:1:2 | Wike2000 is daily page views of 2000 Wikipedia pages |

# D    CODE, DESIGN, AND IMPLEMENTATION DETAILS

**Code**    The code is shared with the supplementary material as a compressed folder. A Readme file can be found describing the repository. The code is based on the repository[4] developed for the TFB benchmark (Qiu et al., 2024) under the MIT license. In the `.\ts_benchmark\ baselines\time_series_library` folder, we develop our models and insert them in the `models` folder to use the same methods as the studied vanilla models defined in the `adapters_ for_transformers.py` file. Results from the vanilla models were taken from the OpenTS multivariate time-series leaderboard[5], a dashboard developed for the TFB benchmark. Everything was done in accordance with the MIT license, which grants the right to use the code and datasets of the TFB benchmark without restriction.

**`DYN` added model design**    We studied multiple ways to add the linear `DYN` layer. We first tested the performance of the `DYN` and post-processing models on the ILI, NASDAQ, and NYSE datasets, the three lightest ones. When each model got comparable or better results to the vanilla version, while being diverse in the way the linear `DYN` layer was added, we extended the experiments to all the other datasets.

**Resources and implementation**    All the introduced and rerun models were trained on CentOs 7.9.2009, on either:

- Intel Xeon Gold 6230 20C @ 2.1GHz with 768 GB memory with up to four NVIDIA Tesla V100 with 32 GB GRAM,

- Intel Xeon Gold 6346 16C @ 3.1GHz with 1024 GB memory with up to four Nvidia HGX A100 with 40 GB memory.

Training was done in PyTorch [6], using the learning procedure developed in the `adapters_for_transformers.py` file: MSE loss criterion, Adam optimizer, learning rate divided by two from one epoch to another. Hyperparameters can be found in the `.\scripts` folder presented in `.sh` files. Datasets are split following the rolling forecast method to prevent information leakage from the future to the past. We needed at most 4 GPUs with 30 CPUs per task, running for 26 hours, on the PEMS-BAY dataset (the largest of the benchmark) for FiLM-based models, for $H = 192$ and $H = 336$. Experiments on the seven largest datasets (PEMS-BAY, Traffic, Electricity, Solar, METR-LA, PEMS04, PEMS08) could run up to 15 hours on 4 GPUs

---

[4]https://github.com/decisionintelligence/TFB

[5]https://decisionintelligence.github.io/OpenTS

[6]Torch version was 1.12.1 with cuda toolkit 11.2.0 and Python 3.9.7

with 15 CPUs per task. Otherwise, each experiment runs in several minutes on at most 2 GPUs with 15 CPUs per task.

**PatchTST-PEMS04 resource issue** We have not been able to run PatchTST post-processing `DYN` on the PEMS04 dataset with $H = 720$ due to memory limit issues with our experimental setup. In order to fill this gap, we set results, for the post-processing version, better by $0.001$ on both MSE and MAE than the vanilla one for the comparison in Sections 4.2 and 4.3. Doing so, we get a worst-case scenario for our analysis (see Table 9). No matter the results obtained on this particular case, the tendencies and thus the analysis remain the same for PatchTST.

# E   DETAILED RESULTS

## E.1   RAW RESULTS

In this section, MSE and MAE results are shown. Vanilla model results are directly copied from the TFB benchmark (Qiu et al., 2024). Post-processing models from RQ2 are identified as *DYN-Post* model to refer to post-processing arrangement with a linear `DYN` layer at the beginning. When the `DYN` version is better than the vanilla one, the `DYN` score is in **bold**. For RQ1, when the `PRO` version is better than the `DYN` one, the `PRO` score is underlined. When a vanilla score is starred*, it means the shown result is a better one, obtained with our updated learning hyperparameters, than in the TFB benchmark. For each table, NLinear performance (copied from the TFB benchmark) is added as a reference. Context length $L$ and horizon length $H$ are also shown.

Table 4: MSE and MAE scores of Informer-based models. The lower, the better. DYN score is in **bold** when it is better than the vanilla version. PRO score is underlined when it is better than the DYN one. Vanilla score is starred* when we obtained better results than the TFB benchmark with updated learning hyperparameters.

| Dataset | Cont. L | Hor. H | Informer DYN MSE | MAE | Informer PRO MSE | MAE | Informer MSE | MAE | NLinear MSE | MAE |
|---|---|---|---|---|---|---|---|---|---|---|
| ETTh1 | 96 | 96 | **0.503** | **0.480** | 0.503 | 0.480 | 0.715 | 0.571 | 0.385 | 0.403 |
| | 96 | 192 | **0.558** | **0.507** | 0.623 | 0.535 | 0.726 | 0.574 | 0.422 | 0.426 |
| | 96 | 336 | **0.597** | **0.528** | 0.727 | 0.583 | 0.741 | 0.588 | 0.431 | 0.429 |
| | 96 | 720 | **0.661** | **0.570** | 0.732 | 0.605 | 0.772 | 0.623 | 0.439 | 0.452 |
| ETTh2 | 96 | 96 | **0.345** | **0.384** | 0.345 | 0.384 | 0.362 | 0.394 | 0.276 | 0.338 |
| | 96 | 192 | **0.435** | **0.436** | 0.435 | 0.436 | 0.460 | 0.448 | 0.345 | 0.382 |
| | 336 | 336 | **0.413** | **0.447** | 0.413 | 0.447 | 0.454 | 0.464 | 0.368 | 0.408 |
| | 512 | 720 | 0.410 | **0.453** | 0.407 | 0.449 | 0.454 | 0.454 | 0.406 | 0.441 |
| ETTm1 | 96 | 96 | **0.386** | **0.408** | 0.386 | 0.408 | 0.419 | 0.422 | 0.301 | 0.343 |
| | 96 | 192 | **0.484** | **0.449** | 0.501 | 0.457 | 0.547 | 0.480 | 0.355 | 0.379 |
| | 96 | 336 | **0.521** | **0.472** | 0.528 | 0.480 | 0.654 | 0.531 | 0.372 | 0.385 |
| | 96 | 720 | **0.553** | **0.497** | 0.639 | 0.536 | 0.715 | 0.578 | 0.430 | 0.418 |
| ETTm2 | 96 | 96 | **0.204** | **0.287** | 0.204 | 0.287 | 0.216 | 0.302 | 0.163 | 0.252 |
| | 96 | 192 | **0.271** | **0.326** | 0.273 | 0.327 | 0.320 | 0.365 | 0.218 | 0.290 |
| | 96 | 336 | **0.345** | **0.381** | 0.345 | 0.381 | 0.400 | 0.414 | 0.273 | 0.326 |
| | 96 | 720 | **0.439** | **0.434** | 0.441 | 0.434 | 0.512 | 0.468 | 0.361 | 0.382 |
| Exchange | 96 | 96 | 0.126 | 0.257 | 0.126 | 0.257 | 0.125 | 0.257 | 0.085 | 0.204 |
| | 96 | 192 | 0.222 | 0.341 | 0.206 | 0.329 | 0.208 | 0.335 | 0.175 | 0.297 |
| | 96 | 336 | 0.386 | 0.453 | 0.382 | 0.453 | 0.336 | 0.426 | 0.320 | 0.409 |
| | 96 | 720 | 0.974 | 0.752 | 1.018 | 0.768 | 0.663 | 0.631 | 0.838 | 0.690 |
| Weather | 96 | 96 | **0.189** | **0.235** | 0.189 | 0.235 | 0.210 | 0.256 | 0.180 | 0.226 |
| | 96 | 192 | **0.246** | **0.284** | 0.245 | 0.282 | 0.261 | 0.300 | 0.218 | 0.261 |
| | 96 | 336 | **0.296** | **0.316** | 0.301 | 0.321 | 0.309 | 0.332 | 0.266 | 0.296 |
| | 96 | 720 | **0.368** | **0.361** | 0.375 | 0.370 | 0.390 | 0.388 | 0.334 | 0.345 |
| Electricity | 96 | 96 | **0.178** | **0.283** | 0.178 | 0.283 | 0.215 | 0.321 | 0.140 | 0.236 |
| | 96 | 192 | **0.208** | **0.311** | 0.226 | 0.328 | 0.263 | 0.362 | 0.155 | 0.248 |
| | 96 | 336 | **0.232** | **0.331** | 0.267 | 0.363 | 0.334 | 0.416 | 0.171 | 0.264 |
| | 96 | 720 | **0.243** | **0.337** | 0.343 | 0.417 | 0.502 | 0.525 | 0.210 | 0.297 |
| ILI | 104 | 24 | **2.215** | **1.015** | 2.213 | 0.994 | 2.832 | 1.174 | 1.998 | 0.919 |
| | 104 | 36 | **2.429** | **1.024** | 2.481 | 1.036 | 2.889 | 1.147 | 1.920 | 0.916 |
| | 104 | 48 | **2.278** | **0.995** | 2.177 | 0.979 | 2.944 | 1.177 | 1.895 | 0.924 |
| | 36 | 60 | **2.340** | **0.981** | 2.340 | 0.979 | 2.902 | 1.158 | 1.964 | 0.947 |
| Solar | 96 | 96 | 0.356 | 0.387 | 0.356 | 0.387 | 0.329 | 0.368 | 0.202 | 0.245 |
| | 96 | 192 | 0.378 | 0.395 | 0.395 | 0.408 | 0.370 | 0.388 | 0.223 | 0.258 |
| | 96 | 336 | **0.401** | **0.410** | 0.405 | 0.412 | 0.419 | 0.420 | 0.238 | 0.265 |
| | 96 | 720 | 0.408 | **0.389** | 0.441 | 0.431 | 0.386 | 0.405 | 0.246 | 0.268 |
| Traffic | 96 | 96 | **0.609** | **0.327** | 0.609 | 0.327 | 0.682* | 0.388* | 0.395 | 0.272 |
| | 96 | 192 | 2.953 | 1.299 | 2.716 | 1.257 | 1.561* | 0.852* | 0.407 | 0.277 |
| | 96 | 336 | **0.648** | **0.346** | 0.688 | 0.384 | 0.862 | 0.477 | 0.417 | 0.282 |
| | 96 | 720 | **3.005** | **1.224** | 3.086 | 1.268 | 3.203 | 1.294 | 0.453 | 0.302 |
| PEMS-BAY | 96 | 96 | **0.807** | **0.435** | 0.807 | 0.435 | 0.818 | 0.439 | 0.642 | 0.402 |
| | 96 | 192 | **0.893** | **0.459** | 0.884 | 0.460 | 0.900 | 0.470 | 0.687 | 0.420 |
| | 96 | 336 | **0.831** | **0.443** | 0.826 | 0.447 | 0.844 | 0.454 | 0.735 | 0.437 |
| | 96 | 720 | **0.960** | **0.481** | 0.981 | 0.494 | 1.013 | 0.508 | 0.924 | 0.514 |
| METR-LA | 96 | 96 | **1.359** | **0.656** | 1.359 | 0.656 | 1.366 | 0.679 | 1.042 | 0.651 |
| | 96 | 192 | **1.543** | **0.701** | 1.525 | 0.700 | 1.558 | 0.712 | 1.218 | 0.720 |
| | 96 | 336 | 1.609 | **0.710** | 1.600 | 0.716 | 1.595 | 0.719 | 1.334 | 0.756 |
| | 96 | 720 | 1.851 | **0.792** | 1.846 | 0.800 | 1.833 | 0.806 | 1.683 | 0.886 |
| PEMS04 | 96 | 96 | 0.255 | 0.365 | 0.255 | 0.365 | 0.249 | 0.361 | 0.208 | 0.301 |
| | 96 | 192 | **0.287** | **0.388** | 0.296 | 0.396 | 0.288 | 0.390 | 0.229 | 0.312 |
| | 96 | 336 | **0.266** | **0.366** | 0.279 | 0.378 | 0.275 | 0.374 | 0.251 | 0.331 |
| | 96 | 720 | **0.348** | **0.427** | 0.396 | 0.454 | 0.376 | 0.451 | 0.346 | 0.398 |
| PEMS08 | 96 | 96 | 0.457 | 0.467 | 0.457 | 0.467 | 0.385 | 0.419 | 0.340 | 0.327 |
| | 96 | 192 | **0.571** | **0.487** | 0.591 | 0.503 | 0.589 | 0.501 | 0.450 | 0.353 |
| | 96 | 336 | 0.570 | 0.449 | 0.602 | 0.473 | 0.568 | 0.457 | 0.454 | 0.369 |
| | 96 | 720 | **0.699** | **0.511** | 0.775 | 0.559 | 0.739 | 0.540 | 0.494 | 0.429 |
| AQShunyi | 512 | 96 | **0.693** | **0.505** | 0.688 | 0.503 | 0.754 | 0.542 | 0.653 | 0.486 |
| | 512 | 192 | **0.719** | **0.517** | 0.717 | 0.516 | 0.759 | 0.536 | 0.701 | 0.506 |
| | 96 | 336 | 0.838 | 0.560 | 0.845 | 0.562 | 0.837 | 0.560 | 0.722 | 0.519 |
| | 512 | 720 | 0.805 | 0.554 | 0.800 | 0.554 | 0.777 | 0.543 | 0.777 | 0.545 |
| AQWan | 96 | 96 | **0.865** | **0.501** | 0.865 | 0.501 | 0.902 | 0.522 | 0.758 | 0.475 |
| | 512 | 192 | **0.805** | **0.499** | 0.815 | 0.503 | 0.833 | 0.521 | 0.809 | 0.496 |
| | 512 | 336 | **0.834** | **0.511** | 0.824 | 0.506 | 0.847 | 0.525 | 0.830 | 0.508 |
| | 512 | 720 | 0.909 | 0.543 | 0.909 | 0.543 | 0.883 | 0.532 | 0.906 | 0.538 |
| Wind | 96 | 96 | 0.975 | **0.665** | 0.975 | 0.665 | 0.954 | 0.663 | 0.923 | 0.640 |
| | 96 | 192 | 1.161 | **0.756** | 1.171 | 0.762 | 1.145 | 0.764 | 1.081 | 0.734 |
| | 96 | 336 | 1.321 | 0.833 | 1.343 | 0.844 | 1.313 | 0.826 | 1.228 | 0.805 |
| | 512 | 720 | 1.332 | 0.865 | 1.335 | 0.870 | 1.318 | 0.862 | 1.328 | 0.852 |
| ZafNoo | 96 | 96 | **0.531** | **0.452** | 0.531 | 0.452 | 0.547 | 0.477 | 0.447 | 0.410 |
| | 96 | 192 | **0.610** | **0.493** | 0.614 | 0.506 | 0.698 | 0.570 | 0.503 | 0.447 |
| | 96 | 336 | **0.623** | **0.505** | 0.677 | 0.544 | 0.832 | 0.655 | 0.545 | 0.470 |
| | 96 | 720 | **0.690** | **0.536** | 0.890 | 0.687 | 0.876 | 0.699 | 0.589 | 0.497 |
| CzeLan | 96 | 96 | 0.257 | 0.305 | 0.257 | 0.305 | 0.238 | 0.296 | 0.178 | 0.228 |
| | 96 | 192 | **0.284** | **0.327** | 0.280 | 0.325 | 0.299 | 0.340 | 0.210 | 0.252 |
| | 96 | 336 | **0.307** | **0.341** | 0.321 | 0.349 | 0.351 | 0.366 | 0.243 | 0.280 |
| | 336 | 720 | **0.357** | **0.378** | 0.382 | 0.404 | 0.384 | 0.416 | 0.290 | 0.326 |
| Covid-19 | 36 | 24 | **2.115** | **0.067** | 2.122 | 0.067 | 2.687 | 0.080 | 1.139 | 0.070 |
| | 36 | 36 | **2.465** | **0.075** | 2.465 | 0.075 | 3.071 | 0.087 | 1.582 | 0.091 |
| | 36 | 48 | **2.766** | **0.083** | 2.792 | 0.082 | 3.548 | 0.094 | 1.932 | 0.099 |
| | 36 | 60 | **3.258** | **0.091** | 3.361 | 0.091 | 4.052 | 0.104 | 2.682 | 0.127 |
| NASDAQ | 36 | 24 | **0.835** | **0.681** | 0.880 | 0.700 | 1.076 | 0.727 | 0.557 | 0.522 |
| | 104 | 36 | **1.234** | **0.865** | 1.241 | 0.865 | 1.411 | 0.897 | 0.869 | 0.668 |
| | 104 | 48 | **1.152** | **0.835** | 1.142 | 0.832 | 1.289 | 0.863 | 1.152 | 0.770 |
| | 104 | 60 | **1.072** | **0.810** | 1.072 | 0.810 | 1.158 | 0.829 | 1.284 | 0.809 |
| NYSE | 36 | 24 | 0.284 | 0.363 | 0.284 | 0.363 | 0.276 | 0.349 | 0.193 | 0.283 |
| | 36 | 36 | 0.408 | 0.435 | 0.408 | 0.435 | 0.407 | 0.417 | 0.315 | 0.356 |
| | 36 | 48 | **0.546** | **0.496** | 0.543 | 0.493 | 0.567 | 0.496 | 0.464 | 0.438 |
| | 36 | 60 | **0.796** | **0.602** | 0.706 | 0.570 | 0.899 | 0.670 | 0.631 | 0.522 |
| FRED-MD | 36 | 24 | **61.700** | **1.471** | 60.306 | 1.461 | 70.055 | 1.562 | 32.125 | 0.931 |
| | 36 | 36 | **82.488** | **1.706** | 82.488 | 1.706 | 101.858 | 1.860 | 58.332 | 1.260 |
| | 36 | 48 | **116.781** | **1.998** | 114.924 | 1.984 | 137.938 | 2.147 | 82.184 | 1.609 |
| | 36 | 60 | **155.983** | **2.297** | 153.512 | 2.277 | 184.740 | 2.460 | 109.625 | 1.882 |
| NN5 | 104 | 24 | **1.289** | **0.916** | 1.290 | 0.916 | 1.362 | 0.949 | 0.758 | 0.592 |
| | 104 | 36 | **1.270** | **0.913** | 1.265 | 0.910 | 1.341 | 0.944 | 0.693 | 0.577 |
| | 104 | 48 | **1.275** | **0.920** | 1.275 | 0.920 | 1.337 | 0.942 | 0.688 | 0.587 |
| | 104 | 60 | **1.281** | **0.925** | 1.284 | 0.926 | 1.335 | 0.945 | 0.679 | 0.587 |
| Wike2000 | 36 | 24 | 1196.199 | 1.578 | 1133.634 | 1.580 | 1190.077* | 1.566* | 1135.609 | 1.350 |
| | 36 | 36 | 1315.755 | **1.648** | 1315.755 | 1.648 | 1274.502 | 1.657 | 1060.939 | 1.388 |
| | 36 | 48 | **1328.209** | **1.700** | 1333.007 | 1.741 | 1372.456 | 1.718 | 1899.937 | 1.733 |
| | 36 | 60 | 1371.842 | **1.769** | 1485.935 | 1.821 | 1368.723 | 1.808 | 1281.245 | 1.570 |

Table 5: MSE and MAE scores of FiLM-based models. The lower, the better. DYN score is in **bold** when it is better than the vanilla version. PRO score is underlined when it is better than the DYN one. Vanilla score is starred* when we obtained better results than the TFB benchmark with updated learning hyperparameters.

| Dataset | Cont. L | Hor. H | FiLM DYN MSE | FiLM DYN MAE | FiLM PRO MSE | FiLM PRO MAE | FiLM MSE | FiLM MAE | NLinear MSE | NLinear MAE |
|---|---|---|---|---|---|---|---|---|---|---|
| ETTh1 | 512 | 96 | **0.367** | **0.392** | 0.366 | 0.392 | 0.370 | 0.394 | 0.385 | 0.403 |
| | 512 | 192 | **0.403** | 0.416 | 0.402 | 0.415 | 0.405 | 0.416 | 0.422 | 0.426 |
| | 512 | 336 | **0.429** | **0.433** | 0.428 | 0.433 | 0.434 | 0.435 | 0.431 | 0.429 |
| | 336 | 720 | **0.459** | **0.470** | 0.464 | 0.472 | 0.463 | 0.474 | 0.439 | 0.452 |
| ETTh2 | 512 | 96 | **0.275** | **0.339** | 0.269 | 0.335 | 0.275* | 0.340* | 0.276 | 0.338 |
| | 336 | 192 | **0.337** | **0.383** | 0.338 | 0.384 | 0.340* | 0.385* | 0.345 | 0.382 |
| | 512 | 336 | **0.367** | **0.415** | 0.373 | 0.417 | 0.372 | 0.425 | 0.368 | 0.408 |
| | 336 | 720 | **0.424** | **0.449** | 0.429 | 0.455 | 0.425 | 0.455 | 0.406 | 0.441 |
| ETTm1 | 512 | 96 | 0.306 | 0.348 | 0.305 | 0.347 | 0.301 | 0.343 | 0.301 | 0.343 |
| | 336 | 192 | **0.337** | **0.365** | 0.345 | 0.371 | 0.339 | 0.365 | 0.355 | 0.379 |
| | 336 | 336 | **0.371** | **0.383** | 0.371 | 0.383 | 0.374 | 0.385 | 0.372 | 0.385 |
| | 512 | 720 | **0.422** | 0.414 | 0.433 | 0.422 | 0.423 | 0.414 | 0.430 | 0.418 |
| ETTm2 | 512 | 96 | **0.163** | **0.252** | 0.163 | 0.252 | 0.165 | 0.254 | 0.163 | 0.252 |
| | 512 | 192 | 0.220 | **0.290** | 0.218 | 0.290 | 0.220 | 0.291 | 0.218 | 0.290 |
| | 336 | 336 | **0.274** | **0.326** | 0.274 | 0.326 | 0.277 | 0.329 | 0.273 | 0.326 |
| | 512 | 720 | **0.361** | **0.385** | 0.362 | 0.385 | 0.363 | 0.386 | 0.361 | 0.382 |
| Exchange | 512 | 96 | 0.088 | 0.207 | 0.087 | 0.206 | 0.087 | 0.210 | 0.085 | 0.204 |
| | 96 | 192 | **0.173** | **0.294** | 0.172 | 0.294 | 0.182 | 0.308 | 0.175 | 0.297 |
| | 96 | 336 | 0.318 | 0.407 | 0.315 | 0.405 | 0.318 | 0.409 | 0.320 | 0.409 |
| | 96 | 720 | 0.821 | 0.682 | 0.800 | 0.672 | 0.815 | 0.681 | 0.838 | 0.690 |
| Weather | 336 | 96 | **0.174** | **0.223** | 0.177 | 0.226 | 0.178 | 0.229 | 0.180 | 0.226 |
| | 512 | 192 | **0.213** | **0.258** | 0.214 | 0.259 | 0.218 | 0.263 | 0.218 | 0.261 |
| | 336 | 336 | **0.264** | **0.293** | 0.264 | 0.293 | 0.266 | 0.295 | 0.266 | 0.296 |
| | 336 | 720 | **0.331** | **0.339** | 0.331 | 0.339 | 0.332 | 0.341 | 0.334 | 0.345 |
| Electricity | 192 | 96 | **0.153** | 0.246 | 0.153 | 0.246 | 0.154 | 0.246 | 0.140 | 0.236 |
| | 192 | 192 | 0.172 | 0.268 | 0.172 | 0.268 | 0.168 | 0.261 | 0.155 | 0.248 |
| | 192 | 336 | **0.188** | **0.283** | 0.188 | 0.283 | 0.189 | 0.284 | 0.171 | 0.264 |
| | 192 | 720 | **0.248** | **0.339** | 0.249 | 0.340 | 0.249 | 0.340 | 0.210 | 0.297 |
| ILI | 104 | 24 | 2.337 | 1.026 | 2.143 | 0.966 | 2.256 | 0.996 | 1.998 | 0.919 |
| | 104 | 36 | 2.290 | 1.038 | 2.067 | 0.972 | 2.133 | 0.992 | 1.920 | 0.916 |
| | 104 | 48 | 2.511 | 1.070 | 2.193 | 1.011 | 2.034 | 0.969 | 1.895 | 0.924 |
| | 104 | 60 | 1.981 | 0.920 | 2.021 | 0.936 | 1.974 | 0.929 | 1.964 | 0.947 |
| Solar | 512 | 96 | **0.202** | **0.245** | 0.202 | 0.244 | 0.214 | 0.259 | 0.202 | 0.245 |
| | 512 | 192 | **0.224** | **0.256** | 0.224 | 0.256 | 0.226 | 0.257 | 0.223 | 0.258 |
| | 512 | 336 | **0.238** | **0.264** | 0.238 | 0.263 | 0.241 | 0.265 | 0.238 | 0.265 |
| | 512 | 720 | **0.245** | **0.267** | 0.245 | 0.267 | 0.247 | 0.268 | 0.246 | 0.268 |
| Traffic | 512 | 96 | **0.397** | **0.276** | 0.396 | 0.274 | 0.412 | 0.284 | 0.395 | 0.272 |
| | 512 | 192 | **0.413** | 0.286 | 0.410 | 0.282 | 0.415 | 0.285 | 0.407 | 0.277 |
| | 512 | 336 | **0.428** | 0.299 | 0.426 | 0.297 | 0.430 | 0.299 | 0.417 | 0.282 |
| | 512 | 720 | **0.517** | **0.364** | 0.517 | 0.364 | 0.525 | 0.371 | 0.453 | 0.302 |
| PEMS-BAY | 512 | 96 | **0.637** | **0.392** | 0.636 | 0.391 | 0.658 | 0.421 | 0.642 | 0.402 |
| | 512 | 192 | **0.677** | 0.401 | 0.676 | 0.401 | 0.680 | 0.401 | 0.687 | 0.420 |
| | 512 | 336 | 0.725 | 0.417 | 0.725 | 0.417 | 0.728 | 0.418 | 0.735 | 0.437 |
| | 720 | 720 | **0.894** | 0.483 | 0.894 | 0.483 | 0.895 | 0.482 | 0.924 | 0.514 |
| METR-LA | 96 | 96 | 1.276 | **0.681** | 1.276 | 0.681 | 1.287 | 0.697 | 1.042 | 0.651 |
| | 720 | 192 | **1.196** | **0.710** | 1.199 | 0.718 | 1.223 | 0.733 | 1.218 | 0.720 |
| | 512 | 336 | **1.327** | 0.742 | 1.322 | 0.736 | 1.334 | 0.741 | 1.334 | 0.756 |
| | 512 | 720 | **1.590** | **0.818** | 1.592 | 0.819 | 1.602 | 0.822 | 1.683 | 0.886 |
| PEMS04 | 720 | 96 | **0.203** | **0.295** | 0.202 | 0.294 | 0.208 | 0.299 | 0.208 | 0.301 |
| | 720 | 192 | **0.218** | **0.306** | 0.218 | 0.305 | 0.221 | 0.308 | 0.229 | 0.312 |
| | 720 | 336 | **0.239** | **0.322** | 0.238 | 0.322 | 0.246 | 0.327 | 0.251 | 0.331 |
| | 720 | 720 | **0.317** | **0.383** | 0.317 | 0.383 | 0.329 | 0.391 | 0.346 | 0.398 |
| PEMS08 | 512 | 96 | **0.343** | **0.326** | 0.343 | 0.330 | 0.349 | 0.330 | 0.340 | 0.327 |
| | 512 | 192 | **0.418** | 0.352 | 0.416 | 0.351 | 0.428 | 0.352 | 0.450 | 0.353 |
| | 512 | 336 | **0.453** | **0.370** | 0.454 | 0.373 | 0.463 | 0.374 | 0.454 | 0.369 |
| | 512 | 720 | **0.493** | **0.429** | 0.493 | 0.430 | 0.498 | 0.434 | 0.494 | 0.429 |
| AQShunyi | 336 | 96 | **0.659** | **0.483** | 0.659 | 0.483 | 0.664 | 0.486 | 0.653 | 0.486 |
| | 336 | 192 | **0.700** | **0.501** | 0.700 | 0.501 | 0.705 | 0.504 | 0.701 | 0.506 |
| | 336 | 336 | **0.720** | **0.514** | 0.720 | 0.514 | 0.725 | 0.517 | 0.722 | 0.519 |
| | 336 | 720 | **0.778** | **0.542** | 0.779 | 0.543 | 0.782 | 0.544 | 0.777 | 0.545 |
| AQWan | 336 | 96 | **0.758** | **0.471** | 0.760 | 0.474 | 0.766 | 0.475 | 0.758 | 0.475 |
| | 336 | 192 | **0.804** | **0.490** | 0.804 | 0.490 | 0.809 | 0.494 | 0.809 | 0.496 |
| | 336 | 336 | **0.824** | **0.502** | 0.824 | 0.502 | 0.831 | 0.505 | 0.830 | 0.508 |
| | 336 | 720 | **0.903** | **0.534** | 0.902 | 0.533 | 0.906 | 0.536 | 0.906 | 0.538 |
| Wind | 336 | 96 | **0.926** | **0.646** | 0.927 | 0.646 | 0.933 | 0.649 | 0.923 | 0.640 |
| | 512 | 192 | **1.087** | **0.737** | 1.087 | 0.739 | 1.097 | 0.743 | 1.081 | 0.734 |
| | 512 | 336 | **1.226** | **0.801** | 1.230 | 0.803 | 1.241 | 0.809 | 1.228 | 0.805 |
| | 512 | 720 | **1.324** | **0.847** | 1.325 | 0.847 | 1.334 | 0.852 | 1.328 | 0.852 |
| ZafNoo | 336 | 96 | **0.447** | 0.409 | 0.447 | 0.408 | 0.451 | 0.411 | 0.447 | 0.410 |
| | 512 | 192 | **0.503** | **0.445** | 0.503 | 0.444 | 0.508 | 0.448 | 0.503 | 0.447 |
| | 512 | 336 | **0.544** | **0.469** | 0.545 | 0.469 | 0.549 | 0.471 | 0.545 | 0.470 |
| | 512 | 720 | **0.598** | **0.503** | 0.593 | 0.500 | 0.598 | 0.504 | 0.589 | 0.497 |
| CzeLan | 336 | 96 | **0.178** | **0.230** | 0.178 | 0.230 | 0.180 | 0.232 | 0.178 | 0.228 |
| | 336 | 192 | **0.210** | **0.253** | 0.209 | 0.253 | 0.212 | 0.255 | 0.210 | 0.252 |
| | 336 | 336 | **0.242** | **0.279** | 0.242 | 0.279 | 0.243 | 0.281 | 0.243 | 0.280 |
| | 512 | 720 | **0.281** | **0.311** | 0.281 | 0.310 | 0.282 | 0.312 | 0.290 | 0.326 |
| Covid-19 | 36 | 24 | **1.055** | **0.045** | 1.058 | 0.045 | 1.183 | 0.047 | 1.139 | 0.070 |
| | 36 | 36 | **1.371** | **0.054** | 1.371 | 0.054 | 1.470 | 0.059 | 1.582 | 0.091 |
| | 36 | 48 | **1.739** | **0.063** | 1.754 | 0.064 | 1.859 | 0.069 | 1.932 | 0.099 |
| | 36 | 60 | **2.142** | **0.073** | 2.187 | 0.074 | 2.278 | 0.078 | 2.682 | 0.127 |
| NASDAQ | 36 | 24 | **0.642** | **0.567** | 0.639 | 0.563 | 0.767 | 0.563 | 0.557 | 0.522 |
| | 36 | 36 | **1.000** | **0.707** | 1.000 | 0.707 | 1.271* | 0.800* | 0.869 | 0.668 |
| | 104 | 48 | **1.172** | **0.773** | 1.311 | 0.815 | 1.179 | 0.829 | 1.152 | 0.770 |
| | 104 | 60 | **1.342** | **0.825** | 1.380 | 0.835 | 1.303 | 0.853 | 1.284 | 0.809 |
| NYSE | 36 | 24 | **0.195** | **0.283** | 0.192 | 0.282 | 0.276* | 0.339* | 0.193 | 0.283 |
| | 36 | 36 | **0.314** | **0.358** | 0.314 | 0.358 | 0.376* | 0.406* | 0.315 | 0.356 |
| | 36 | 48 | **0.452** | **0.427** | 0.451 | 0.428 | 0.523* | 0.474* | 0.464 | 0.438 |
| | 36 | 60 | **0.613** | **0.517** | 0.611 | 0.514 | 0.698* | 0.554* | 0.631 | 0.522 |
| FRED-MD | 36 | 24 | 31.809 | 0.960 | 31.050 | 0.948 | 37.426* | 1.092* | 32.125 | 0.931 |
| | 36 | 36 | **60.333** | **1.331** | 60.333 | 1.311 | 90.434 | 1.670 | 58.332 | 1.258 |
| | 104 | 48 | **99.858** | **1.709** | 105.524 | 1.719 | 131.081 | 2.119 | 82.184 | 1.609 |
| | 36 | 60 | **126.751** | **1.941** | 130.846 | 1.974 | 180.367 | 2.397 | 109.625 | 1.882 |
| NN5 | 104 | 24 | 0.821 | 0.637 | 0.785 | 0.613 | 0.846 | 0.651 | 0.758 | 0.592 |
| | 104 | 36 | **0.812** | **0.652** | 0.716 | 0.593 | 0.883 | 0.702 | 0.693 | 0.577 |
| | 36 | 48 | **0.822** | **0.657** | 0.805 | 0.649 | 0.969 | 0.741 | 0.688 | 0.587 |
| | 104 | 60 | 0.657 | 0.572 | 0.650 | 0.567 | 0.633 | 0.556 | 0.679 | 0.587 |
| Wike2000 | 36 | 24 | **631.760** | **1.166** | 623.585 | 1.162 | 959.454 | 1.318 | 1135.609 | 1.350 |
| | 36 | 36 | **667.587** | **1.245** | 667.587 | 1.245 | 983.239 | 1.419 | 1060.939 | 1.388 |
| | 36 | 48 | **724.318** | **1.334** | 708.970 | 1.320 | 1358.260 | 1.581 | 1899.937 | 1.733 |
| | 36 | 60 | **743.718** | **1.389** | 750.659 | 1.387 | 1157.091 | 1.582 | 1281.245 | 1.570 |

Table 6: MSE and MAE scores of MICN-based models. The lower, the better. DYN score is in **bold** when it is better than the vanilla version. Vanilla score is starred* when we obtained better results than the TFB benchmark with updated learning hyperparameters.

| Dataset | Cont. $L$ | Hor. $H$ | MICN PRO MSE | MICN PRO MAE | MICN MSE | MICN MAE | NLinear MSE | NLinear MAE |
|---|---|---|---|---|---|---|---|---|
| ETTh1 | 96 | 96 | **0.375** | **0.410** | 0.377* | 0.412 | 0.385 | 0.403 |
| | 512 | 192 | **0.398** | **0.428** | 0.400 | 0.430 | 0.422 | 0.426 |
| | 336 | 336 | 0.425 | 0.446 | 0.425* | 0.445* | 0.431 | 0.429 |
| | 512 | 720 | 0.479 | 0.501 | 0.474 | 0.499 | 0.439 | 0.452 |
| ETTh2 | 336 | 96 | 0.312 | **0.370** | 0.312* | 0.371* | 0.276 | 0.338 |
| | 336 | 192 | 0.393 | 0.423 | 0.393* | 0.423* | 0.345 | 0.382 |
| | 336 | 336 | 0.473 | 0.474 | 0.472* | 0.474* | 0.368 | 0.408 |
| | 336 | 720 | **0.703** | **0.591** | 0.723 | 0.600 | 0.406 | 0.441 |
| ETTm1 | 336 | 96 | **0.301** | **0.348** | 0.302* | 0.349 | 0.301 | 0.343 |
| | 336 | 192 | 0.331 | **0.368** | 0.331* | 0.369 | 0.355 | 0.379 |
| | 512 | 336 | 0.368 | **0.388** | 0.368* | 0.389* | 0.372 | 0.385 |
| | 512 | 720 | 0.416 | **0.420** | 0.410 | 0.421 | 0.430 | 0.418 |
| ETTm2 | 512 | 96 | 0.172 | 0.269 | 0.172* | 0.269* | 0.163 | 0.252 |
| | 512 | 192 | 0.223 | 0.299 | 0.223* | 0.299* | 0.218 | 0.290 |
| | 512 | 336 | **0.297** | **0.360** | 0.303 | 0.366* | 0.273 | 0.326 |
| | 512 | 720 | **0.438** | **0.446** | 0.461* | 0.477* | 0.361 | 0.382 |
| Exchange | 96 | 96 | 0.079 | 0.203 | 0.079 | 0.203 | 0.085 | 0.204 |
| | 96 | 192 | 0.159 | 0.299 | 0.158 | 0.299 | 0.175 | 0.297 |
| | 96 | 336 | 0.296 | 0.418 | 0.296 | 0.418 | 0.320 | 0.409 |
| | 96 | 720 | 0.763 | 0.677 | 0.745 | 0.675 | 0.838 | 0.690 |
| Weather | 512 | 96 | 0.172 | 0.231 | 0.172 | 0.231* | 0.180 | 0.226 |
| | 512 | 192 | 0.214 | **0.270** | 0.214 | 0.270* | 0.218 | 0.261 |
| | 512 | 336 | **0.258** | **0.307** | 0.259 | 0.309 | 0.266 | 0.296 |
| | 96 | 720 | 0.313 | 0.358 | 0.309 | 0.343 | 0.334 | 0.345 |
| Electricity | 96 | 96 | 0.160 | 0.270 | 0.158 | 0.266 | 0.140 | 0.236 |
| | 96 | 192 | 0.174 | 0.285 | 0.173* | 0.284* | 0.155 | 0.248 |
| | 96 | 336 | 0.183 | 0.295 | 0.183* | 0.294* | 0.171 | 0.264 |
| | 96 | 720 | 0.203 | 0.314 | 0.200 | 0.310 | 0.210 | 0.297 |
| ILI | 104 | 24 | **2.256** | **1.013** | 2.279 | 1.020 | 1.998 | 0.919 |
| | 104 | 36 | 2.491 | 1.095 | 2.451 | 1.085 | 1.920 | 0.916 |
| | 104 | 48 | 2.469 | 1.085 | 2.440 | 1.077 | 1.895 | 0.924 |
| | 104 | 60 | **2.273** | **1.006** | 2.303 | 1.012 | 1.964 | 0.947 |
| Solar | 96 | 96 | **0.180** | 0.252 | 0.190 | 0.250 | 0.202 | 0.245 |
| | 96 | 192 | **0.207** | **0.246** | 0.226 | 0.284 | 0.223 | 0.258 |
| | 96 | 336 | **0.224** | **0.279** | 0.259 | 0.308 | 0.238 | 0.265 |
| | 96 | 720 | **0.276** | **0.319** | 0.341 | 0.365 | 0.246 | 0.268 |
| Traffic | 96 | 96 | 0.523 | **0.305** | 0.508* | 0.299* | 0.395 | 0.272 |
| | 96 | 192 | 0.537 | 0.316 | 0.518* | 0.298* | 0.407 | 0.277 |
| | 96 | 336 | 0.550 | 0.322 | 0.545 | 0.307 | 0.417 | 0.282 |
| | 96 | 720 | 0.594 | 0.333 | 0.569 | 0.328 | 0.453 | 0.302 |
| PEMS-BAY | 96 | 96 | **0.637** | 0.419 | 0.647 | 0.393 | 0.642 | 0.402 |
| | 96 | 192 | **0.760** | **0.443** | 0.767 | 0.444 | 0.687 | 0.420 |
| | 96 | 336 | **0.835** | 0.523 | 0.903 | 0.468 | 0.735 | 0.437 |
| | 96 | 720 | **1.046** | **0.613** | 1.119 | 0.634 | 0.924 | 0.514 |
| METR-LA | 96 | 96 | **1.161** | **0.724** | 1.195 | 0.729 | 1.042 | 0.651 |
| | 96 | 192 | 1.359 | 0.735 | 1.309 | 0.729 | 1.218 | 0.720 |
| | 96 | 336 | 1.342 | 0.778 | 1.332 | 0.773 | 1.334 | 0.756 |
| | 96 | 720 | 1.502 | 0.805 | 1.501 | 0.793 | 1.683 | 0.886 |
| PEMS04 | 96 | 96 | 0.257 | 0.362 | 0.241 | 0.350 | 0.208 | 0.301 |
| | 96 | 192 | **0.290** | **0.390** | 0.358 | 0.444 | 0.229 | 0.312 |
| | 96 | 336 | **0.310** | **0.413** | 0.383 | 0.460 | 0.251 | 0.331 |
| | 96 | 720 | 0.749 | 0.679 | 0.675 | 0.633 | 0.346 | 0.398 |
| PEMS08 | 96 | 96 | 0.477 | 0.450 | 0.391 | 0.412 | 0.340 | 0.327 |
| | 96 | 192 | **0.462** | **0.457** | 0.517 | 0.507 | 0.450 | 0.353 |
| | 96 | 336 | **0.472** | **0.445** | 0.494 | 0.479 | 0.454 | 0.369 |
| | 96 | 720 | 0.792 | **0.631** | 0.776 | 0.647 | 0.494 | 0.429 |
| AQShunyi | 96 | 96 | 0.693 | **0.511** | 0.688* | 0.512 | 0.653 | 0.486 |
| | 96 | 192 | 0.716 | 0.528 | 0.710* | 0.528* | 0.701 | 0.506 |
| | 96 | 336 | **0.724** | **0.539** | 0.729 | 0.544 | 0.722 | 0.519 |
| | 336 | 720 | **0.724** | **0.530** | 0.732* | 0.531* | 0.777 | 0.545 |
| AQWan | 336 | 96 | 0.773 | **0.492** | 0.773* | 0.493* | 0.758 | 0.475 |
| | 96 | 192 | 0.819 | 0.518 | 0.819 | 0.518 | 0.809 | 0.496 |
| | 96 | 336 | **0.823** | 0.522 | 0.826 | 0.522* | 0.830 | 0.508 |
| | 96 | 720 | **0.872** | **0.542** | 0.875 | 0.544 | 0.906 | 0.538 |
| Wind | 96 | 96 | **0.864** | 0.623 | 0.865* | 0.623* | 0.923 | 0.640 |
| | 96 | 192 | **1.024** | **0.709** | 1.066 | 0.719 | 1.081 | 0.734 |
| | 512 | 336 | **1.152** | **0.778** | 1.160* | 0.781* | 1.228 | 0.805 |
| | 512 | 720 | 1.247 | 0.820 | 1.245 | 0.820 | 1.328 | 0.852 |
| ZafNoo | 96 | 96 | **0.437** | **0.418** | 0.442 | 0.421 | 0.447 | 0.410 |
| | 336 | 192 | 0.507 | 0.464 | 0.493 | 0.455 | 0.503 | 0.447 |
| | 96 | 336 | **0.497** | **0.451** | 0.504 | 0.455 | 0.545 | 0.470 |
| | 96 | 720 | **0.532** | **0.470** | 0.540 | 0.476 | 0.589 | 0.497 |
| CzeLan | 512 | 96 | 0.194 | 0.275 | 0.194 | 0.275 | 0.178 | 0.228 |
| | 336 | 192 | **0.254** | **0.322** | 0.275 | 0.362 | 0.210 | 0.252 |
| | 512 | 336 | 0.310 | **0.356** | 0.309 | 0.359 | 0.243 | 0.280 |
| | 512 | 720 | **0.339** | **0.387** | 0.358 | 0.395 | 0.290 | 0.326 |
| Covid-19 | 36 | 24 | **35.655** | **0.448** | 38.500 | 0.469 | 1.139 | 0.070 |
| | 104 | 36 | 47.277 | 0.519 | 45.425 | 0.489 | 1.582 | 0.091 |
| | 104 | 48 | **30.860** | **0.434** | 46.565 | 0.566 | 1.932 | 0.099 |
| | 104 | 60 | 80.919 | 0.645 | 77.972 | 0.616 | 2.682 | 0.127 |
| NASDAQ | 36 | 24 | **0.743** | **0.618** | 0.769* | 0.624* | 0.557 | 0.522 |
| | 36 | 36 | **1.277** | **0.816** | 1.319* | 0.834* | 0.869 | 0.668 |
| | 36 | 48 | **1.693** | **0.940** | 1.788 | 0.977 | 1.152 | 0.770 |
| | 36 | 60 | **2.262** | **1.117** | 2.280* | 1.124* | 1.284 | 0.809 |
| NYSE | 36 | 24 | 0.411 | 0.461 | 0.408* | 0.458* | 0.193 | 0.283 |
| | 36 | 36 | **0.640** | **0.600** | 0.733 | 0.635 | 0.315 | 0.356 |
| | 104 | 48 | 0.843 | 0.695 | 0.768 | 0.663 | 0.464 | 0.438 |
| | 36 | 60 | 1.218 | 0.854 | 1.204 | 0.843 | 0.631 | 0.522 |
| FRED-MD | 36 | 24 | **63.213** | **1.518** | 63.217 | 1.521 | 32.125 | 0.931 |
| | 36 | 36 | 102.826 | 1.947 | 102.800 | 1.945 | 58.332 | 1.258 |
| | 36 | 48 | **147.335** | **2.360** | 147.405 | 2.362 | 82.184 | 1.609 |
| | 36 | 60 | **202.442** | **2.784** | 202.988 | 2.800 | 109.625 | 1.882 |
| NN5 | 36 | 24 | 0.736 | 0.593 | 0.728 | 0.589 | 0.758 | 0.592 |
| | 104 | 36 | **0.656** | **0.561** | 0.658 | 0.562 | 0.693 | 0.577 |
| | 104 | 48 | **0.626** | **0.550** | 0.646 | 0.562 | 0.688 | 0.587 |
| | 104 | 60 | **0.623** | **0.552** | 0.645 | 0.564 | 0.679 | 0.587 |
| Wike2000 | 36 | 24 | 507.237 | 1.369 | 506.895 | 1.365 | 1135.609 | 1.350 |
| | 36 | 36 | **577.394** | 1.596 | 577.706 | 1.596 | 1060.939 | 1.388 |
| | 36 | 48 | **612.004** | **1.684** | 612.170 | 1.688 | 1899.937 | 1.733 |
| | 36 | 60 | **654.285** | 1.819 | 654.767 | 1.804 | 1281.245 | 1.570 |

Table 7: MSE and MAE scores of FEDformer-based models. The lower, the better. DYN score is in **bold** when it is better than the vanilla version. PRO score is underlined when it is better than the DYN one. Vanilla score is starred* when we obtained better results than the TFB benchmark with updated learning hyperparameters.

| Dataset | Cont. L | Hor. H | FEDformer DYN MSE | MAE | FEDformer PRO MSE | MAE | FEDformer MSE | MAE | NLinear MSE | MAE |
|---|---|---|---|---|---|---|---|---|---|---|
| ETTh1 | 96 | 96 | **0.378** | 0.419 | 0.389 | 0.429 | 0.379 | 0.419 | 0.385 | 0.403 |
| | 96 | 192 | **0.419** | **0.442** | 0.420 | 0.443 | 0.420 | 0.444 | 0.422 | 0.426 |
| | 96 | 336 | **0.456** | **0.464** | 0.457 | 0.463 | 0.458 | 0.466 | 0.431 | 0.429 |
| | 96 | 720 | 0.479 | 0.491 | 0.495 | 0.497 | 0.474 | 0.488 | 0.439 | 0.452 |
| ETTh2 | 96 | 96 | 0.338 | 0.380 | 0.337 | 0.380 | 0.337 | 0.380 | 0.276 | 0.338 |
| | 96 | 192 | **0.414** | **0.427** | 0.416 | 0.429 | 0.415 | 0.428 | 0.345 | 0.382 |
| | 512 | 336 | **0.372** | **0.444** | 0.376 | 0.449 | 0.389 | 0.457 | 0.368 | 0.408 |
| | 96 | 720 | **0.475** | **0.483** | 0.481 | 0.486 | 0.483 | 0.488 | 0.406 | 0.441 |
| ETTm1 | 96 | 96 | 0.464 | 0.465 | 0.464 | 0.465 | 0.463 | 0.463 | 0.301 | 0.343 |
| | 96 | 192 | **0.571** | 0.522 | 0.571 | 0.510 | 0.575 | 0.516 | 0.355 | 0.379 |
| | 512 | 336 | 0.632 | 0.554 | 0.746 | 0.603 | 0.618 | 0.544 | 0.372 | 0.385 |
| | 336 | 720 | 0.615 | 0.554 | 0.616 | 0.551 | 0.612 | 0.551 | 0.430 | 0.418 |
| ETTm2 | 96 | 96 | 0.218 | 0.319 | 0.280 | 0.370 | 0.216 | 0.309 | 0.163 | 0.252 |
| | 96 | 192 | 0.318 | 0.377 | 0.321 | 0.382 | 0.297 | 0.360 | 0.218 | 0.290 |
| | 96 | 336 | **0.357** | **0.393** | 0.364 | 0.399 | 0.366 | 0.400 | 0.273 | 0.326 |
| | 96 | 720 | **0.455** | **0.447** | 0.458 | 0.449 | 0.459 | 0.450 | 0.361 | 0.382 |
| Exchange | 96 | 96 | 0.151 | 0.288 | 0.152 | 0.281 | 0.138 | 0.268 | 0.085 | 0.204 |
| | 96 | 192 | **0.269** | **0.376** | 0.234 | 0.355 | 0.273 | 0.379 | 0.175 | 0.297 |
| | 96 | 336 | 0.437 | **0.484** | 0.441 | 0.487 | 0.437 | 0.485 | 0.320 | 0.409 |
| | 96 | 720 | 1.162 | 0.830 | 1.139 | 0.826 | 1.158 | 0.828 | 0.838 | 0.690 |
| Weather | 96 | 96 | 0.267 | 0.339 | 0.274 | 0.342 | 0.229 | 0.298 | 0.180 | 0.226 |
| | 96 | 192 | 0.261 | **0.319** | 0.281 | 0.345 | 0.258* | 0.323* | 0.218 | 0.261 |
| | 336 | 336 | 0.334 | 0.372 | 0.357 | 0.403 | 0.329* | 0.371* | 0.266 | 0.296 |
| | 96 | 720 | **0.418** | **0.411** | 0.420 | 0.414 | 0.423 | 0.418 | 0.334 | 0.345 |
| Electricity | 96 | 96 | 0.192 | 0.307 | 0.191 | 0.306 | 0.191 | 0.305 | 0.140 | 0.236 |
| | 96 | 192 | **0.201** | **0.314** | 0.202 | 0.315 | 0.202* | 0.315* | 0.155 | 0.248 |
| | 96 | 336 | **0.220** | **0.332** | 0.221 | 0.334 | 0.221 | 0.333 | 0.171 | 0.264 |
| | 96 | 720 | **0.254** | **0.360** | 0.256 | 0.362 | 0.256* | 0.362* | 0.210 | 0.297 |
| ILI | 36 | 24 | **2.389** | 1.020 | 2.362 | 1.012 | 2.398 | 1.020 | 1.998 | 0.919 |
| | 36 | 36 | 2.627 | 1.056 | 2.479 | 1.016 | 2.410 | 1.005 | 1.920 | 0.916 |
| | 36 | 48 | 2.673 | 1.059 | 2.771 | 1.079 | 2.591 | 1.033 | 1.895 | 0.924 |
| | 36 | 60 | 2.541 | 1.079 | 2.512 | 1.078 | 2.539 | 1.070 | 1.964 | 0.947 |
| Solar | 96 | 96 | 0.462 | 0.560 | 0.471 | 0.557 | 0.462* | 0.547* | 0.202 | 0.245 |
| | 96 | 192 | **0.403** | 0.514 | 0.379 | 0.469 | 0.408* | 0.477* | 0.223 | 0.258 |
| | 96 | 336 | 1.009 | 0.857 | 0.950 | 0.799 | 1.008 | 0.839 | 0.238 | 0.265 |
| | 96 | 720 | **0.329** | **0.423** | 0.327 | 0.421 | 0.343* | 0.443* | 0.246 | 0.268 |
| Traffic | 96 | 96 | 0.594 | 0.366 | 0.593 | 0.366 | 0.593 | 0.365 | 0.395 | 0.272 |
| | 96 | 192 | 0.612 | **0.379** | 0.611 | 0.379 | 0.614 | 0.381 | 0.407 | 0.277 |
| | 96 | 336 | **0.608** | **0.371** | 0.618 | 0.378 | 0.613* | 0.376* | 0.417 | 0.282 |
| | 96 | 720 | 0.652 | 0.397 | 0.656 | 0.400 | 0.646 | 0.394 | 0.453 | 0.302 |
| PEMS-BAY | 96 | 96 | **0.776** | **0.529** | 0.980 | 0.625 | 0.854 | 0.586 | 0.642 | 0.402 |
| | 96 | 192 | **0.909** | **0.583** | 0.884 | 0.572 | 1.217 | 0.714 | 0.687 | 0.420 |
| | 96 | 336 | 1.173 | **0.677** | 1.175 | 0.695 | 1.184 | 0.689 | 0.735 | 0.437 |
| | 96 | 720 | **1.061** | **0.623** | 1.021 | 0.632 | 1.082 | 0.642 | 0.924 | 0.514 |
| METR-LA | 96 | 96 | **1.308** | **0.787** | 1.285 | 0.792 | 1.464 | 0.813 | 1.042 | 0.651 |
| | 96 | 192 | **1.573** | **0.884** | 1.550 | 0.865 | 2.042 | 0.999 | 1.218 | 0.720 |
| | 96 | 336 | **1.516** | **0.877** | 1.567 | 0.921 | 1.672 | 0.889 | 1.334 | 0.756 |
| | 96 | 720 | **1.676** | **0.883** | 1.778 | 0.902 | 2.015 | 0.963 | 1.683 | 0.886 |
| PEMS04 | 96 | 96 | **0.539** | **0.561** | 0.556 | 0.557 | 0.598 | 0.571 | 0.208 | 0.301 |
| | 96 | 192 | **0.753** | **0.696** | 0.798 | 0.714 | 0.996 | 0.804 | 0.229 | 0.312 |
| | 96 | 336 | 1.826 | 1.073 | 1.869 | 1.081 | 1.817 | 1.067 | 0.251 | 0.331 |
| | 96 | 720 | **0.501** | **0.552** | 0.588 | 0.598 | 1.047 | 0.795 | 0.346 | 0.398 |
| PEMS08 | 96 | 96 | 0.738 | 0.609 | 0.664 | 0.570 | 0.722 | 0.585 | 0.340 | 0.327 |
| | 96 | 192 | **0.937** | **0.705** | 0.861 | 0.670 | 1.159 | 0.810 | 0.450 | 0.353 |
| | 96 | 336 | **2.109** | **1.114** | 2.225 | 1.145 | 2.222 | 1.142 | 0.454 | 0.369 |
| | 96 | 720 | **0.844** | **0.645** | 0.914 | 0.682 | 1.243 | 0.825 | 0.494 | 0.429 |
| AQShunyi | 512 | 96 | 0.694 | **0.517** | 0.694 | 0.516 | 0.706 | 0.525 | 0.653 | 0.486 |
| | 336 | 192 | **0.728** | 0.531 | 0.731 | 0.535 | 0.729 | 0.531 | 0.701 | 0.506 |
| | 96 | 336 | 0.825 | 0.571 | 0.827 | 0.569 | 0.824 | 0.569 | 0.722 | 0.519 |
| | 512 | 720 | **0.791** | 0.559 | 0.793 | 0.561 | 0.794 | 0.561 | 0.777 | 0.545 |
| AQWan | 336 | 96 | **0.790** | **0.507** | 0.796 | 0.507 | 0.796 | 0.508 | 0.758 | 0.475 |
| | 336 | 192 | **0.824** | 0.517 | 0.824 | 0.517 | 0.825 | 0.517 | 0.809 | 0.496 |
| | 336 | 336 | 0.863 | 0.537 | 0.863 | 0.537 | 0.863 | 0.537 | 0.830 | 0.508 |
| | 512 | 720 | **0.905** | **0.551** | 0.907 | 0.552 | 0.907 | 0.552 | 0.906 | 0.538 |
| Wind | 96 | 96 | **0.967** | **0.688** | 1.024 | 0.706 | 1.034 | 0.699 | 0.923 | 0.640 |
| | 96 | 192 | **1.200** | **0.795** | 1.198 | 0.792 | 1.280 | 0.805 | 1.081 | 0.734 |
| | 336 | 336 | **1.284** | **0.850** | 1.318 | 0.868 | 1.302 | 0.864 | 1.228 | 0.805 |
| | 336 | 720 | **1.361** | **0.881** | 1.327 | 0.871 | 1.373 | 0.881 | 1.328 | 0.852 |
| ZafNoo | 96 | 96 | **0.472** | **0.446** | 0.477 | 0.456 | 0.476 | 0.450 | 0.447 | 0.410 |
| | 96 | 192 | **0.541** | **0.479** | 0.545 | 0.482 | 0.544 | 0.479 | 0.503 | 0.447 |
| | 96 | 336 | 0.619 | 0.515 | 0.619 | 0.515 | 0.619* | 0.515* | 0.545 | 0.470 |
| | 512 | 720 | 0.648 | 0.557 | 0.672 | 0.577 | 0.647* | 0.557* | 0.589 | 0.497 |
| CzeLan | 96 | 96 | 0.239 | 0.311 | 0.234 | 0.308 | 0.231 | 0.311 | 0.178 | 0.228 |
| | 96 | 192 | **0.282** | **0.348** | 0.270 | 0.344 | 0.283 | 0.349 | 0.210 | 0.252 |
| | 512 | 336 | **0.294** | **0.362** | 0.295 | 0.362 | 0.298 | 0.363 | 0.243 | 0.280 |
| | 96 | 720 | **0.368** | **0.410** | 0.376 | 0.418 | 0.426 | 0.449 | 0.290 | 0.326 |
| Covid-19 | 36 | 24 | 2.125 | **0.174** | 2.124 | 0.175 | 2.125 | 0.176* | 1.139 | 0.070 |
| | 36 | 36 | 2.463 | 0.177 | 2.489 | 0.228 | 2.463 | 0.177 | 1.582 | 0.091 |
| | 36 | 48 | 2.847 | 0.192 | 2.847 | 0.193 | 2.847 | 0.192 | 1.932 | 0.099 |
| | 36 | 60 | **3.273** | **0.205** | 3.272 | 0.205 | 3.275 | 0.212 | 2.682 | 0.127 |
| NASDAQ | 36 | 24 | **0.462** | **0.453** | 0.481 | 0.473 | 0.537 | 0.481 | 0.557 | 0.522 |
| | 36 | 36 | 0.844 | 0.636 | 0.818 | 0.629 | 0.808 | 0.628 | 0.869 | 0.668 |
| | 36 | 48 | **1.105** | **0.738** | 1.138 | 0.742 | 1.126* | 0.742* | 1.152 | 0.770 |
| | 36 | 60 | 1.252 | 0.786 | 1.229 | 0.783 | 1.251 | 0.783 | 1.284 | 0.809 |
| NYSE | 36 | 24 | 0.229 | 0.337 | 0.261 | 0.359 | 0.159 | 0.254 | 0.193 | 0.283 |
| | 36 | 36 | **0.283** | **0.335** | 0.279 | 0.334 | 0.289 | 0.344 | 0.315 | 0.356 |
| | 36 | 48 | 0.498 | 0.472 | 0.469 | 0.451 | 0.477 | 0.457 | 0.464 | 0.438 |
| | 36 | 60 | **0.649** | **0.549** | 0.669 | 0.551 | 0.693 | 0.586 | 0.631 | 0.522 |
| FRED-MD | 36 | 24 | 66.029 | 1.613 | 66.031 | 1.614 | 66.023* | 1.613* | 32.125 | 0.931 |
| | 36 | 36 | 94.379 | 1.872 | 94.334 | 1.878 | 94.359 | 1.863 | 58.332 | 1.258 |
| | 36 | 48 | 129.920 | **2.132** | 129.865 | 2.130 | 129.798 | 2.135 | 82.184 | 1.609 |
| | 36 | 60 | 173.275 | 2.426 | 173.648 | 2.430 | 173.596* | 2.432* | 109.625 | 1.882 |
| NN5 | 36 | 24 | 0.792 | 0.623 | 0.799 | 0.626 | 0.785 | 0.618 | 0.758 | 0.592 |
| | 36 | 36 | **0.706** | **0.591** | 0.733 | 0.608 | 0.727 | 0.606 | 0.693 | 0.577 |
| | 36 | 48 | 0.648 | 0.568 | 0.660 | 0.570 | 0.623 | 0.555 | 0.688 | 0.587 |
| | 36 | 60 | **0.626** | **0.555** | 0.634 | 0.562 | 0.630 | 0.559 | 0.679 | 0.587 |
| Wike2000 | 36 | 24 | **681.084** | **3.742** | 681.118 | 3.751 | 681.306 | 3.775 | 1135.609 | 1.350 |
| | 36 | 36 | 717.057 | 3.260 | 716.522 | 3.254 | 716.194 | 3.237 | 1060.939 | 1.388 |
| | 36 | 48 | **748.880** | **3.030** | 748.927 | 3.040 | 750.039 | 3.071 | 1899.937 | 1.733 |
| | 36 | 60 | **786.595** | **2.950** | 786.670 | 2.952 | 786.667 | 2.953 | 1281.245 | 1.570 |

Table 8: MSE and MAE scores of iTransformer-based models. The lower, the better. DYN-Post-processing score is in **bold** when it is better than the vanilla version.

| Dataset | Cont. L | Hor. H | iTransformer DYN-Post MSE | MAE | iTransformer MSE | MAE | NLinear MSE | MAE |
|---|---|---|---|---|---|---|---|---|
| ETTh1 | 96 | 96 | 0.386 | **0.404** | 0.386 | 0.405 | 0.385 | 0.403 |
|  | 512 | 192 | **0.423** | **0.438** | 0.424 | 0.440 | 0.422 | 0.426 |
|  | 512 | 336 | **0.439** | **0.452** | 0.449 | 0.460 | 0.431 | 0.429 |
|  | 96 | 720 | 0.552 | 0.518 | 0.495 | 0.487 | 0.439 | 0.452 |
| ETTh2 | 96 | 96 | 0.306 | 0.353 | 0.297 | 0.348 | 0.276 | 0.338 |
|  | 512 | 192 | **0.370** | **0.402** | 0.372 | 0.403 | 0.345 | 0.382 |
|  | 336 | 336 | **0.384** | **0.413** | 0.388 | 0.417 | 0.368 | 0.408 |
|  | 96 | 720 | **0.420** | **0.442** | 0.424 | 0.444 | 0.406 | 0.441 |
| ETTm1 | 336 | 96 | 0.301 | 0.356 | 0.300 | 0.353 | 0.301 | 0.343 |
|  | 336 | 192 | 0.351 | 0.386 | 0.341 | 0.380 | 0.355 | 0.379 |
|  | 336 | 336 | 0.379 | 0.401 | 0.374 | 0.396 | 0.372 | 0.385 |
|  | 512 | 720 | **0.428** | **0.429** | 0.429 | 0.430 | 0.430 | 0.418 |
| ETTm2 | 336 | 96 | 0.179 | 0.269 | 0.175 | 0.266 | 0.163 | 0.252 |
|  | 336 | 192 | **0.236** | **0.309** | 0.242 | 0.312 | 0.218 | 0.290 |
|  | 336 | 336 | 0.283 | 0.337 | 0.282 | 0.337 | 0.273 | 0.326 |
|  | 336 | 720 | 0.386 | 0.396 | 0.375 | 0.394 | 0.361 | 0.382 |
| Exchange | 96 | 96 | 0.087 | 0.207 | 0.086 | 0.205 | 0.085 | 0.204 |
|  | 96 | 192 | 0.177 | 0.299 | 0.177 | 0.299 | 0.175 | 0.297 |
|  | 96 | 336 | 0.331 | 0.417 | 0.331 | 0.417 | 0.320 | 0.409 |
|  | 96 | 720 | **0.843** | **0.691** | 0.846 | 0.693 | 0.838 | 0.690 |
| Weather | 336 | 96 | 0.158 | 0.207 | 0.157 | 0.207 | 0.180 | 0.226 |
|  | 512 | 192 | 0.203 | 0.249 | 0.200 | 0.248 | 0.218 | 0.261 |
|  | 336 | 336 | **0.251** | **0.286** | 0.252 | 0.287 | 0.266 | 0.296 |
|  | 512 | 720 | **0.319** | 0.336 | 0.320 | 0.336 | 0.334 | 0.345 |
| Electricity | 336 | 96 | **0.132** | **0.229** | 0.134 | 0.230 | 0.140 | 0.236 |
|  | 336 | 192 | **0.153** | **0.248** | 0.154 | 0.250 | 0.155 | 0.248 |
|  | 336 | 336 | 0.169 | 0.265 | 0.169 | 0.265 | 0.171 | 0.264 |
|  | 336 | 720 | **0.193** | **0.287** | 0.194 | 0.288 | 0.210 | 0.297 |
| ILI | 104 | 24 | **1.632** | **0.793** | 1.783 | 0.846 | 1.998 | 0.919 |
|  | 104 | 36 | **1.718** | 0.886 | 1.746 | 0.860 | 1.920 | 0.916 |
|  | 104 | 48 | 1.822 | 0.932 | 1.716 | 0.898 | 1.895 | 0.924 |
|  | 36 | 60 | **2.151** | **0.953** | 2.183 | 0.963 | 1.964 | 0.947 |
| Solar | 336 | 96 | 0.200 | 0.247 | 0.190 | 0.244 | 0.202 | 0.245 |
|  | 512 | 192 | 0.193 | 0.260 | 0.193 | 0.257 | 0.223 | 0.258 |
|  | 512 | 336 | **0.200** | **0.264** | 0.203 | 0.266 | 0.238 | 0.265 |
|  | 512 | 720 | **0.221** | **0.279** | 0.223 | 0.281 | 0.246 | 0.268 |
| Traffic | 512 | 96 | **0.362** | **0.264** | 0.363 | 0.265 | 0.395 | 0.272 |
|  | 512 | 192 | 0.384 | **0.272** | 0.384 | 0.273 | 0.407 | 0.277 |
|  | 512 | 336 | **0.395** | 0.278 | 0.396 | 0.277 | 0.417 | 0.282 |
|  | 512 | 720 | **0.440** | **0.304** | 0.445 | 0.308 | 0.453 | 0.302 |
| PEMS-BAY | 336 | 96 | 0.562 | 0.339 | 0.498 | 0.319 | 0.642 | 0.402 |
|  | 512 | 192 | 0.577 | 0.366 | 0.547 | 0.340 | 0.687 | 0.420 |
|  | 512 | 336 | 0.677 | 0.401 | 0.580 | 0.355 | 0.735 | 0.437 |
|  | 512 | 720 | 0.741 | 0.426 | 0.662 | 0.391 | 0.924 | 0.514 |
| METR-LA | 512 | 96 | 1.117 | 0.650 | 1.077 | 0.633 | 1.042 | 0.651 |
|  | 336 | 192 | 1.284 | **0.672** | 1.267 | 0.690 | 1.218 | 0.720 |
|  | 512 | 336 | 1.401 | 0.787 | 1.397 | 0.730 | 1.334 | 0.756 |
|  | 512 | 720 | **1.636** | 0.828 | 1.676 | 0.826 | 1.683 | 0.886 |
| PEMS04 | 512 | 96 | 0.127 | 0.235 | 0.125 | 0.230 | 0.208 | 0.301 |
|  | 512 | 192 | 0.163 | 0.261 | 0.145 | 0.249 | 0.229 | 0.312 |
|  | 512 | 336 | 0.164 | 0.266 | 0.158 | 0.263 | 0.251 | 0.331 |
|  | 512 | 720 | 0.204 | 0.305 | 0.195 | 0.300 | 0.346 | 0.398 |
| PEMS08 | 336 | 96 | 0.214 | 0.254 | 0.194 | 0.235 | 0.340 | 0.327 |
|  | 336 | 192 | 0.366 | 0.315 | 0.288 | 0.255 | 0.450 | 0.353 |
|  | 336 | 336 | 0.440 | 0.343 | 0.345 | 0.276 | 0.454 | 0.369 |
|  | 512 | 720 | **0.353** | **0.309** | 0.366 | 0.318 | 0.494 | 0.429 |
| AQShunyi | 512 | 96 | **0.647** | 0.481 | 0.650 | 0.479 | 0.653 | 0.486 |
|  | 336 | 192 | **0.689** | 0.500 | 0.693 | 0.498 | 0.701 | 0.506 |
|  | 336 | 336 | 0.713 | 0.511 | 0.713 | 0.510 | 0.722 | 0.519 |
|  | 512 | 720 | 0.776 | 0.547 | 0.776 | 0.537 | 0.777 | 0.545 |
| AQWan | 512 | 96 | **0.745** | **0.469** | 0.747 | 0.470 | 0.758 | 0.475 |
|  | 512 | 192 | 0.793 | 0.492 | 0.787 | 0.486 | 0.809 | 0.496 |
|  | 336 | 336 | **0.808** | 0.498 | 0.814 | 0.497 | 0.830 | 0.508 |
|  | 336 | 720 | 0.899 | 0.534 | 0.889 | 0.529 | 0.906 | 0.538 |
| Wind | 512 | 96 | **0.900** | 0.648 | 0.901 | 0.646 | 0.923 | 0.640 |
|  | 512 | 192 | 1.089 | 0.742 | 1.085 | 0.740 | 1.081 | 0.734 |
|  | 512 | 336 | 1.228 | 0.807 | 1.222 | 0.805 | 1.228 | 0.805 |
|  | 512 | 720 | 1.348 | 0.860 | 1.325 | 0.850 | 1.328 | 0.852 |
| ZafNoo | 336 | 96 | 0.439 | 0.410 | 0.439 | 0.408 | 0.447 | 0.410 |
|  | 336 | 192 | **0.503** | 0.449 | 0.505 | 0.443 | 0.503 | 0.447 |
|  | 336 | 336 | 0.573 | 0.489 | 0.555 | 0.473 | 0.545 | 0.470 |
|  | 512 | 720 | 0.619 | 0.519 | 0.591 | 0.501 | 0.589 | 0.497 |
| CzeLan | 512 | 96 | **0.170** | **0.234** | 0.177 | 0.239 | 0.178 | 0.228 |
|  | 512 | 192 | 0.203 | 0.259 | 0.201 | 0.257 | 0.210 | 0.252 |
|  | 512 | 336 | 0.234 | 0.287 | 0.232 | 0.282 | 0.243 | 0.280 |
|  | 512 | 720 | **0.258** | **0.308** | 0.261 | 0.311 | 0.290 | 0.326 |
| Covid-19 | 36 | 24 | **0.932** | **0.035** | 1.001 | 0.038 | 1.139 | 0.070 |
|  | 36 | 36 | **1.227** | **0.041** | 1.236 | 0.042 | 1.582 | 0.091 |
|  | 36 | 48 | **1.658** | **0.053** | 1.710 | 0.056 | 1.932 | 0.099 |
|  | 36 | 60 | 2.036 | **0.061** | 2.005 | 0.062 | 2.682 | 0.127 |
| NASDAQ | 104 | 24 | **0.506** | **0.513** | 0.570 | 0.540 | 0.557 | 0.522 |
|  | 104 | 36 | 0.855 | 0.695 | 0.691 | 0.600 | 0.869 | 0.668 |
|  | 104 | 48 | **1.091** | **0.758** | 1.188 | 0.773 | 1.152 | 0.770 |
|  | 36 | 60 | 1.425 | 0.839 | 1.325 | 0.820 | 1.284 | 0.809 |
| NYSE | 36 | 24 | 0.225 | 0.304 | 0.225 | 0.302 | 0.193 | 0.283 |
|  | 36 | 36 | **0.390** | 0.418 | 0.392 | 0.409 | 0.315 | 0.356 |
|  | 36 | 48 | **0.510** | **0.460** | 0.529 | 0.480 | 0.464 | 0.438 |
|  | 36 | 60 | 0.700 | **0.554** | 0.687 | 0.557 | 0.631 | 0.522 |
| FRED-MD | 36 | 24 | **27.040** | **0.903** | 28.581 | 0.917 | 32.125 | 0.931 |
|  | 36 | 36 | **48.389** | **1.226** | 54.221 | 1.276 | 58.332 | 1.258 |
|  | 36 | 48 | **85.573** | **1.563** | 89.574 | 1.607 | 82.184 | 1.609 |
|  | 36 | 60 | **124.354** | **1.918** | 130.061 | 1.947 | 109.625 | 1.882 |
| NN5 | 104 | 24 | **0.719** | 0.570 | 0.727 | 0.568 | 0.758 | 0.592 |
|  | 104 | 36 | **0.658** | 0.552 | 0.664 | 0.552 | 0.693 | 0.577 |
|  | 104 | 48 | **0.632** | 0.546 | 0.633 | 0.543 | 0.688 | 0.587 |
|  | 104 | 60 | 0.617 | 0.541 | 0.615 | 0.537 | 0.679 | 0.587 |
| Wike2000 | 36 | 24 | 458.891 | 1.017 | 453.475 | 1.011 | 1135.609 | 1.350 |
|  | 36 | 36 | 635.855 | 1.225 | 515.830 | 1.132 | 1060.939 | 1.388 |
|  | 36 | 48 | **539.686** | **1.176** | 578.335 | 1.214 | 1899.937 | 1.733 |
|  | 104 | 60 | 644.895 | **1.398** | 634.947 | 1.402 | 1281.245 | 1.570 |

Table 9: MSE and MAE scores of PatchTST-based models. The lower, the better. `DYN-Post`-processing score is in **bold** when it is better than the vanilla version.

| Dataset | Cont. $L$ | Hor. $H$ | PatchTST DYN-Post MSE | MAE | PatchTST MSE | MAE | NLinear MSE | MAE |
|---|---|---|---|---|---|---|---|---|
| ETTh1 | 96 | 96 | 0.379 | **0.395** | 0.377 | 0.397 | 0.385 | 0.403 |
| | 512 | 192 | **0.401** | **0.418** | 0.409 | 0.425 | 0.422 | 0.426 |
| | 512 | 336 | **0.423** | **0.433** | 0.431 | 0.444 | 0.431 | 0.429 |
| | 512 | 720 | 0.457 | **0.474** | 0.457 | 0.477 | 0.439 | 0.452 |
| ETTh2 | 512 | 96 | 0.281 | 0.345 | 0.274 | 0.337 | 0.276 | 0.338 |
| | 512 | 192 | **0.341** | **0.382** | 0.348 | 0.384 | 0.345 | 0.382 |
| | 512 | 336 | **0.372** | **0.413** | 0.377 | 0.416 | 0.368 | 0.408 |
| | 336 | 720 | 0.415 | 0.445 | 0.406 | 0.441 | 0.406 | 0.441 |
| ETTm1 | 336 | 96 | 0.290 | 0.345 | 0.289 | 0.343 | 0.301 | 0.343 |
| | 336 | 192 | **0.328** | **0.367** | 0.329 | 0.368 | 0.355 | 0.379 |
| | 336 | 336 | **0.361** | **0.386** | 0.362 | 0.390 | 0.372 | 0.385 |
| | 512 | 720 | 0.417 | **0.421** | 0.416 | 0.423 | 0.430 | 0.418 |
| ETTm2 | 336 | 96 | 0.168 | 0.256 | 0.165 | 0.255 | 0.163 | 0.252 |
| | 336 | 192 | 0.223 | 0.296 | 0.221 | 0.293 | 0.218 | 0.290 |
| | 512 | 336 | **0.274** | 0.332 | 0.276 | 0.327 | 0.273 | 0.326 |
| | 336 | 720 | 0.367 | 0.389 | 0.362 | 0.381 | 0.361 | 0.382 |
| Exchange | 96 | 96 | **0.084** | **0.201** | 0.087 | 0.204 | 0.085 | 0.204 |
| | 96 | 192 | 0.177 | **0.298** | 0.177 | 0.300 | 0.175 | 0.297 |
| | 512 | 336 | 0.338 | 0.424 | 0.297 | 0.399 | 0.320 | 0.409 |
| | 96 | 720 | 0.885 | 0.709 | 0.843 | 0.692 | 0.838 | 0.690 |
| Weather | 336 | 96 | 0.156 | 0.207 | 0.150 | 0.200 | 0.180 | 0.226 |
| | 512 | 192 | 0.195 | 0.246 | 0.191 | 0.239 | 0.218 | 0.261 |
| | 512 | 336 | 0.247 | 0.283 | 0.242 | 0.279 | 0.266 | 0.296 |
| | 512 | 720 | 0.319 | 0.336 | 0.312 | 0.330 | 0.334 | 0.345 |
| Electricity | 512 | 96 | 0.151 | 0.257 | 0.143 | 0.247 | 0.140 | 0.236 |
| | 512 | 192 | 0.162 | 0.266 | 0.158 | 0.260 | 0.155 | 0.248 |
| | 512 | 336 | 0.167 | **0.264** | 0.168 | 0.267 | 0.171 | 0.264 |
| | 512 | 720 | 0.215 | 0.307 | 0.214 | 0.307 | 0.210 | 0.297 |
| ILI | 104 | 24 | 2.204 | 0.998 | 1.932 | 0.872 | 1.998 | 0.919 |
| | 104 | 36 | 2.162 | 0.995 | 1.869 | 0.866 | 1.920 | 0.916 |
| | 104 | 48 | 2.193 | 0.998 | 1.891 | 0.883 | 1.895 | 0.924 |
| | 104 | 60 | 2.177 | 1.000 | 1.914 | 0.896 | 1.964 | 0.947 |
| Solar | 512 | 96 | 0.181 | 0.239 | 0.170 | 0.234 | 0.202 | 0.245 |
| | 512 | 192 | 0.207 | **0.289** | 0.203* | 0.302 | 0.223 | 0.258 |
| | 512 | 336 | 0.216 | 0.305 | 0.212 | 0.293 | 0.238 | 0.265 |
| | 512 | 720 | 0.223 | 0.309 | 0.215 | 0.307 | 0.246 | 0.268 |
| Traffic | 512 | 96 | 0.385 | 0.275 | 0.370 | 0.262 | 0.395 | 0.272 |
| | 512 | 192 | 0.397 | 0.281 | 0.386 | 0.269 | 0.407 | 0.277 |
| | 512 | 336 | 0.409 | 0.288 | 0.396 | 0.275 | 0.417 | 0.282 |
| | 512 | 720 | 0.446 | 0.309 | 0.435 | 0.295 | 0.453 | 0.302 |
| PEMS-BAY | 512 | 96 | 0.586 | 0.367 | 0.566 | 0.355 | 0.642 | 0.402 |
| | 512 | 192 | 0.654 | 0.401 | 0.649 | 0.394 | 0.687 | 0.420 |
| | 512 | 336 | 0.701 | 0.416 | 0.700 | 0.412 | 0.735 | 0.437 |
| | 512 | 720 | **0.830** | 0.462 | 0.843 | 0.460 | 0.924 | 0.514 |
| METR-LA | 512 | 96 | 1.036 | 0.647 | 1.028 | 0.643 | 1.042 | 0.651 |
| | 512 | 192 | 1.209 | 0.714 | 1.156 | 0.683 | 1.218 | 0.720 |
| | 512 | 336 | 1.324 | 0.743 | 1.252 | 0.736 | 1.334 | 0.756 |
| | 512 | 720 | 1.557 | 0.845 | 1.433 | 0.793 | 1.683 | 0.886 |
| PEMS04 | 336 | 96 | 0.174 | 0.284 | 0.169 | 0.282 | 0.208 | 0.301 |
| | 336 | 192 | 0.205 | 0.309 | 0.202 | 0.309 | 0.229 | 0.312 |
| | 336 | 336 | 0.218 | 0.320 | 0.211 | 0.315 | 0.251 | 0.331 |
| | 336 | 720 | **0.256** worst case | **0.351** worst case | 0.257 | 0.352 | 0.346 | 0.398 |
| PEMS08 | 512 | 96 | **0.240** | **0.281** | 0.248 | 0.287 | 0.340 | 0.327 |
| | 512 | 192 | 0.330 | **0.301** | 0.319 | 0.310 | 0.450 | 0.353 |
| | 512 | 336 | 0.382 | **0.323** | 0.361 | 0.324 | 0.454 | 0.369 |
| | 512 | 720 | **0.396** | **0.351** | 0.399 | 0.368 | 0.494 | 0.429 |
| AQShunyi | 512 | 96 | 0.651 | 0.483 | 0.646 | 0.478 | 0.653 | 0.486 |
| | 512 | 192 | 0.691 | 0.502 | 0.688 | 0.498 | 0.701 | 0.506 |
| | 512 | 336 | 0.713 | 0.515 | 0.710 | 0.513 | 0.722 | 0.519 |
| | 512 | 720 | 0.769 | 0.540 | 0.768 | 0.539 | 0.777 | 0.545 |
| AQWan | 512 | 96 | 0.750 | 0.472 | 0.745 | 0.468 | 0.758 | 0.475 |
| | 512 | 192 | 0.797 | 0.493 | 0.793 | 0.490 | 0.809 | 0.496 |
| | 512 | 336 | 0.820 | 0.504 | 0.819 | 0.502 | 0.830 | 0.508 |
| | 512 | 720 | 0.894 | 0.535 | 0.890 | 0.533 | 0.906 | 0.538 |
| Wind | 512 | 96 | 0.908 | 0.653 | 0.877 | 0.643 | 0.923 | 0.640 |
| | 512 | 192 | 1.090 | 0.743 | 1.065 | 0.741 | 1.081 | 0.734 |
| | 512 | 336 | 1.229 | 0.808 | 1.202 | 0.802 | 1.228 | 0.805 |
| | 512 | 720 | 1.337 | 0.861 | 1.300 | 0.846 | 1.328 | 0.852 |
| ZafNoo | 512 | 96 | 0.443 | 0.419 | 0.429 | 0.405 | 0.447 | 0.410 |
| | 512 | 192 | 0.502 | 0.456 | 0.494 | 0.449 | 0.503 | 0.447 |
| | 512 | 336 | 0.545 | 0.479 | 0.538 | 0.475 | 0.545 | 0.470 |
| | 512 | 720 | 0.583 | 0.499 | 0.573 | 0.486 | 0.589 | 0.497 |
| CzeLan | 512 | 96 | 0.184 | 0.240 | 0.176 | 0.232 | 0.178 | 0.228 |
| | 512 | 192 | 0.211 | 0.268 | 0.205 | 0.263 | 0.210 | 0.252 |
| | 512 | 336 | **0.232** | **0.279** | 0.236 | 0.286 | 0.243 | 0.280 |
| | 512 | 720 | **0.269** | **0.313** | 0.270 | 0.316 | 0.290 | 0.326 |
| Covid-19 | 36 | 24 | **1.016** | 0.042 | 1.045 | 0.042 | 1.139 | 0.070 |
| | 36 | 36 | **1.372** | 0.052 | 1.397 | 0.051 | 1.582 | 0.091 |
| | 36 | 48 | **1.692** | **0.060** | 1.769 | 0.062 | 1.932 | 0.099 |
| | 36 | 60 | **2.161** | 0.069 | 2.216 | 0.068 | 2.682 | 0.127 |
| NASDAQ | 36 | 24 | 0.677 | 0.607 | 0.649 | 0.567 | 0.557 | 0.522 |
| | 104 | 36 | 0.834 | **0.677** | 0.821 | 0.682 | 0.869 | 0.668 |
| | 104 | 48 | **1.021** | **0.752** | 1.169 | 0.793 | 1.152 | 0.770 |
| | 104 | 60 | 1.247 | **0.842** | 1.247 | 0.843 | 1.284 | 0.809 |
| NYSE | 36 | 24 | 0.232 | 0.305 | 0.226 | 0.296 | 0.193 | 0.283 |
| | 36 | 36 | **0.350** | **0.370** | 0.380 | 0.389 | 0.315 | 0.356 |
| | 36 | 48 | **0.483** | **0.436** | 0.575 | 0.492 | 0.464 | 0.438 |
| | 36 | 60 | **0.642** | **0.523** | 0.749 | 0.572 | 0.631 | 0.522 |
| FRED-MD | 36 | 24 | 33.754 | 1.007 | 32.808 | 0.962 | 32.125 | 0.931 |
| | 36 | 36 | 61.611 | 1.358 | 61.035 | 1.345 | 58.332 | 1.258 |
| | 36 | 48 | 92.930 | 1.657 | 91.835 | 1.648 | 82.184 | 1.609 |
| | 36 | 60 | 130.365 | 2.026 | 127.018 | 1.958 | 109.625 | 1.882 |
| NN5 | 104 | 24 | **0.726** | **0.590** | 0.740 | 0.596 | 0.758 | 0.592 |
| | 104 | 36 | **0.679** | **0.582** | 0.694 | 0.595 | 0.693 | 0.577 |
| | 104 | 48 | **0.648** | **0.571** | 0.667 | 0.585 | 0.688 | 0.587 |
| | 104 | 60 | **0.643** | **0.576** | 0.653 | 0.582 | 0.679 | 0.587 |
| Wike2000 | 36 | 24 | **449.087** | 1.040 | 457.183 | 1.023 | 1135.609 | 1.350 |
| | 36 | 36 | **494.895** | 1.184 | 511.944 | 1.115 | 1060.939 | 1.388 |
| | 36 | 48 | **508.306** | 1.219 | 531.900 | 1.179 | 1899.937 | 1.733 |
| | 36 | 60 | 629.172 | 1.336 | 554.829 | 1.244 | 1281.245 | 1.570 |

Table 10: MSE and MAE scores of Crossformer-based models. The lower, the better. `DYN`-Post-processing score is in **bold** when it is better than the vanilla version. Vanilla score is starred* when we obtained better results than the TFB benchmark with updated learning hyperparameters.

| Dataset | Cont. $L$ | Hor. $H$ | Crossformer `DYN`-Post MSE | MAE | Crossformer MSE | MAE | NLinear MSE | MAE |
|---|---|---|---|---|---|---|---|---|
| ETTh1 | 96 | 96 | 0.439 | 0.452 | 0.411 | 0.435 | 0.385 | 0.403 |
| | 512 | 192 | 0.436 | 0.457 | 0.409 | 0.438 | 0.422 | 0.426 |
| | 512 | 336 | 0.460 | 0.472 | 0.433 | 0.457 | 0.431 | 0.429 |
| | 512 | 720 | 0.515 | 0.525 | 0.501 | 0.514 | 0.439 | 0.452 |
| ETTh2 | 336 | 96 | **0.670** | **0.579** | 0.728 | 0.603 | 0.276 | 0.338 |
| | 336 | 192 | 0.843 | 0.659 | 0.723 | 0.607 | 0.345 | 0.382 |
| | 336 | 336 | 0.845 | 0.659 | 0.740 | 0.628 | 0.368 | 0.408 |
| | 336 | 720 | **1.090** | **0.781** | 1.386 | 0.882 | 0.406 | 0.441 |
| ETTm1 | 512 | 96 | 0.336 | 0.386 | 0.314 | 0.367 | 0.301 | 0.343 |
| | 512 | 192 | 0.406 | 0.431 | 0.374 | 0.410 | 0.355 | 0.379 |
| | 512 | 336 | 0.415 | 0.435 | 0.413 | 0.432 | 0.372 | 0.385 |
| | 336 | 720 | **0.720** | **0.582** | 0.753 | 0.613 | 0.430 | 0.418 |
| ETTm2 | 512 | 96 | **0.266** | **0.361** | 0.296 | 0.391 | 0.163 | 0.252 |
| | 512 | 192 | 0.474 | 0.472 | 0.369 | 0.416 | 0.218 | 0.290 |
| | 96 | 336 | **0.530** | **0.518** | 0.588 | 0.600 | 0.273 | 0.326 |
| | 512 | 720 | 1.563 | 0.861 | 0.750 | 0.612 | 0.361 | 0.382 |
| Exchange | 96 | 96 | **0.222** | **0.350** | 0.231 | 0.356 | 0.085 | 0.204 |
| | 96 | 192 | 0.480 | 0.522 | 0.460 | 0.509 | 0.175 | 0.297 |
| | 96 | 336 | **0.769** | **0.701** | 1.034 | 0.825 | 0.320 | 0.409 |
| | 336 | 720 | **1.253** | **0.949** | 1.576 | 1.021 | 0.838 | 0.690 |
| Weather | 512 | 96 | 0.154 | 0.229 | 0.143 | 0.210 | 0.180 | 0.226 |
| | 336 | 192 | 0.198 | 0.264 | 0.195 | 0.260 | 0.218 | 0.261 |
| | 96 | 336 | 0.276 | 0.340 | 0.254 | 0.319 | 0.266 | 0.296 |
| | 512 | 720 | 0.346 | 0.388 | 0.335 | 0.385 | 0.334 | 0.345 |
| Electricity | 512 | 96 | 0.136 | 0.235 | 0.134 | 0.231 | 0.140 | 0.236 |
| | 512 | 192 | 0.151 | 0.248 | 0.146 | 0.243 | 0.155 | 0.248 |
| | 512 | 336 | 0.186 | 0.280 | 0.165 | 0.264 | 0.171 | 0.264 |
| | 512 | 720 | 0.237 | **0.313** | 0.237 | 0.314 | 0.210 | 0.297 |
| ILI | 104 | 24 | 3.222 | 1.197 | 2.981 | 1.096 | 1.998 | 0.919 |
| | 104 | 36 | 3.903 | 1.356 | 3.549 | 1.196 | 1.920 | 0.916 |
| | 104 | 48 | 4.032 | 1.377 | 3.851 | 1.288 | 1.895 | 0.924 |
| | 104 | 60 | 4.896 | 1.509 | 4.692 | 1.450 | 1.964 | 0.947 |
| Solar | 512 | 96 | **0.164** | 0.231 | 0.183 | 0.208 | 0.202 | 0.245 |
| | 512 | 192 | **0.206** | 0.244 | 0.208 | 0.226 | 0.223 | 0.258 |
| | 512 | 336 | **0.206** | 0.260 | 0.212 | 0.239 | 0.238 | 0.265 |
| | 512 | 720 | **0.201** | **0.237** | 0.215 | 0.256 | 0.246 | 0.268 |
| Traffic | 96 | 96 | 0.547 | 0.311 | 0.526 | 0.288 | 0.395 | 0.272 |
| | 336 | 192 | **0.489** | 0.273 | 0.503 | 0.263 | 0.407 | 0.277 |
| | 512 | 336 | 0.528 | 0.298 | 0.505 | 0.276 | 0.417 | 0.282 |
| | 336 | 720 | 0.600 | 0.338 | 0.552 | 0.301 | 0.453 | 0.302 |
| PEMS-BAY | 512 | 96 | 0.454 | 0.305 | 0.435 | 0.297 | 0.642 | 0.402 |
| | 512 | 192 | 0.472 | 0.319 | 0.470 | 0.317 | 0.687 | 0.420 |
| | 512 | 336 | **0.493** | **0.323** | 0.495 | 0.326 | 0.735 | 0.437 |
| | 512 | 720 | 0.607 | **0.363** | 0.605 | 0.364 | 0.924 | 0.514 |
| METR-LA | 512 | 96 | 1.212 | 0.648 | 1.069 | 0.606 | 1.042 | 0.651 |
| | 512 | 192 | 1.392 | **0.660** | 1.166 | 0.689 | 1.218 | 0.720 |
| | 512 | 336 | **1.383** | **0.776** | 1.405 | 0.777 | 1.334 | 0.756 |
| | 512 | 720 | 1.566 | **0.782** | 1.421 | 0.808 | 1.683 | 0.886 |
| PEMS04 | 336 | 96 | 0.123 | 0.237 | 0.122 | 0.236 | 0.208 | 0.301 |
| | 336 | 192 | **0.168** | **0.267** | 0.173 | 0.271 | 0.229 | 0.312 |
| | 336 | 336 | **0.188** | **0.298** | 0.208 | 0.307 | 0.251 | 0.331 |
| | 512 | 720 | **0.239** | **0.333** | 0.279 | 0.362 | 0.346 | 0.398 |
| PEMS08 | 512 | 96 | **0.201** | **0.235** | 0.230 | 0.260 | 0.340 | 0.327 |
| | 512 | 192 | **0.226** | **0.249** | 0.239 | 0.264 | 0.450 | 0.353 |
| | 512 | 336 | **0.245** | **0.268** | 0.272 | 0.289 | 0.454 | 0.369 |
| | 512 | 720 | **0.301** | **0.304** | 0.320 | 0.316 | 0.494 | 0.429 |
| AQShunyi | 512 | 96 | 0.666 | 0.503 | 0.652 | 0.484 | 0.653 | 0.486 |
| | 512 | 192 | 0.725 | 0.519 | 0.674 | 0.499 | 0.701 | 0.506 |
| | 336 | 336 | 0.710 | 0.517 | 0.704 | 0.515 | 0.722 | 0.519 |
| | 512 | 720 | 0.764 | 0.535 | 0.747 | 0.518 | 0.777 | 0.545 |
| AQWan | 512 | 96 | 0.757 | 0.473 | 0.750 | 0.465 | 0.758 | 0.475 |
| | 512 | 192 | 0.799 | 0.495 | 0.762 | 0.479 | 0.809 | 0.496 |
| | 336 | 336 | 0.805 | 0.505 | 0.802 | 0.504 | 0.830 | 0.508 |
| | 512 | 720 | 0.837 | 0.515 | 0.829 | 0.512 | 0.906 | 0.538 |
| Wind | 512 | 96 | 0.836 | 0.630 | 0.784 | 0.590 | 0.923 | 0.640 |
| | 512 | 192 | 1.097 | 0.739 | 0.977 | 0.697 | 1.081 | 0.734 |
| | 512 | 336 | 1.162 | 0.776 | 1.073 | 0.755 | 1.228 | 0.805 |
| | 96 | 720 | 1.201 | 0.808 | 1.191 | 0.803 | 1.328 | 0.852 |
| ZafNoo | 96 | 96 | 0.433 | 0.421 | 0.430 | 0.418 | 0.447 | 0.410 |
| | 96 | 192 | 0.482 | **0.448** | 0.479 | 0.449 | 0.503 | 0.447 |
| | 96 | 336 | 0.513 | 0.469 | 0.505 | 0.464 | 0.545 | 0.470 |
| | 96 | 720 | 0.565 | 0.503 | 0.560 | 0.494 | 0.589 | 0.497 |
| CzeLan | 512 | 96 | 0.705 | 0.496 | 0.581 | 0.443 | 0.178 | 0.228 |
| | 336 | 192 | 0.810 | 0.562 | 0.705 | 0.503 | 0.210 | 0.252 |
| | 336 | 336 | 1.016 | 0.625 | 0.971 | 0.596 | 0.243 | 0.280 |
| | 96 | 720 | 1.655 | 0.797 | 1.566 | 0.762 | 0.290 | 0.326 |
| Covid-19 | 104 | 24 | 1771.717 | 2.427 | 1768.817 | 2.314 | 1.139 | 0.070 |
| | 104 | 36 | 1773.242 | 2.507 | 1770.939 | 2.346 | 1.582 | 0.091 |
| | 36 | 48 | **1772.789** | **2.443** | 1773.447 | 2.450 | 1.932 | 0.099 |
| | 104 | 60 | **1772.296** | **2.456** | 1772.833 | 2.486 | 2.682 | 0.127 |
| NASDAQ | 36 | 24 | **1.123** | **0.737** | 1.149 | 0.745 | 0.557 | 0.522 |
| | 36 | 36 | 1.716 | 0.969 | 1.414 | 0.885 | 0.869 | 0.668 |
| | 36 | 48 | 2.204 | 1.143 | 2.108 | 1.136 | 1.152 | 0.770 |
| | 104 | 60 | 2.801 | 1.355 | 2.276 | 1.201 | 1.284 | 0.809 |
| NYSE | 36 | 24 | **0.478** | **0.631** | 0.820 | 0.841 | 0.193 | 0.283 |
| | 36 | 36 | **0.854** | **0.830** | 0.942 | 0.904 | 0.315 | 0.356 |
| | 36 | 48 | **0.758** | **0.751** | 1.049 | 0.955 | 0.464 | 0.438 |
| | 36 | 60 | 1.353 | 1.018 | 1.121 | 0.937 | 0.631 | 0.522 |
| FRED-MD | 104 | 24 | **381.897** | 3.785 | 385.599 | 3.559 | 32.125 | 0.931 |
| | 104 | 36 | **395.499** | 3.887 | 398.728 | 3.716 | 58.332 | 1.258 |
| | 104 | 48 | 414.542 | 4.300 | 414.353 | 3.939 | 82.184 | 1.609 |
| | 104 | 60 | 423.996 | 4.690 | 422.864 | 4.093 | 109.625 | 1.882 |
| NN5 | 104 | 24 | 0.732 | 0.590 | 0.734* | 0.584* | 0.758 | 0.592 |
| | 104 | 36 | **0.674** | 0.570 | 0.678* | 0.589 | 0.693 | 0.577 |
| | 104 | 48 | 0.660 | 0.570 | 0.634* | 0.551* | 0.688 | 0.587 |
| | 104 | 60 | 0.665 | 0.580 | 0.660* | 0.570* | 0.679 | 0.587 |
| Wike2000 | 104 | 24 | 639.111 | 1.873 | 638.794 | 1.734 | 1135.609 | 1.350 |
| | 104 | 36 | **692.271** | 1.845 | 692.485 | 1.808 | 1060.939 | 1.388 |
| | 104 | 48 | **717.538** | 1.976 | 718.072 | 1.899 | 1899.937 | 1.733 |
| | 104 | 60 | 730.390 | **1.983** | 730.368 | 2.021 | 1281.245 | 1.570 |

## E.2 DETAILED RESULTS OF SECTION 4.2

Detailed results of Section 4.2 are presented here. For Figure 4, counts of the distribution are shown, with the p-values $p_v$ of MSE and MAE. For RQ1 models, in Table 11, to assess statistical significance, we perform the following unilateral Wilcoxon test: $H_0$: *"DYN added model MAE (or MSE) is not statistically lower than its vanilla version"*, and $H_1$: *"DYN added model MAE (or MSE) is statistically lower than its vanilla version"*. For RQ2 models, in Table 12, the performed unilateral Wilcoxon test is: $H_0$: *"Vanilla model MAE (or MSE) is not statistically lower than its post-processing version"*, and $H_1$: *"Vanilla model MAE (or MSE) is statistically lower than its post-processing version"*.

Table 11: Detailed counts of RQ1 DYN added models relative performance against their vanilla version with p-values $p_v$ (in **bold** if less than 0.05).

| Model | Better | Iso | Worse $\leq 1\%$ | Worse $> 1\%$ | $p_{v,MSE}$ | $p_{v,MAE}$ |
|---|---|---|---|---|---|---|
| Informer | 153 (77%) | 4 (2%) | 13 (6%) | 30 (15%) | **4.0e−8** | **9.7e−11** |
| FiLM | 169 (85%) | 11 (5%) | 6 (3%) | 14 (7%) | **7.6e−11** | **1.2e−11** |
| MICN | 101 (51%) | 31 (15%) | 28 (14%) | 40 (20%) | **1.3e−2** | 6.5e−2 |
| FEDformer | 113 (57%) | 20 (10%) | 34 (17%) | 33 (16%) | **3.8e−3** | **2.3e−2** |

Table 12: Detailed counts of RQ2 post-processing models relative performance against their vanilla version with p-values $p_v$ (in **bold** if less than 0.05).

| Model | Better | Iso | Worse $\leq 1\%$ | Worse $> 1\%$ | $p_{v,MSE}$ | $p_{v,MAE}$ |
|---|---|---|---|---|---|---|
| iTransformer | 85 (43%) | 16 (8%) | 31 (15%) | 68 (34%) | 0.18 | **1.1e−2** |
| PatchTST | 60 (30%) | 7 (4%) | 38 (19%) | 95 (47%) | **1.0e−3** | **4.8e−5** |
| Crossformer | 65 (32%) | 1 (1%) | 29 (15%) | 105 (52%) | **7.9e−3** | **2.4e−4** |

Figure 7 is a detailed version of Table 2 where it proposes a visual of the performance shift due to the addition of a linear DYN layer. It also adds the average gain and loss of the dynamics addition, normalized by NLinear scores. It shows that **the mean performance is better and the average gain is greater than the average loss** with the DYN layer addition, supporting our model update. The average gain is greater on Informer, which is the vanilla model with the most static DYN function (0-padding).

Then, in Figure 8, we show the same scatter plot with the outliers that were removed for visualization and consistency in Figure 7. Outliers, with $q_\alpha$ the $\alpha$-quantile where $\alpha \in [0, 1]$, were the ones:

- upper than $q_{0.985}$ and lower than $q_{0.015}$ along both $x$ and $y$ axis for Informer,

- none for FiLM,

- upper than $q_{0.965}$ and lower than $q_{0.035}$ along both $x$ and $y$ axis for MICN,

- upper than $q_{0.99}$ and lower than $q_{0.01}$ along both $x$ and $y$ axis for FEDformer.

Tendencies are the same as in Figure 7, except for Informer, where the average loss is greater than the average gain. That is due to the Traffic case when $H = 192$ where the DYN added model is worse than the vanilla version by a large margin, which is an isolated case, removed for consistency.

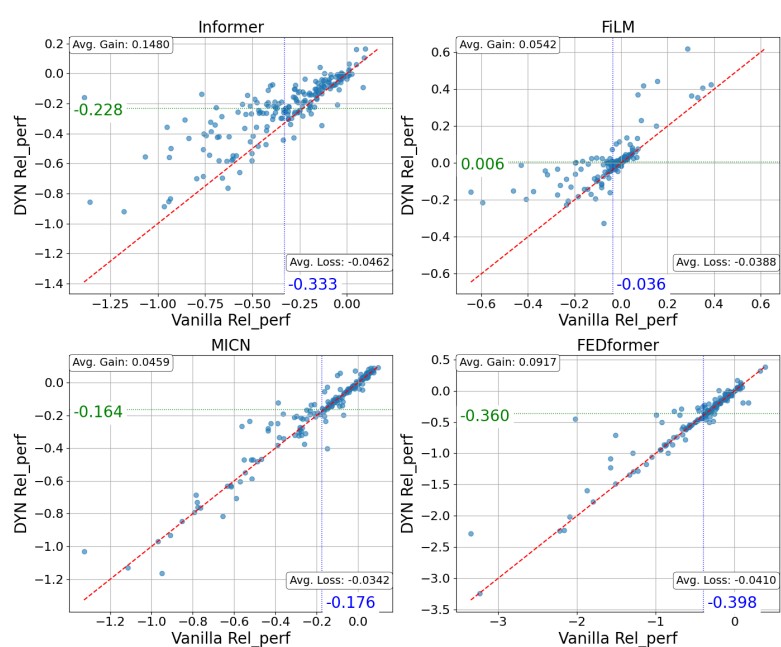

Figure 7: Comparison between `DYN` added model performance against their vanilla version, normalized by NLinear scores. Each point corresponds to $(x; y) : (\text{Rel\_perf}(\text{Vanilla}\|\text{NLinear}); \text{Rel\_perf}(\texttt{DYN}\|\text{NLinear}))$, where $\text{Rel\_perf}(.\|\text{NLinear}) = \frac{\text{score}(\text{NLinear}) - \text{score}(.)}{\text{score}(\text{NLinear})}$, where score is MSE or MAE, for each dataset and forecasting horizon. Points above the diagonal indicate improvement with `DYN` addition. The mean is shown on the corresponding axis (reported in Table 2). Average gain is $\mathbb{E}[\, y - x \mid y > x \,]$, while average loss is $\mathbb{E}[\, y - x \mid y < x \,]$. Outliers are removed for clarity and consistency (see Figure 8 below with the outliers).

.

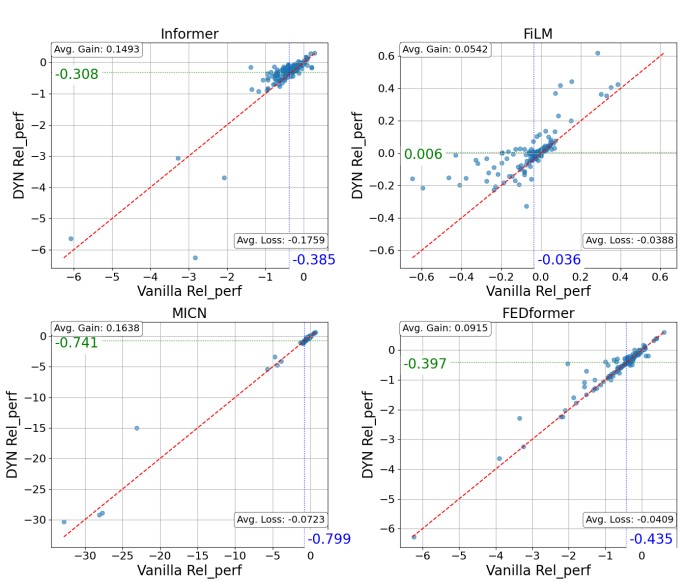

Figure 8: Same scatter plot as in Figure 7 above with outliers included.

.

### E.3 DETAILED RESULTS OF SECTION 4.3

Detailed results of Section 4.3 are presented here. Again, RQ1 DYN added models are compared to their PRO versions, while RQ2 post-processing models are compared to their vanilla versions. For Figure 5, counts of the distribution are shown, with the p-values $p_v$ of MSE and MAE in Table 13 for RQ1 and in Table 14 for RQ2. For RQ1 models, in Table 13, the unilateral Wilcoxon test is: $H_0$: *"DYN added model MAE (or MSE) is not statistically lower than its PRO version"*, and $H_1$: *"DYN added model MAE (or MSE) is statistically lower than its PRO version"*. For RQ2 models, in Table 14, the unilateral Wilcoxon test is the same as in Section 4.2, conditioned by the setup.

Table 13: Detailed results of RQ1 DYN added models relative performance with setup conditioning against their PRO version with p-values $p_v$ (in **bold** if less than 0.05). For the counts and percentages, the order corresponds to the scores where: DYN is (Better || Iso || Worse) than PRO.

| Model | Informer | FiLM | FEDformer |
|---|---|---|---|
| DYN VS PRO | 89\|\|65\|\|46 | 54\|\|75\|\|71 | 119\|\|18\|\|63 |
| % | 45%\|\|32%\|\|23% | 27%\|\|38%\|\|35% | 60%\|\|9%\|\|31% |
| $p_{v,MSE}; p_{v,MAE}$ | **3.1**e−**3**; **9.1**e−**5** | 0.74; 0.83 | **9.1**e−**3**; **6.0**e−**3** |
| DYN VS PRO $\mid H > L$ | 77\|\|13\|\|28 | 23\|\|13\|\|18 | 77\|\|6\|\|37 |
| % | 65%\|\|11%\|\|24% | 43%\|\|24%\|\|33% | 64%\|\|5%\|\|31% |
| $p_{v,MSE}; p_{v,MAE}$ | **3.7**e−**4**; **1.4**e−**5** | 0.27; 0.60 | **2.7**e−**2**; **3.0**e−**2** |
| DYN VS PRO $\mid H = L$ | 0\|\|44\|\|0 | 0\|\|30\|\|0 | 25\|\|6\|\|21 |
| % | 0%\|\|100%\|\|0% | 0%\|\|100%\|\|0% | 48%\|\|12%\|\|40% |
| $p_{v,MSE}; p_{v,MAE}$ | | | 0.30; 8.1e−2 |
| DYN VS PRO $\mid H < L$ | 12\|\|8\|\|18 | 31\|\|32\|\|53 | 17\|\|6\|\|5 |
| % | 32%\|\|21%\|\|47% | 27%\|\|27%\|\|46% | 61%\|\|21%\|\|18% |
| $p_{v,MSE}; p_{v,MAE}$ | 0.77; 0.81 | 0.89; 0.85 | **2.0**e−**2**; 0.11 |

Table 14: Detailed results of RQ2 post-processing models relative performance with setup conditioning against their vanilla version with p-values $p_v$ (in **bold** if less than 0.05). For the counts and percentages, the order corresponds to the scores where: post-processing version is (Better || Iso || Worse) than vanilla.

| Model | iTransformer | PatchTST | Crossformer |
|---|---|---|---|
| DYN-Post VS van. | 85\|\|19\|\|99 | 60\|\|7\|\|133 | 65\|\|2\|\|133 |
| % | 43%\|\|8%\|\|49% | 30%\|\|4%\|\|66% | 32%\|\|1%\|\|67% |
| $p_{v,MSE}; p_{v,MAE}$ | 0.18; **1.1**e−**2** | **1.0**e−**3**; **4.8**e−**5** | **7.9**e−**3**; **2.4**e−**4** |
| DYN-Post VS van. $\mid H > L$ | 32\|\|5\|\|21 | 18\|\|3\|\|33 | 24\|\|1\|\|31 |
| % | 55%\|\|9%\|\|36% | 33%\|\|6%\|\|61% | 43%\|\|2%\|\|55% |
| $p_{v,MSE}; p_{v,MAE}$ | 0.63; 0.66 | 0.15; **4.3**e−**2** | 0.46; 0.64 |
| DYN-Post VS van. $\mid H = L$ | 11\|\|5\|\|16 | 9\|\|0\|\|7 | 6\|\|0\|\|16 |
| % | 34%\|\|16%\|\|50% | 56%\|\|0%\|\|44% | 27%\|\|0%\|\|73% |
| $p_{v,MSE}; p_{v,MAE}$ | 0.30; 6.1e−2 | 0.77; 0.42 | 0.10; 0.12 |
| DYN-Post VS van. $\mid H < L$ | 42\|\|6\|\|62 | 33\|\|4\|\|93 | 35\|\|0\|\|87 |
| % | 38%\|\|6%\|\|56% | 25%\|\|3%\|\|72% | 29%\|\|0%\|\|71% |
| $p_{v,MSE}; p_{v,MAE}$ | 0.12; **4.2**e−**3** | **2.6**e−**4**; **3.1**e−**4** | **7.8**e−**3**; **2.0**e−**5** |

### E.4 ADDITIONAL RESULTS

We include here the setup conditioning for the comparison of RQ1 DYN added models against their vanilla versions. Distributions are shown in Figure 9, with detailed results (counts and p-values $p_v$) in Table 15. The performed unilateral Wilcoxon test is the same as in Section 4.2, conditioned by the setup.

We can observe that distributions are influenced by the setups but remain quite similar across each, without symmetry in performance for DYN models. It confirms the positive effect of added dynamics over the data length variation side-effect. In addition, there is a performance drop of DYN Informer, MICN, and FEDformer, for the $(H = L)$ setup, which are the three models where performance is driven by the added dynamics. As the DYN layer can be seen as a simple PRO feed-forward, it

is less impactful. The drop is particularly pronounced for FEDformer, where the recomputation of $\mathbf{X}_{trunc}$ seems to harm the performance, confirming that it does not give any data length advantage to FEDformer `DYN`.

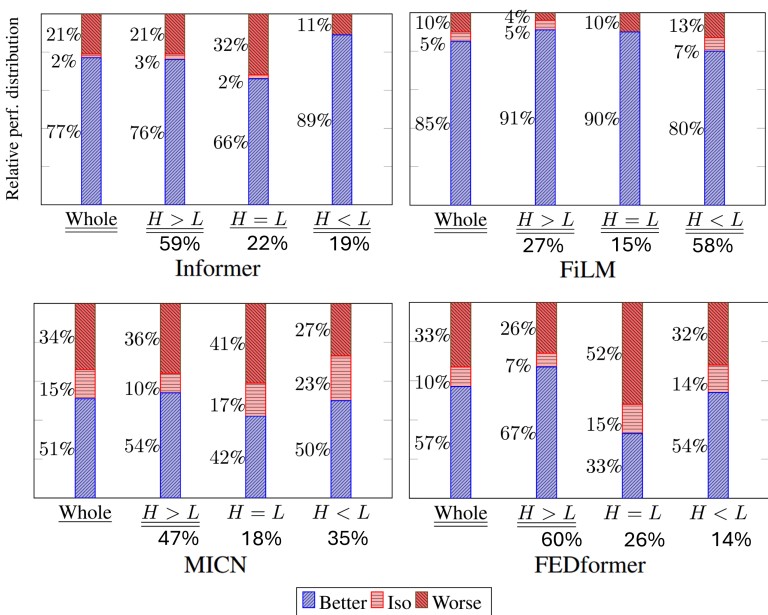

Figure 9: RQ1 `DYN` added models performance distribution against their vanilla version with setup conditioning. As in Figure 4, a setup is underlined (resp. double-underlined) when the `DYN` added model (left) is statistically better than its vanilla version on either MSE or MAE (resp. both). Setup distribution for each model is shown under the proper case.

.

Table 15: Detailed results of RQ1 `DYN` added models relative performance with setup conditioning against their vanilla version with p-values $p_v$ (in **bold** if less than 0.05). For the counts and percentages, the order corresponds to the scores where: `DYN` is (Better || Iso || Worse) than vanilla.

| Model | Informer | FiLM | MICN | FEDformer |
|---|---|---|---|---|
| `DYN` VS van. | 153\|\|4\|\|43 | 169\|\|11\|\|20 | 101\|\|31\|\|68 | 113\|\|20\|\|67 |
| % | 77%\|\|2%\|\|21% | 85%\|\|5%\|\|10% | 51%\|\|15%\|\|34% | 57%\|\|10%\|\|33% |
| $p_{v,MSE}$; $p_{v,MAE}$ | **4.0**e$-$**8**; **9.7**e$-$**11** | **7.6**e$-$**11**; **1.2**e$-$**11** | **1.3**e$-$**2**; 6.5e$-$2 | **3.8**e$-$**3**; **2.3**e$-$**2** |
| `DYN` VS van.\|$H > L$ | 90\|\|3\|\|25 | 49\|\|3\|\|2 | 51\|\|9\|\|34 | 81\|\|8\|\|31 |
| % | 76%\|\|3%\|\|21% | 91%\|\|5%\|\|4% | 54%\|\|10%\|\|36% | 67%\|\|7%\|\|26% |
| $p_{v,MSE}$; $p_{v,MAE}$ | **3.7**e$-$**5**; **8.4**e$-$**7** | **2.8**e$-$**5**; **6.3**e$-$**6** | **4.9**e$-$**3**; **4.9**e$-$**2** | **2.1**e$-$**4**; **2.0**e$-$**3** |
| `DYN` VS van.\|$H = L$ | 29\|\|1\|\|14 | 27\|\|0\|\|3 | 15\|\|6\|\|15 | 17\|\|8\|\|27 |
| % | 66%\|\|2%\|\|32% | 90%\|\|0%\|\|10% | 42%\|\|17%\|\|41% | 33%\|\|15%\|\|52% |
| $p_{v,MSE}$; $p_{v,MAE}$ | 5.6e$-$2; **9.9**e$-$**3** | **4.3**e$-$**4**; **1.3**e$-$**3** | 0.24; 0.71 | 0.65; 0.73 |
| `DYN` VS van.\|$H < L$ | 34\|\|0\|\|4 | 93\|\|8\|\|15 | 35\|\|16\|\|19 | 15\|\|4\|\|9 |
| % | 89%\|\|0%\|\|11% | 80%\|\|7%\|\|13% | 50%\|\|23%\|\|27% | 54%\|\|14%\|\|32% |
| $p_{v,MSE}$; $p_{v,MAE}$ | **5.9**e$-$**4**; **1.3**e$-$**5** | **2.5**e$-$**5**; **6.5**e$-$**6** | 0.38; 0.18 | 0.35; 0.36 |

# F PREDICTION VISUALIZATIONS

In this section, for each model, predictions on three different setups (dataset and horizon length) are shown, where the performance of RQ1 DYN added models and RQ2 post-processing models, in blue, are compared on the same sample to their vanilla version (on the left) and, for RQ1 models, to their PRO version (on the right), both in orange. The target prediction is in a dotted gray line. We chose two setups where the vanilla model performs well and one setup where it struggles more, to visualize different starting point situations. The dataset name and the horizon length $H$ are specified in the right column.

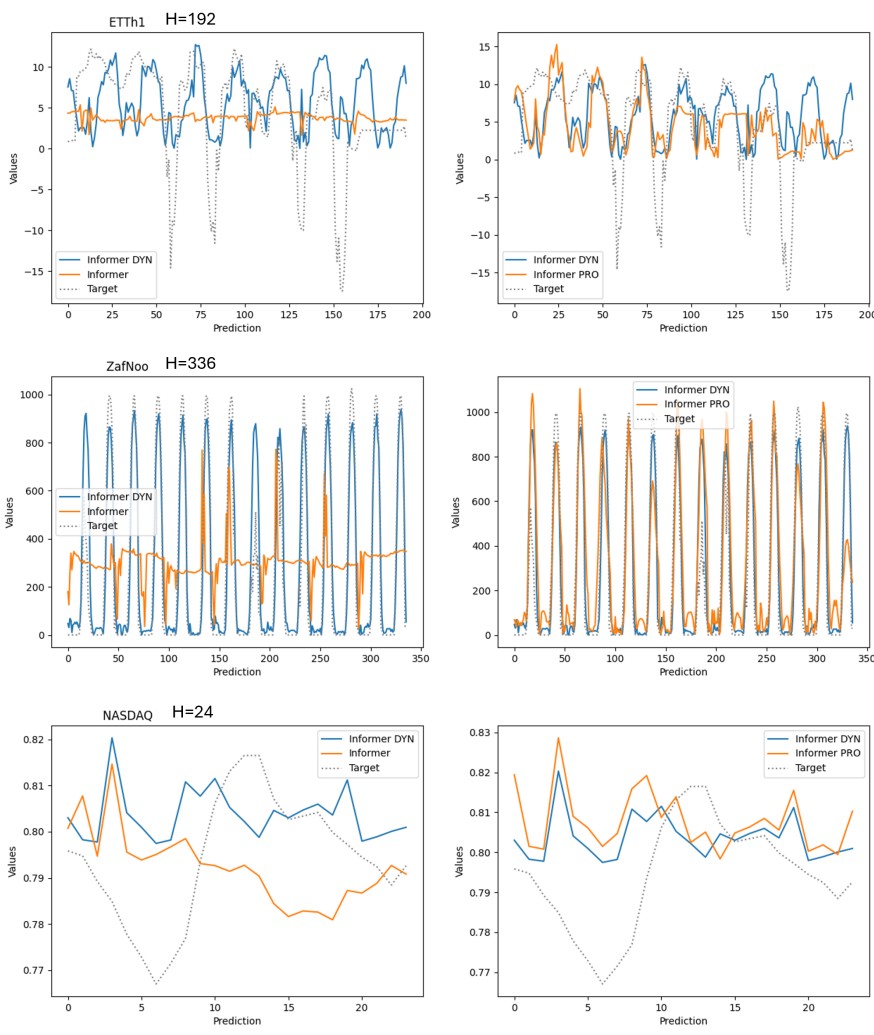

Figure 10: Informer prediction examples.

.

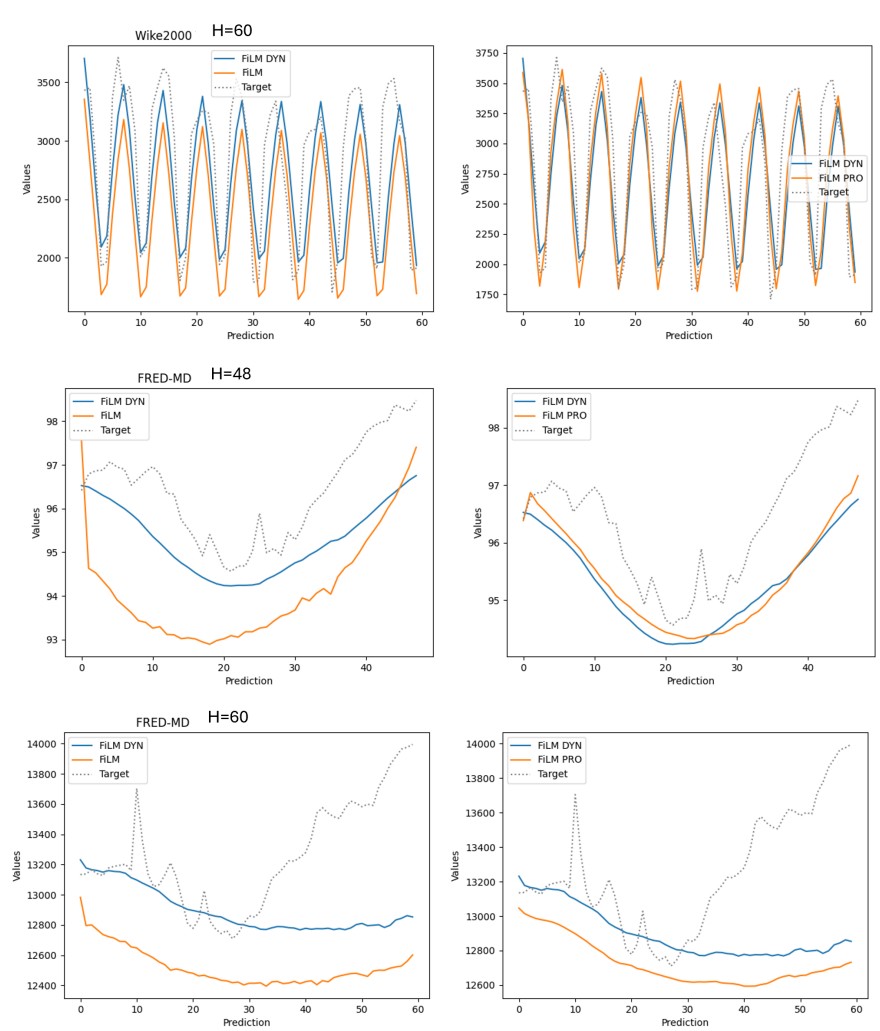

Figure 11: FiLM prediction examples.
.

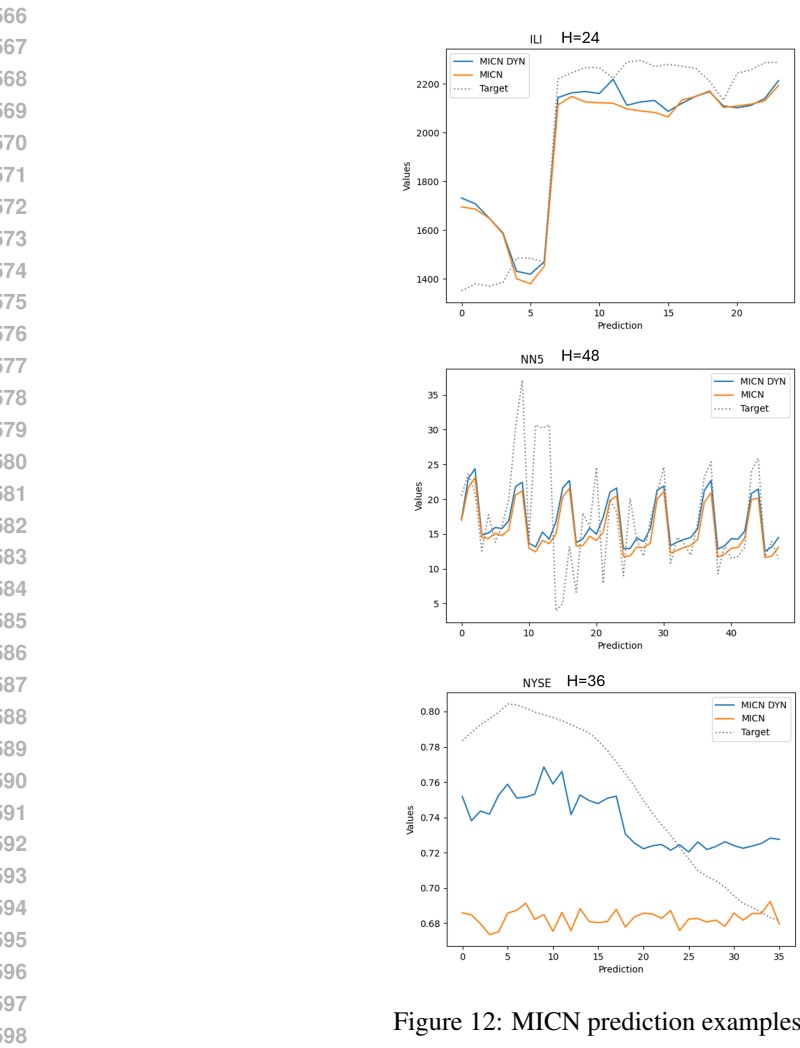

Figure 12: MICN prediction examples.
.

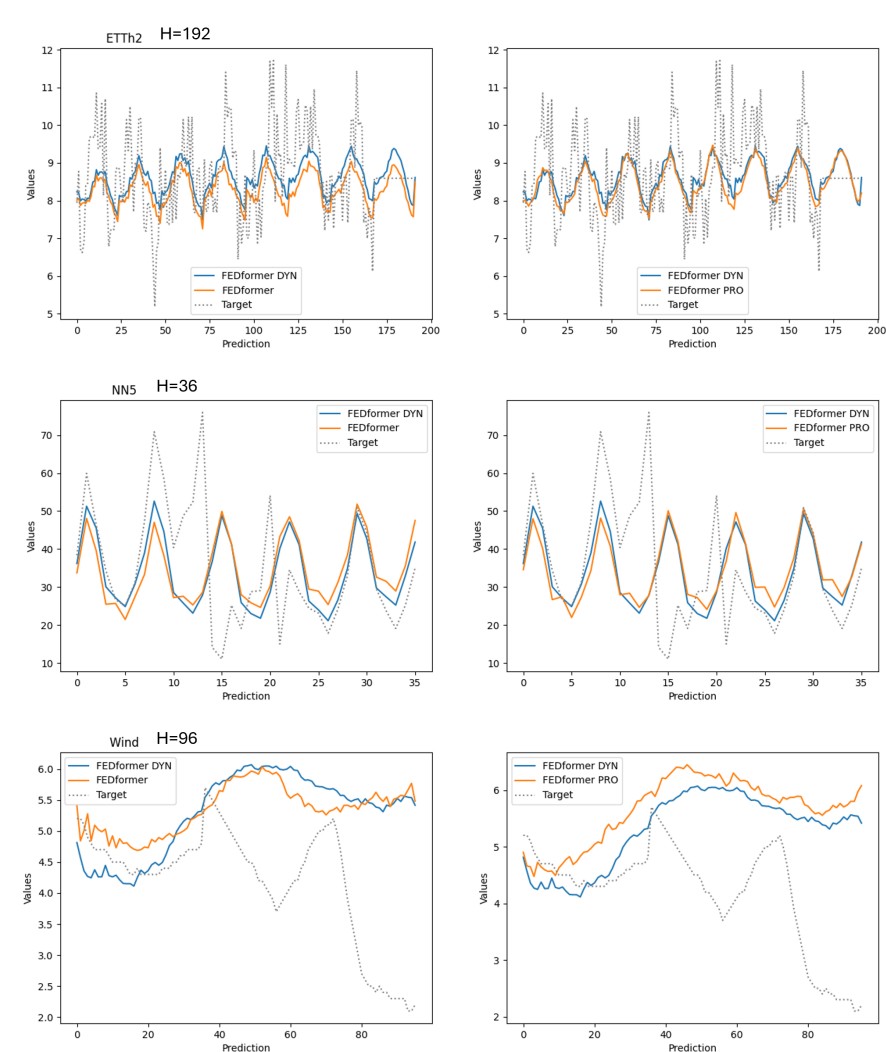

Figure 13: FEDformer prediction examples.

.

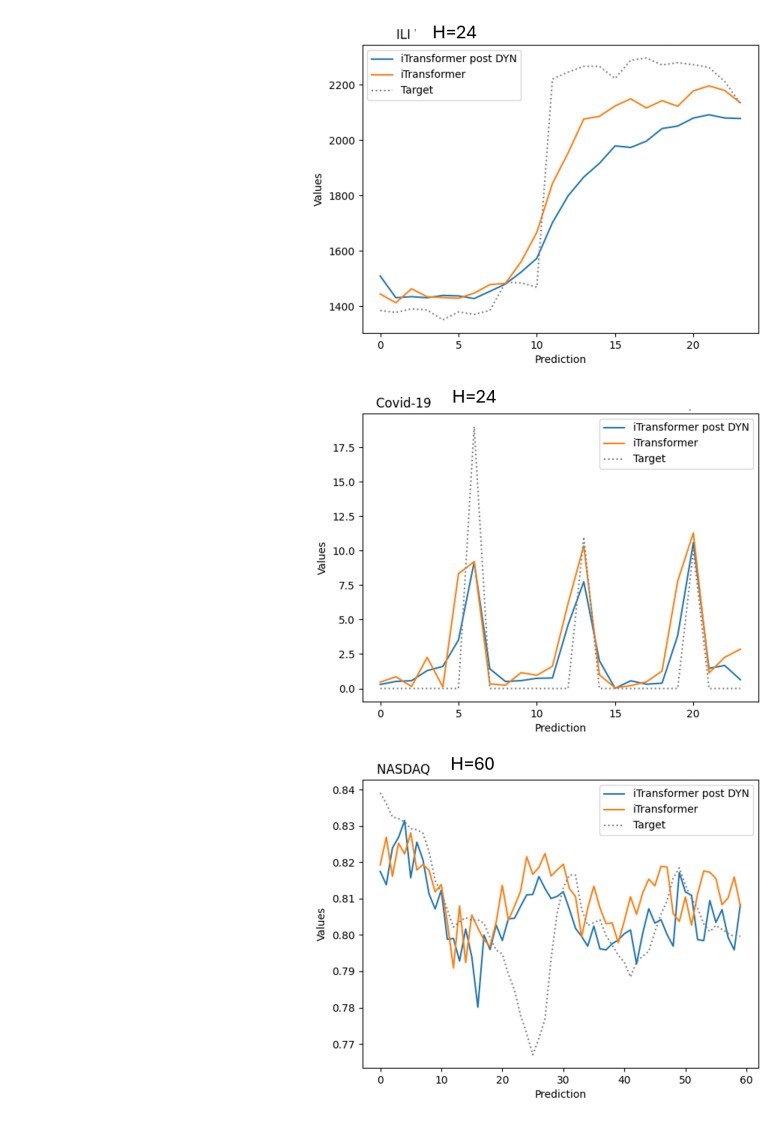

Figure 14: iTransformer prediction examples.

.

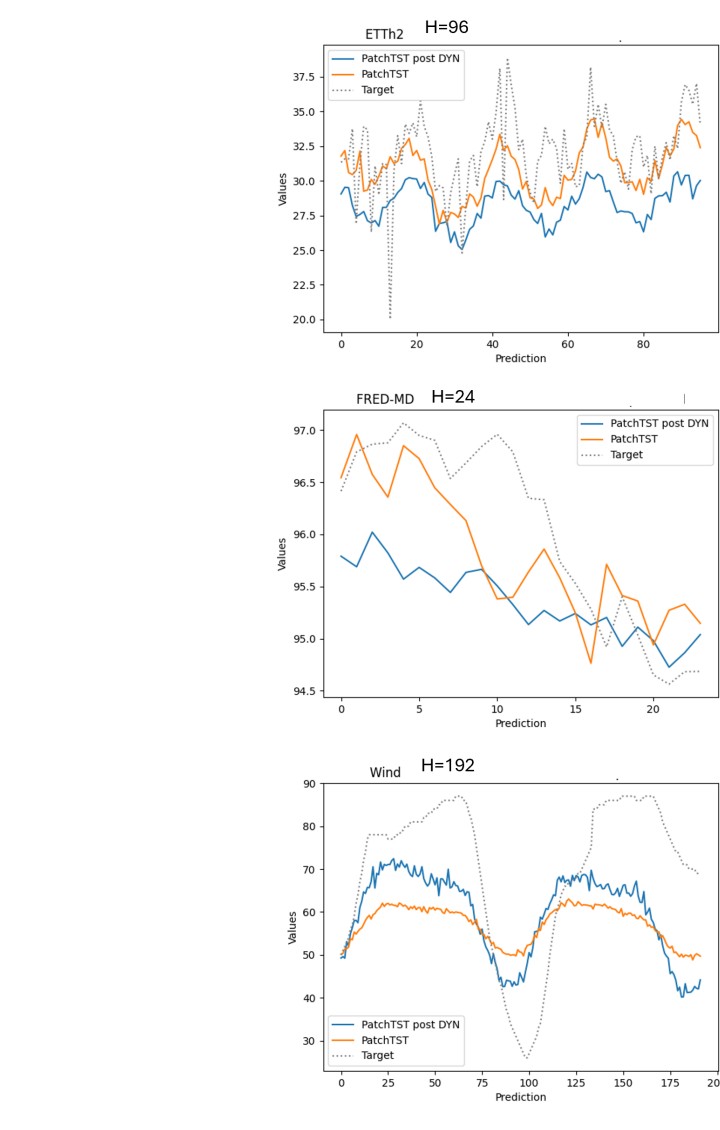

Figure 15: PatchTST prediction examples.
.

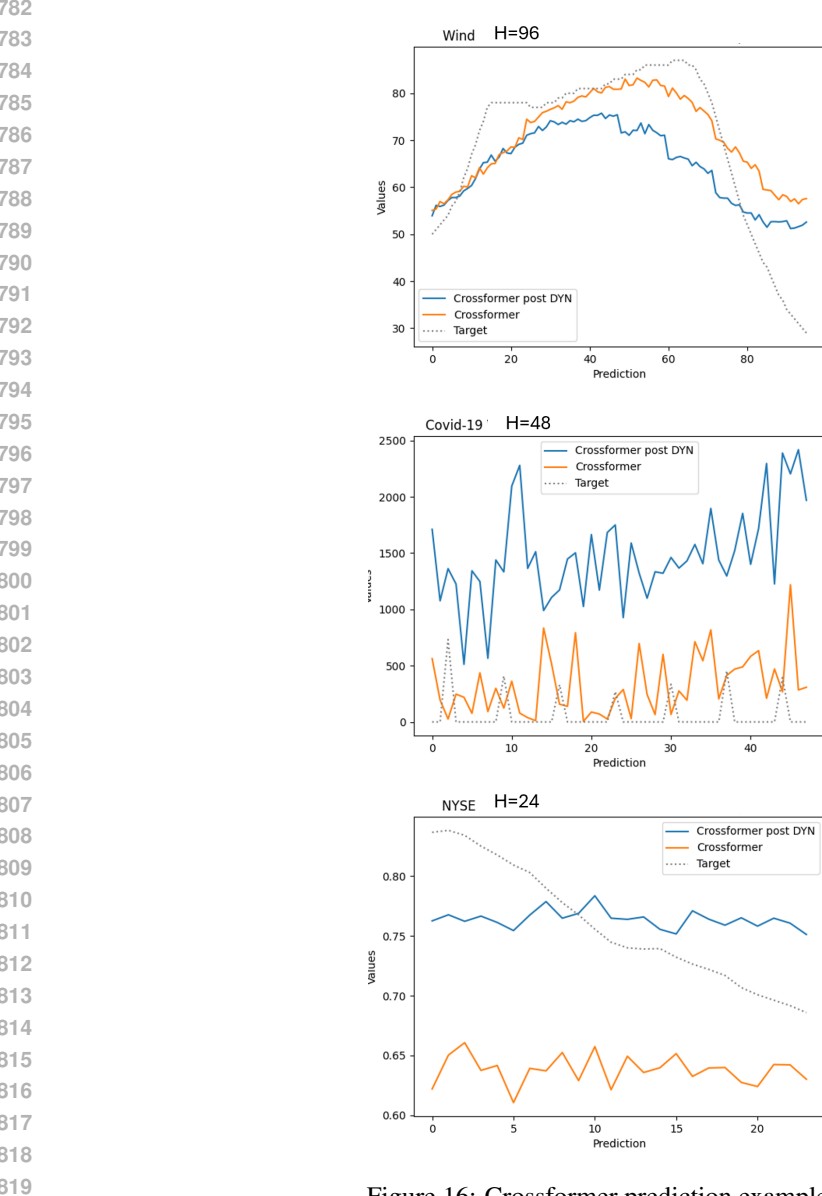

Figure 16: Crossformer prediction examples.
.

