# OpenReview forum: "Dynamics is what you need for time-series forecasting!"
_ICLR.cc/2026/Conference — Submitted to ICLR 2026_

### Official Review · Reviewer_63RZ · 2025-10-20

**Soundness:** 1
**Presentation:** 1
**Contribution:** 1
**Rating:** 0
**Confidence:** 3

**Summary:**

The paper hypothesizes that effective time-series forecasting requires models that can learn underlying time series dynamics and introduces the PRO-DYN framework to analyze this.

**Strengths:**

- This paper tackles an interesting and important question: whether time-series models can effectively learn temporal dynamics and presents a novel framework to try to analyze this.
- The authors’ decision to open-source their code in the supplemental material is commendable.

**Weaknesses:**

Overall, the paper is convoluted and difficult to parse, with sections that could benefit from improved organization and figure clarity. The arguments supporting the main hypothesis are not entirely convincing and could be strengthened with more formal theoretical formulations. Specific points are detailed below.
- Lines 122–125: The discussion of dynamical systems is very brief, limited to just two sentences, and the treatment of time-series dynamics is insufficient. A more thorough discussion is warranted, given that dynamics are central to the paper’s focus.
- Lines 56–57: The statement, “We thus hypothesize that TSF models should be able to learn time-series dynamics,” is insufficiently supported by the preceding text. The authors should clarify whether they mean that a “good” TSF model should learn time-series dynamics and explicitly define what is meant by this term before stating the hypothesis.
- Line 72: The paper references iTransformer, PatchTST, and Crossformer as “SOTA foundation models,” but these are not true TSFMs. TSFMs are typically pretrained on large-scale, cross-domain time-series datasets (e.g., Moment, Chronos, TinyTimeMixers, Moirai, LagLlama). While PatchTST is reasonable as a strong baseline, other cited models are not representative of SOTA time-series forecasting. It is also worth noting that simpler models, such as DLinear, have outperformed some transformer variants in long-term forecasting.
- Lines 72 and 97: References to a “linear dynamics layer” or “linear functions” are too broad. For instance, PatchTST uses a linear layer to project patched time-series tokens. The authors should clarify how their proposed linear layer differs from such projection layers.
- Line 72: The placement of the dynamic linear layer seems inconsistent across architectures (e.g., after the encoder in Informer versus first in FiLM, according to Fig. 2). The rationale behind these placement choices should be explicitly explained.
- Line 203: The claim that “LSTF-Linear models do have learnable dynamics modeling capabilities” is not well-supported with evidence.
- Figure 1: The meaning of the pink and white rectangular boxes is unclear, as are the outputs from the functions $f_\theta$., which are represented as non-annotated empty boxes. If the final output represents a forecast, it should be denoted as $\hat{y}$ instead of $y$.
- Table 1: It is unclear what “better than NLinear” refers to and why NLinear was chosen as the reference. The rationale should be explicitly stated, particularly if the claim relates to the LSTF-Linear models’ learnable dynamics.
- Evaluation (lines 353–355): The evaluation method is unconventional and non-intuitive: “We count the number of cases where the modified models are better, equal (iso), worse by at most 1% (low degradation), or worse by at least 1%, than their vanilla version.” A more standard and interpretable approach would be to report percent changes in error metrics across datasets between the original (vanilla) and modified models, ideally in a table.

Other:
- The abbreviation “LSTF” is not defined. It appears to refer to long-term time-series forecasting, but this should be explicitly stated.

**Questions:**

1. Have the experiments been repeated across multiple random seeds with metric results reported as averages?

---

> ### Author Response · Authors · 2025-11-28
>
> Dear reviewer 63RZ, thanks for your insightful review. In the following, we reply to both the Weaknesses section and the questions.
>
> ### Weaknesses
>
> - Lines 122–125: While being brief, we tried to be **concise** in the dynamical systems definition, showing that it is a function that allows for **going forward in time** from the current state. We will add lines on discrete dynamical systems to ease the link with a linear layer for prediction (please see the global response).
>
> - Lines 56–57: This hypothesis is based on the following assumptions: **A.** generative models should reproduce the underlying mechanism which generates the studied data (supported by works made on the textual modality). **B.** The underlying generative mechanism for the time-series data we study can be modelled by a dynamical system (supported by works that integrate prior knowledge on time-series data). Combining **A** and **B**, we derive our main hypothesis, which means that **a TSF model should be able to approach a dynamics when generating the predictions to be effective**. We then support it with the introduced nomenclature, the alignment between observations from the nomenclature and model performance, and the experiments.
>
> - Line 72: As we study dynamics learning, we consider models that are trained only on one dataset from scratch. Studying large-scale foundational models would be interesting for future work indeed. The fact that DLinear has outperformed transformer variants is the core of our analysis: these variants lack dynamics modelling capabilities, whereas DLinear possesses them (see Section 3.2).
>
> - Lines 72 and 97: In line 97, we should have specified: '*to support the integration of Linear functions* **as the predictors** '. What we mean by linear functions is mentioned in the introduction and is clarified in the following sections. As you mention, in PatchTST, linear layers are applied along the temporal dimension in the encoder. With our nomenclature, we consider these blocks as $\texttt{PRO}$, as they do not explicitly map the temporal dimension to the prediction dimension, nor map their input to a prediction, but rather on a latent, arbitrary dimension. **We can see them as down our up-sampling performed on the observed time interval**.
>
> - Line 72: We studied multiple ways to add the linear $\texttt{DYN}$ layer. The $\texttt{DYN}$ layer takes an input living in the historical time interval. For Informer and FEDformer, we then had two options: before or after the encoder. We tested both designs and selected the best one for each configuration. For FiLM, we also had the same two options: before or after the whole FiLM block. The latter option involved adjusting the hyperparameters of the original model too much. Thus, comparing vanilla FiLM with the latter modification would not have been meaningful. That is why we put the $\texttt{DYN}$ layer before.
>
> - Line 203: We propose an alternative to Section 3.2 to address concerns on the identification of a linear $\texttt{DYN}$ layer to a dynamical system. Please find it in the global response. In short, LTSF-Linear models can be viewed as a **relaxed version of a discrete time-delay dynamical system** where its evolution law is applied $H$ times (where $H$ is the prediction length).  This formalisation facilitates the analogy between a linear layer and a discrete dynamical system.
>
> - Table 1: We explain in lines 211-212 why NLinear is chosen as the reference: '*chosen as the reference as it is the best performing simple model*'. It is also supported by the introduction, where we recall that LTSF-Linear models have outperformed transformer variants, and Section 3.2, where we demonstrate that LTSF-Linear models, including NLinear, approach a discrete dynamical system. This section will be updated in accordance with the revisions proposed in the global response.
>
> - Evaluation (lines 353–355): We will add this comparison to our work.
>
> - Thanks for pointing this out. We will explicitly reference "LTSF" to long-term time-series forecasting.
>
> ### Questions
>
> 1. We performed only one run for each case when the best set of hyperparameters was found in order to compare them in the same conditions as in the TFB benchmark [1].
>
> [1] Qiu, X., Hu, J., Zhou, L., Wu, X., Du, J., Zhang, B., Guo, C., Zhou, A., Jensen, C. S., Sheng, Z., \& Yang, B. (2024). TFB : Towards Comprehensive and Fair Benchmarking of Time Series Forecasting Methods. arXiv.org. https://arxiv.org/abs/2403.20150

---

### Official Review · Reviewer_mnFs · 2025-10-28

**Soundness:** 2
**Presentation:** 2
**Contribution:** 2
**Rating:** 4
**Confidence:** 4

**Summary:**

This paper analyzes existing time-series forecasting (TSF) models, aiming to address why simple linear models often outperform complex deep learning models in this domain. The authors posit that TSF tasks necessitate models capable of learning the underlying dynamics of the data. To investigate this, a "PRO-DYN" nomenclature is introduced, decomposing models into processing (PRO) and dynamics (DYN) units . Two primary insights emerge from this analysis: 1) Models underperforming relative to linear counterparts often lack a DYN module or possess only a partially learnable one; 2) The positioning of the DYN module within the model architecture is crucial, with placement at the terminal end generally yielding optimal performance. Extensive experiments are conducted to substantiate these claims .

**Strengths:**

1. The paper introduces an original and insightful PRO-DYN nomenclature for deconstructing TSF models. This framework is effectively utilized to rationalize the "anomalous" effectiveness of current LSTF-Linear models and formulate two key observations regarding the role and placement of dynamics modules.
2. The authors provide substantial empirical evidence through extensive experiments to validate the proposed insights concerning the importance of the DYN module and its architectural positioning for model performance .

**Weaknesses:**

1. The paper's analysis of dynamics relies almost exclusively on adding or relocating a linear DYN layer to represent the temporal space transformation. While sufficient to demonstrate that incorporating some dynamics is superior to its absence (e.g., 0-padding in Informer ), it lacks the evidence to generalize this finding to the claim that "dynamics," specifically linear dynamics, is the optimal or necessary form.
2. The work primarily focuses on evaluating the importance of the dynamics module, offering valuable guidance, but the actual technical innovation presented is limited .
3. The exposition suffers from clarity issues. The formal definition of the TSF task is arguably over-complicated , yet the explanation of core concepts and arguments lacks sufficient mathematical formalization, relying instead on potentially ambiguous verbal descriptions, thereby increasing the difficulty of comprehension .

**Questions:**

1. The paper fails to provide a detailed explication of the crucial concept of "dynamics." While lines 343-344 draw an analogy to physics ("dynamics matrix," "external force"), the correspondence is not elaborated upon . How exactly these terms map to physical dynamics and why this analogy explains the performance of LSTF-Linear models remains unclear.
2. The primary empirical support rests on Figures 4 and 5. However, in the upper panel of Figure 5, the advantage of DYN added models over PRO added models is not consistently evident, particularly for the FiLM model. This suggests that the performance gains observed for FiLM in Figure 4 might stem largely from increased parameter count rather than the dynamics aspect itself , thus providing insufficient support for RQ1.
3. Figure 2 illustrates varied designs for implementing the DYN added modifications across different models (e.g., replacing 0-padding vs. adding at the entry) . Does a unified principle or rationale underpin these differing design choices, or are they ad-hoc modifications?

---

> ### Author Response · Authors · 2025-11-28
>
> Dear reviewer mnFs, thanks for your insightful review. In the following, we reply to both the Weaknesses section and the questions.
>
> ### Weaknesses
>
> 1. With our paper, we do not intend to demonstrate that the linear block is the optimal way to model dynamics with neural architectures. We identify that LTSF-Linear and the majority of well-performing Transformer-based models employ a linear layer as their dynamics block. Thus, we implement it in time-series forecasting (TSF) models without dynamics modelling capabilities to assess whether it helps to achieve better results, which is the case. In our next work, we plan to consider other $\\texttt{DYN}$ functions, such as autoregressive ones, and include dynamics-based models (please see global response), and compare them. Indeed, **our work here aims to demonstrate that dynamics modelling is important and drives the performance**.
>
> 2. In this work, the innovation does not rely on a technical contribution, but on a methodology contribution. We propose a novel approach to TSF models and their computational blocks. While the linear $\texttt{DYN}$ block remains quite simple, **its identification as a discrete dynamics system unveils its expressiveness** (please see global response).
>
> 3. The challenge was to introduce the notion of time interval in the TSF task, which seems not to have been done yet in other TSF papers. We will work on clarifying this. As suggested by reviewer Z5jK, we plan to add a glossary to clarify the current notations while simplifying them.
>
> ### Questions
>
> 1. We propose an alternative to Section 3.2 to address concerns on the identification of a linear $\texttt{DYN}$ layer to a dynamical system. Please find it in the global response. In short, LTSF-Linear models can be viewed as a **relaxed version of a discrete time-delay dynamical system** where its evolution law is applied $H$ times (where $H$ is the prediction length). Based on the hypothesis that TSF models should model dynamics when generating new predictions and experimental results, we try to support that this analogy explains why LTSF-Linear models get strong results while being quite simple.
>
> 2. We agree with your remark on FiLM and propose the same analysis in our paper (lines 460-461). We also suppose that the added dynamics get in conflict with the one defined by the SSM in the encoding part.
>
> 3. We studied multiple ways to add the linear $\texttt{DYN}$ layer. The $\texttt{DYN}$ layer takes an input living in the historical time interval. For Informer and FEDformer, we then had two options: before or after the encoder. We tested both designs and selected the best one for each configuration. For FiLM, we also had the same two options: before or after the whole FiLM block. The latter option involved adjusting the hyperparameters of the original model too much. Thus, comparing vanilla FiLM with the latter modification would not have been meaningful. That is why we put the $\texttt{DYN}$ layer before.

---

### Official Review · Reviewer_Z5jK · 2025-10-31

**Soundness:** 3
**Presentation:** 2
**Contribution:** 3
**Rating:** 4
**Confidence:** 3

**Summary:**

This paper bring in a nomenclature for time series forecasting (TSF) methods and comes up with interesting hypothesis such as "under-performing architectures learn dynamics at most partially" and "the location of the dynamics block at the model end matters", thus justifying the importance of the title, dynamics is very important for TSF. Their experiments show that adding dynamics learning blocks to a normal model enhances it's performance on TSF.

**Strengths:**

- The idea of bringing in nomenclature to TSF methods is good. As noted by the authors, TSF methods are somehow a bit different compared with mainstream LLMs but they might not be, and such discussion will help researchers understand if any differences exist.
- The research questions are well motivated and the experiments have answered them to good detail.

**Weaknesses:**

**Expanding nomenclature**: The nomenclature inclusion of TSF methods which incorporate some sort of dynamical systems is missing in Table 1. I’ve seen the authors cite Attraos [1], but they didn’t include it in Table 1. In addition to that, please add other latest dynamical-based TSF methods such as DeepEDM [2], Koopa [3], KNF [4] to your Table 1. In case a method can’t be nomenclature using PRO-DYN, the authors should explicitly mention the limitations associated with it on why it's not possible.

**Inclusion of dynamical-based TSF methods in experiments**: Following previous point on nomenclature expansion, pick one/few dynamical-based TSF methods and supplement the experiments. A potential RQ3 that I am interested is something like the following — Whether Informer-DYN will be better than already built-in dynamical-based TSF method i.e if adding DYN block to a normal baseline will make the model better compared with already built-in dynamical-based TSF methods. This sort of expands RQ1. Specifically, it's interesting to know the tradeoffs associated. Feel free to make certain choices to speed up experimentation and compute limitations. Eg: Picking up latest models might be enough to compare against.

1. Attraos (NeurIPS 2024) - https://openreview.net/forum?id=fEYHZzN7kX
2. DeepEDM (ICML 2025) - https://openreview.net/forum?id=LLk1qYQatJ
3. Koopa (NeurIPS 2023) - https://openreview.net/forum?id=A4zzxu82a7
4. KNF (ICLR 2023) - https://openreview.net/forum?id=kUmdmHxK5N

**Questions:**

I personally felt the presentation can be made much better.

- Figure 1 can be made much better. I had a hard time understanding the flow of information.
- The mathematical notations can be simplified further. Consider adding a Glossary of notations.
- Table 13, 14, 15 - The presentation can be much better. Use sub columns/subrows to represent complex tables.

---

> ### Author Response · Authors · 2025-11-28
>
> Dear reviewer Z5jK, thanks for your insightful review. In the following, we reply to both the Weaknesses section and the questions.
>
> **Expanding nomenclature** Thanks for this suggestion. We have included in the general response an additional analysis of dynamics-based models with the $\texttt{PRO-DYN}$ nomenclature, with specific comments on some models.
>
> **Inclusion of dynamical-based TSF methods in experiments** Thank you for this insightful suggestion. In our opinion, your proposed RQ3 would better fit within our future work, where we plan to **explore the diversity of dynamics modelling** (including autoregressive mechanisms for generation, for instance, which are involved in some dynamics-based models you mentioned). In this, we would analyse in detail these tradeoffs across the different $\texttt{DYN}$ blocks.
>
> We will take into account your remarks on the clarity of the figures and tables. We will put in efforts to make them clearer.

---

### Official Review · Reviewer_jLLD · 2025-11-01

**Soundness:** 2
**Presentation:** 3
**Contribution:** 2
**Rating:** 2
**Confidence:** 5

**Summary:**

The paper “Dynamics Is What You Need for Time-Series Forecasting!” argues that the key to improving time-series forecasting (TSF) is not architectural complexity but explicitly learning underlying dynamics. The paper introduces the PRO-DYN framework, which decomposes models into processing (PRO) functions that operate within the same time interval and dynamics (DYN) functions that map past to future. Analyzing many TSF architectures, they find that top-performing models share two traits: a fully learnable dynamics block and its placement at the model’s end.

**Strengths:**

The paper offers an interesting and conceptually unifying perspective on time-series forecasting (TSF) by reframing existing architectures through the lens of dynamics learning. The proposed PRO-DYN framework provides a clear and intuitive decomposition of model components into processing and dynamics functions, which helps explain why certain architectures perform better. The empirical results are broad and consistent across multiple benchmarks. The work is well-organized, and its emphasis on the role of dynamics could inspire future studies that bridge traditional dynamical systems theory with modern deep learning approaches.

**Weaknesses:**

While the conceptual framing is novel, the paper lacks depth in both theoretical and empirical validation of what it calls “dynamics.” The introduced DYN block is essentially a simple temporal linear or MLP layer, and no evidence is provided that it actually learns or represents true system dynamics. The analysis remains largely phenomenological rather than mechanistic. Moreover, the paper does not evaluate or compare against existing dynamics-based models such as Koopman, Neural ODEs, PINNs, or state-space models, that explicitly learn or approximate underlying temporal evolution laws.  Dynamics is the main focus of the paper, so naturally, one would expect a more detailed discussion on work in that domain as well as some baselines in the main results. Some representative works include:

- Lusch, B., Deep learning for universal linear embeddings of nonlinear dynamics. Nat Commun 9, 4950 (2018).
- Liu, Y., et al. Koopa: Learning non-stationary time series dynamics with Koopman predictors. NeurIPS 2023.
- Hu, Jiaxi, et al. Attractor memory for long-term time series forecasting: A chaos perspective. NeurIPS  2024: 20786-20818.
- Majeedi, A., et al. LETS Forecast: Learning embedology for time series forecasting. ICML 2025.

Finally, to convincingly support the claim that “dynamics is what you need,” the paper should have included experiments on synthetic or controlled dynamical systems, where the true underlying evolution laws are known. Such settings would allow testing whether the proposed DYN block can actually learn or approximate genuine system dynamics, rather than merely fitting temporal statistical correlations in real-world data. Instead, the experiments are limited to standard benchmark datasets, which, while diverse, do not provide ground truth about underlying dynamics, making it difficult to validate the central hypothesis in a rigorous way.

Furthermore some of the claims are confusing, for instance, iTransformer, by inverting the inputs, essentially applies a temporal MLP which is equivalent to the DYN layer. Further more, DLinear also relies on a temporal linear layer, does that mean DLinear learns all the dynamics? More clarification on these would greatly strengthen the support for the claims in the paper.

**Questions:**

1) How do we know that the DYN block is learning any dynamics?
2) Why are none of the Dynamics based models compared in the experiments? For instance to test if adding the DYN block helps in those cases or not.
3) More discussion on the related work in the dynamics space would greatly improve the context of this work.
4) Some more clarity on the scope and claims would be helpful.

---

> ### Author Response · Authors · 2025-11-28
>
> Dear reviewer jLLD, thanks for your insightful review. In the following, we reply to both the Weaknesses section and the questions.
>
> Our work introduces a **novel nomenclature to analyze and understand how time-series forecasting (TSF) models are built from a dynamical perspective**. We propose a new way to approach existing computing units rather than introducing a novel computing unit. In this first work, we focus on a simple linear layer used as a prediction layer. Our main contribution is to unveil the **expressiveness of such a simple computing block**. It starts from identifying the dynamics learnt by LTSF-Linear models, which are essentially a linear layer surrounded by basic computations. We propose an alternative to Section 3.2 to address concerns on the identification of a linear $\texttt{DYN}$ layer to a dynamical system. Please find it in the global response. In short, LTSF-Linear models can be viewed as a **relaxed version of a discrete time-delay dynamical system** where its evolution law is applied $H$ times (where $H$ is the prediction length). This formalisation facilitates the analogy between a linear layer and a discrete dynamical system.
>
> Building on this, the goal of our analysis is to assess whether dynamics plays a crucial role in the TSF task and **incorporate this inductive bias into general prediction models** designed without any a priori knowledge. We did not include dynamics-based models in our experiments as they are already based on dynamics considerations. We have included in the general response additional dynamics-based models analysis with the $\texttt{PRO-DYN}$ nomenclature. Comparing the performances of such models with those in our paper would be suitable for our future work, where we identify and compare multiple $\texttt{DYN}$ functions (such as autoregressive ones, which are included in some dynamics-based models you mentioned).
>
> Indeed, as you demonstrate, evaluating our method on synthetic datasets would strengthen the analysis. System identification would be possible when the inputs and outputs of the $\texttt{DYN}$ layer are of the same nature. In the case of the $\texttt{PRE-DYN}$ configuration, inputs are latent variables while outputs are predictions in the original space. The linear $\texttt{DYN}$ layer goes back to the input space while going forward in time. We can view the encoder as linearizing the dynamics between historical latent variables and the predicted ones. System identification would then be less direct. In addition, as we show that the linear $\texttt{DYN}$ layer learns a relaxed version of a discrete dynamical system, overall, identification remains challenging.
>
> As you mention, in iTransformer, linear layers are applied along the temporal dimension in the encoder. With our nomenclature, we consider these blocks as $\texttt{PRO}$, as they do not explicitly map the temporal dimension to the prediction dimension, nor map their input to a prediction, but rather on a latent, arbitrary dimension. We can see them as down our up-sampling computations performed on the historical time interval. DLinear does learn dynamics as it belongs to the LTSF-Linear family (mentioned in Section 3.2), so it does not contradict our developments.

---

### Author Response · Authors · 2025-11-28
**Global response (1/2)**

We thank all the reviewers for their consideration of our work and their valuable feedback, suggestions and questions. We address several concerns shared among the reviewers.


## 1.Revised version of LTSF-Linear dynamics to clarify how they relate to dynamical systems (reviewers jLLD, mnFs, 63RZ)


We propose a revised development of Section 3.2 **LTSF-Linear dynamics** here [1]. This would help to identify such models to dynamical systems.

**(Defintion 1.1) Discrete time-delay dynamical systems** Based on a current system state $\mathbf{x}(t) \in \mathbb{R}^D$ at time $t\in\mathbb{R}^+$, let suppose we sample at a rate of $\\Delta \\in  \\mathbb{R}^{+*}$  such as $x(t_n) = x(n \\Delta), n \in \mathbb{Z}$. $x$ is governed by a discrete time-delay dynamical system if there exists a map $F:(\mathbb{R}^D)^K\to\mathbb{R}^D$, $K\in\mathbb{N}$, such that:
$$
x(t_n) = F(x(t_{n-1}),\dots,x(t_{n-K}))
$$

**(Definition 1.2) Discrete linear time-delay dynamical systems** Keeping the same notations as above, if $F$ is linear, we can define $M\\in \\mathbb{R}^{L\\times L}$, $L\in\mathbb{N}$ with $L>K$ such as:
$$
[x(t_{n}),\dots,x(t_{n-L+1})]^T = M[x(t_{n-1}),\dots,x(t_{n-L})]^T
$$
$M$ is the discrete dynamical matrix defining the system evolution of $L$ successive observations.

LTSF-Linear models, performing a prediction $\tilde{\mathbf{Y}}$ of the next $H$ steps from $L$ observations, can be viewed as a **relaxed version of a discrete linear time-delay dynamical system**, where its evolution law is applied $H$ times. Indeed, following notations introduced in our paper $\\mathbf{X}(t_L)=\\{x_d(t_1),\\dots,x_d(t_L)\\},d=1,\\dots,D$ with $D$ the number of channels (without loss of generality, $L>H$). Supposing the look-back window is long enough $(L>K)$, let have $\mathbf{X}$ governed by a linear dynamics $M\in\mathbb{R}^{L\times L}$ such that $\mathbf{X}(t_{L+1})=M\mathbf{X}(t_{L})$ (see definition 1.2 and [2] introducing the same framework in the TSF task). We then have $\mathbf{X}(t_{L+H})=M^H\mathbf{X}(t_{L})$. With $\\mathbf{Y}$ the true prediction, we have : $\\mathbf{Y}=(M^H)^{tr}\mathbf{X}$. Where $(M^H)^{tr}\in \mathbb{R}^{H\times L}$ corresponds to $M^H$ where the first $(L-H)$ rows are removed.

LTSF-Linear models, during training of their linear layer weight matrix $W_\theta$, are approaching the above truncated dynamics. Indeed, we can identify $W_\theta$ to $(M^H)^{tr}$: two matrices with the same dimensions having the same temporal impact on a group of historical values. By doing so, LTSF-Linear models can be viewed as a relaxed version of time-delay dynamical systems, as there are no constraints on the learnt coefficients. The bias term in the linear layer can be seen as a corrective step due to the absence of constraints.

This update of Section 3.2, with additional discussion on discrete dynamics time-delay systems, should **help to better understand the link between a linear layer for prediction and dynamical systems**.

[1] Zeng, A., Chen, M., Zhang, L., \& Xu, Q. (2022). Are Transformers Effective for Time Series Forecasting ?  arXiv.org. https://arxiv.org/abs/2205.13504

[2] Zhang, Y., Ma, L., Valkanas, A., Oreshkin, B. N., \& Coates, M. (2025). SKOLR : Structured Koopman Operator Linear RNN for Time-Series Forecasting. arXiv.org. https://arxiv.org/abs/2506.14113

---

> ### Author Response · Authors · 2025-11-28
> **Global response (2/2)**
>
> ## 2. Addition of dynamics-based models in the $\texttt{PRO-DYN}$ nomenclature (reviewers jLLD, Z5jK)
>
> We include here the analysis of well-known dynamics-based models into our nomenclature, as suggested by reviewers jLLD, Z5jK. Please find below an extension of our Table 1, where we consider dynamics-based models though the $\texttt{PRO-DYN}$ nomenclature:
>
> | Model         | Comp. learn. dyn. (RQ1) | Config. (RQ2) | DYN function | PRO backbone |  Reference |
> |---------------|-------------------------|----------------|--------------|--------------|--------------|
> | Attraos        | Yes                     | PRE-DYN        | Linear        | SSM          | [3]       |
> | Koopman-based | Yes                     | PRE-DYN        | AR   | MLP          | [4]       |
> | Koopa         | Yes                     | PRE-DYN-POST   | Linear+AR   | MLP              | [5]       |
> | KNF           | Yes                     | PRE-DYN-POST   | AR            | MLP, fixed     | [6]       |
> | DeepEDM       | Yes                     | DYN-POST       | MLP | MLP | [7]       |
>
>
> AR stands for autoregressive.
>
> **Overall remarks** Models learning an approximation of the linear Koopman operator [4,5,6] model it as a **simple linear layer**, which allows for going forward in time. Links between the Koopman operator and a linear $\texttt{DYN}$ could be interesting to explore.
>
> **Remarks on Koopa**:
> -  The output is the sum of a prediction from a complete learnable dynamics (called the time-invariant component) and a non-learnable one, derived from the output of an encoder (called the time-variant component). Still, the time-variant component is based on dynamical systems analysis, trying to approach local dynamics.
> -  In this model, the Koopman operator maps the input latent dimension to the same output one, where *time and variate dimensions are mixed* in this latent dimension.
>  - The $\texttt{DYN}$ function is the sum of a linear layer (time-invariant component) and an autoregressive one (time-variant component), thus identified as *Linear+AR*.
>
> **Remarks on KNF**: *fixed* in the $\texttt{PRO}$ backbone column refers to the selected set of transformations as inductive bias.
>
>
> [3] Hu, Jiaxi, et al. Attractor memory for long-term time series forecasting: A chaos perspective. NeurIPS 2024: 20786-20818.
>
> [4] Lusch, B., Deep learning for universal linear embeddings of nonlinear dynamics. Nat Commun 9, 4950 (2018).
>
> [5] Liu, Y., et al. Koopa: Learning non-stationary time series dynamics with Koopman predictors. NeurIPS 2023.
>
> [6] Wang, R., et al. Koopman Neural Operator Forecaster for Time-series with Temporal Distributional Shifts. ICLR 2023.
>
> [7] Majeedi, A., et al. LETS Forecast: Learning embedology for time series forecasting. ICML 2025.
>
> NB: In the paper, we wrote 'LSTF-Linear' instead of 'LTSF-Linear'. We will correct this typo.

---

### Meta-Review · Area_Chair_Knwo · 2025-12-27

**Summary:**

All four reviewers share the same concerns regarding the clarity of presentation and the depth of both theoretical and empirical validation. All these concerns require significant changes and improvements, many of which have not been fully addressed in the rebuttal and the revision. I thus recommend a rejection. The authors could further improve the paper for the next submission.

**Reviewer Concerns:**

1. Reviewer jLLD's main concern is the depth of the theoretical and empirical. I do not think the author's rebuttal solves this concern, particularly on "The analysis remains largely phenomenological rather than mechanistic. " Whether the proposed DYN block can actually learn or approximate genuine system dynamics is still a question in the air. I also agree with the reviewer's concern, which is the paper's main limitation.

2. Reviewer Z5jK's concern on adding more recent methods into consideration has been solved by the rebuttal.

3. Reviewer mnFs's first two concerns about the novelty and optimality of the dynamics module have been addressed. But the clarity of presentation has not been improved.

4. Some minor concerns of Reviewer 63RZ regarding the clarity of certain sentences have been solved. But the overal clarity and insufficent supporting evidence have not been addressed.

**Reviewer Scores:**

1. I do not think the Reviewer  jLLD will change his score even if he engages in a full discussion since his concern is about the depth of the paper's investigation.

2. Reviewer Z5jK might improve his score to weak accept since the concern has been solved by the rebuttal.

3. I am not sure about the Reviewer mnFs will or not change his score, since some of his concerns are still remaining.

4. Reviewer 63RZ gave a very low rating to this paper, 0, which concerned the clarity of the paper and the insufficient support of the arguments. And many of her concerns have not been addressed completely. Even if he or she will change the score, at most to 2.

---

### Decision · Program_Chairs · 2026-01-26

Reject